# Secretome translation shaped by lysosomes and lunapark-marked ER junctions

Heejun Choi[1], Ya-Cheng Liao[2], Young J. Yoon[3], Jonathan Grimm[1], Nan Wang[1], Luke D. Lavis[1], Robert H. Singer[3] & Jennifer Lippincott-Schwartz[1✉]

The endoplasmic reticulum (ER) is a highly interconnected membrane network that serves as a central site for protein synthesis and maturation[1]. A crucial subset of ER-associated transcripts, termed secretome mRNAs, encode secretory, lumenal and integral membrane proteins, representing nearly one-third of human protein-coding genes[1]. Unlike cytosolic mRNAs, secretome mRNAs undergo co-translational translocation, and thus require precise coordination between translation and protein insertion[2,3]. Disruption of this process, such as through altered elongation rates[4], activates stress response pathways that impede cellular growth, raising the question of whether secretome translation is spatially organized to ensure fidelity. Here, using live-cell single-molecule imaging, we demonstrate that secretome mRNA translation is preferentially localized to ER junctions that are enriched with the structural protein lunapark and in close proximity to lysosomes. Lunapark depletion reduced ribosome density and translation efficiency of secretome mRNAs near lysosomes, an effect that was dependent on eIF2-mediated initiation and was reversed by the integrated stress response inhibitor ISRIB. Lysosome-associated translation was further modulated by nutrient status: amino acid deprivation enhanced lysosome-proximal translation, whereas lysosomal pH neutralization suppressed it. These findings identify a mechanism by which ER junctional proteins and lysosomal activity cooperatively pattern secretome mRNA translation, linking ER architecture and nutrient sensing to the production of secretory and membrane proteins.

Historically, ER sheets have been considered the predominant site of secretome mRNA translation, largely owing to their enrichment in membrane-bound polysomes[5]. Yet, recent reconstructions from diverse mammalian cell types reveal that ribosomes, including polysome-associated and monosome-bound forms, are distributed across nearly all ER morphologies, encompassing sheets, tubules and tubule–tubule junctions[6]. Notably, a substantial subset of ER-bound ribosomes corresponds to non-translating subunits, particularly the 60S[7,8], suggesting that ribosome association is not synonymous with active elongation and may reflect regulatory or pre-initiation states. Definitive mapping of translation sites through Sec translocon components remains technically intractable because genetic tagging of core constituents such as Sec61α or TRAP often perturbs translocation activity, and immunolabelling approaches are hindered by steric inaccessibility of lumenal or membrane-embedded epitopes[9]. As a result, whether secretome mRNAs exhibit preferential translation in spatially restricted ER subdomains—potentially governed by compartmentalized translation-related factors[10–12]—or whether they are translated non-selectively across the continuous ER network remains an unresolved and compelling question. Here we address this question by tracking the dynamics and localization of individual secretome mRNAs in live cells. We demonstrate that active secretome mRNA translation is concentrated at ER junctions that are enriched in the structural protein lunapark (LNPK) and positioned adjacent to lysosomes. Loss of LNPK selectively reduces ribosome occupancy and translation efficiency of secretome mRNAs at these junctions by impairing eIF2-dependent initiation, an effect that can be rescued by the integrated stress response (ISR) inhibitor ISRIB. Translation at lysosome-proximal ER is further enhanced during amino acid starvation and suppressed by lysosomal pH neutralization, indicating that lysosomal activity locally regulates ER protein synthesis. Together, these findings identify LNPK-enriched ER junctions as organizational hubs where ER architecture and lysosomal signalling intersect to control secretome mRNA translation.

To track the behaviour of secretome mRNAs, we engineered fluorescent reporters by adapting MS2 mRNA tagging[13] to open reading frames encoding a variety of transmembrane and lumenal proteins. As a model, we constructed a mRNA encoding the N-terminal 45 amino acids of the Golgi-localized type II membrane protein sialyltransferase fused to EGFP in tandem with MS2-binding loops (*SiT-EGFP*) (Fig. 1a, inset). The mRNA forms a complex with MS2 protein (*SiT-EGFP*–MS2) and when translated, results in EGFP expression. In live cells, this reporter gave rise to a progressively Golgi-like EGFP distribution, consistent with translation on the ER followed by proper protein trafficking to the Golgi (Extended Data Fig. 1a).

To visualize single-molecule mRNA dynamics, we imaged *SiT-EGFP*–MS2 puncta in cells where the ER was unbiasedly highlighted using a fluorescently tagged, tail-anchored Sec61β reporter, which localizes to

[1]Janelia Research Campus, HHMI, Ashburn, VA, USA. [2]Columbia University, New York, NY, USA. [3]Albert Einstein College of Medicine, Bronx, NY, USA. ✉e-mail: lippincottschwartzj@janelia.hhmi.org

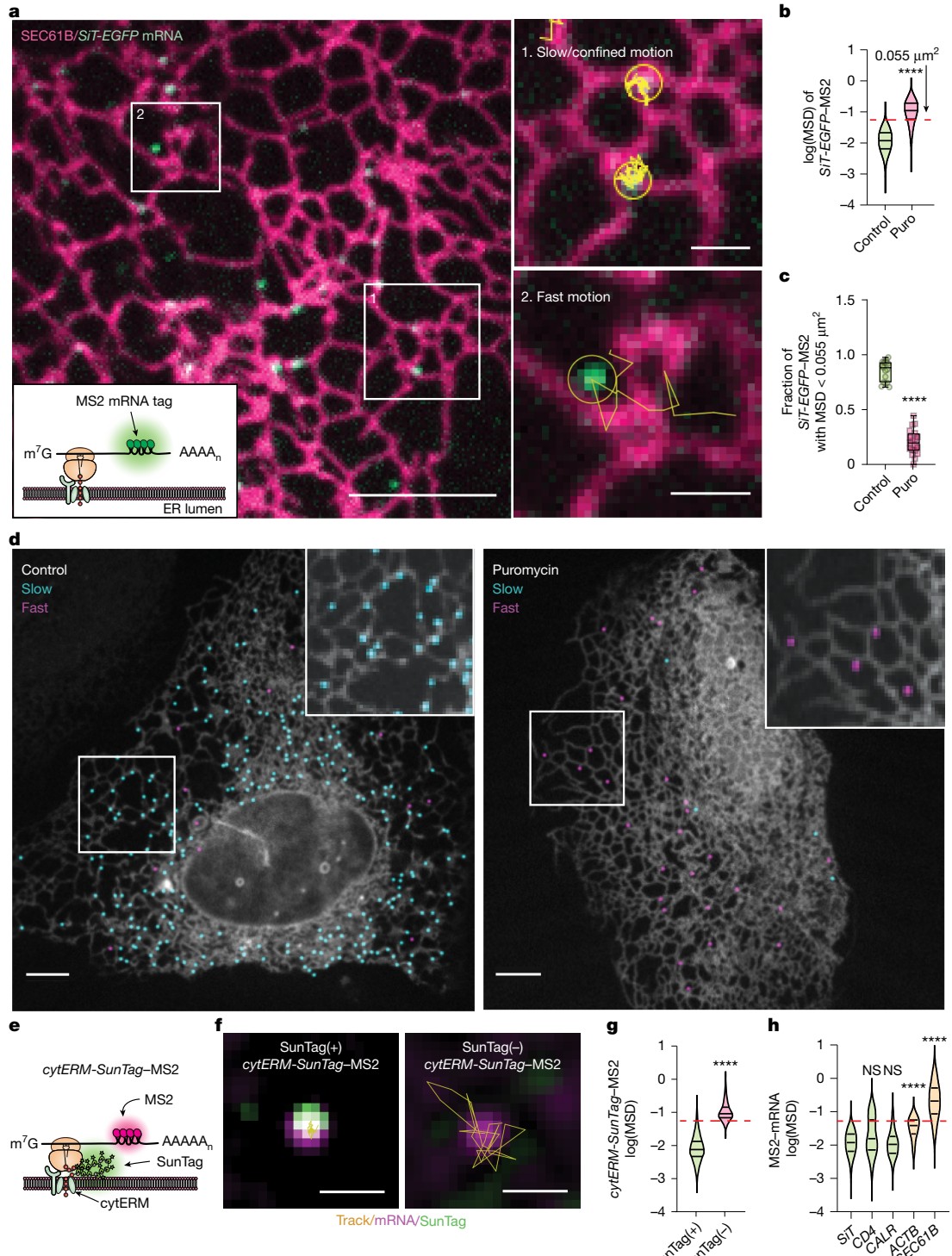

**Fig. 1 | Secretome mRNAs are translated on ER and show confined motion.**
**a**, Inset, design of secretome mRNA reporters (inset) containing a 5′ UTR, an open reading frame, 24 MS2 binding sites and a 3′ UTR. Main image, representative spinning-disk confocal microscopy of *SiT-EGFP* mRNA on ER labelled with SNAP–Sec61β. Scale bar, 5 µm. Right, magnified views of indicated regions showing trajectories of two confined mRNAs (top; yellow) or a rapidly moving mRNA (bottom; yellow). Scale bars, 1 µm. **b**, Violin plots of log($MSD_{\tau=1s}$) for *SiT-EGFP* mRNAs under control (*n* = 1,616) and puromycin (Puro)-treated (*n* = 1,608) conditions. Red dashed line indicates MSD = 0.055 µm². Dunnett's test, ****$P$ < 0.0001. **c**, Box plots showing fraction of slow-moving *SiT-EGFP* mRNAs in control (*n* = 17 cells) and puromycin-treated (*n* = 19) cells. Two-tailed unpaired *t*-test, ****$P$ < 0.0001. **d**, Control (left) and puromycin-treated (right) U-2 OS cells expressing SNAP–Sec61β (grey) and *SiT-EGFP* mRNA. Puncta are

pseudocoloured and designated as translating (blue, MSD < 0.055 µm²) or non-translating (MSD ≥ 0.055 µm²). Scale bars, 5 µm. **e**, ER-specific SunTag reporter design. **f**, Representative merged images of SunTag (green), mRNA (magenta) and trajectories (yellow). Left, SunTag(+) (translating) punctum with confined motion. Right, SunTag(−) (non-translating) punctum showing rapid motion. Scale bars, 1 µm. **g**, Violin plots of log($MSD_{\tau=1s}$) for cytERM–SunTag MS2 with SunTag(+) versus SunTag(−). Dunnett's test, ****$P$ < 0.0001. **h**, Violin plots of log($MSD_{\tau=1s}$) for *SiT-EGFP*–MS2 (*n* = 1,616), *CD4-EGFP*–MS2 (*n* = 251), *CALR-mEmerald*–MS2 (*n* = 250), *ACTB-Halo*–MS2 (*n* = 100) and *SunTag-SEC61B–MS2* (*n* = 314). Red dashed line indicates MSD = 0.055 µm². Statistical comparisons versus *SiT-EGFP*–MS2, Dunnett's test, ***$P$ < 0.001, ****$P$ < 0.0001; NS, not significant ($P$ > 0.05).

the ER without interacting with the translocon[14,15]. Fluorescent puncta corresponding to individual *SiT-EGFP*–MS2 complexes were predominantly associated with the ER (Fig. 1a). Single-particle tracking revealed two distinct behaviours: some puncta remained largely immobile or confined on the ER for tens of seconds (Fig. 1a, region 1), whereas others moved rapidly, transiently associating with and dissociating from ER within a few seconds (Fig. 1a, region 2 and Supplementary Video 1).

Quantitative analysis of these trajectories using mean squared displacement (MSD) showed that about 86% of molecules displayed an MSD at lag time ($\tau$) = 1 s of less than 0.055 $\mu m^2$, consistent with confined motion (Fig. 1b, red dashed line, and Extended Data Fig. 1b,c; additional details in Supplementary Information). The population average was 0.011 $\mu m^2$. Translation termination with puromycin significantly increased mobility: now 79% of mRNAs had $MSD_{\tau=1s} > 0.055 \mu m^2$, with an average of 0.097 $\mu m^2$—an order of magnitude greater than controls (Fig. 1b,c and Extended Data Fig. 1b). Colour-coded video analysis further confirmed that slower puncta were always ER-bound, whereas faster puncta, particularly with puromycin treatment, were often dissociated from the ER (Fig. 1d and Supplementary Videos 2 and 3). These findings suggest that secretome mRNAs exhibit confined motion on ER surfaces during translation and become more motile when translation is disrupted.

To test whether confined ER-associated mRNAs are indeed engaged in active translation, we applied the SunTag system to monitor nascent peptides. We made a reporter encoding the targeting sequence and transmembrane domain of the ER protein cytochrome p450 fused with GCN4 repeats (*cytERM-SunTag*) and MS2 stem loops (Fig. 1e and Extended Data Fig. 1d). With its C-terminal SunTag epitopes exposed to the cytosol, this design enabled fluorescent labelling of translating mRNAs. Translation was inferred when *cytERM–SunTag*–MS2 puncta colocalized with SunTag signal from the translated peptide. In cells, around 40% of *cytERM–SunTag*–MS2 puncta showed colocalized MS2 (mRNA) and SunTag signals (Extended Data Fig. 1e,f). These translation-positive puncta exhibited nearly immobile or highly confined motion, whereas translation-negative puncta moved rapidly along the ER (Fig. 1f,g and Supplementary Videos 4 and 5). Upon puromycin treatment, SunTag signal was lost and all puncta became highly mobile (Supplementary Video 6). Thus, confined ER-associated secretome mRNAs represent actively translating species, whereas mobile ones are non-translating—a behaviour that contrasts with cytoplasmic protein mRNAs, which remain mobile even when translating[16].

Using confined motion as a proxy for translation, we compared different MS2-labelled secretome mRNAs. mRNAs encoding SiT–EGFP, CD4–EGFP and calreticulin–mEmerald showed substantial fractions of confined puncta (MSD < 0.055 $\mu m^2$), indicating active translation, whereas smaller pools displayed motile behaviour suggestive of non-translation (Fig. 1h). By contrast, non-secretome mRNAs encoding proteins such as β-actin, Sec61β or the mitochondrial protein TOMM20 moved predominantly in the mobile fraction ($MSD_{\tau=1s} > 0.055 \mu m^2$), with little evidence of confinement (Fig. 1h, Extended Data Fig. 1g,h and Supplementary Video 7).

## ER junctions as hotspots for translation

The ER is a complex membrane network composed of tubules that intersect to form diverse morphologies, including three-way junctions, dense networks of junctions and sheets[14]. To determine whether specific ER structures serve as subdomains for secretome mRNA translation, we imaged mRNAs in the peripheral ER, where morphologies could be resolved. Translating secretome mRNAs were defined as ER-associated mRNAs with confined or very low mobility ($MSD_{\tau=1s} < 0.055 \mu m^2$). Notably, translating secretome mRNAs were enriched at ER junctions. This was observed for mRNAs encoding CD4,

SiT and calreticulin (Fig. 2a). Quantification showed that translating secretome mRNAs were predominantly located within 0.2 $\mu m$ of junctions (Fig. 2b). By contrast, non-translating secretome mRNAs, identified by faster mobility on ER, showed no junctional enrichment and instead resembled the distribution of non-secretome mRNAs such as *ACTB* or *SEC61B* (Fig. 2b). This junctional localization of translating secretome mRNAs was consistent across U-2 OS, HeLa, COS-7 and HT1080 cells (Extended Data Fig. 2a,b).

We next explored whether endogenous secretome mRNAs also translate preferentially at ER junctions. Using *CD9* mRNA as a representative transcript[15], we performed hybridization chain reaction single-molecule fluorescent in situ hybridization (HCR-smFISH) in U-2 OS cells expressing mEmerald–Sec61β. Although HCR-smFISH does not distinguish translating mRNAs from non-translating mRNAs, localization relative to junctions could be assessed with and without translation inhibition. Under control conditions, *CD9* mRNAs were enriched at junctions; however, puromycin abolished this enrichment and increased the distance between *CD9* puncta and junctions (Fig. 2c,d). These results indicate that endogenous secretome mRNAs preferentially translate at ER junctions.

To further test this, we tracked ribosomes on the ER by imaging Halo-tagged L10A (a large ribosomal subunit protein) with an ER marker. If secretome mRNA translation occurs at junctions, translating ribosomes should show mobility similar to that of translating secretome mRNAs. Indeed, ribosome tracking revealed three diffusive populations—fast, medium and slow. Only the slow-moving population localized at ER junctions (Fig. 2e), with an $MSD_{\tau=1s}$ (0.016 $\mu m^2$) that closely matched that of translating secretome mRNAs (0.011 $\mu m^2$; Extended Data Fig. 2c–e). Puromycin shifted slow and medium populations into faster pools, consistent with their representing translating ribosomes. Together, these findings show that secretome mRNAs and associated ribosomes form slow-moving translating pools at ER junctions, demonstrating that junctions act as hotspots for secretome mRNA translation.

## Translation at LNPK-positive junctions

ER junctions are thought to be formed or stabilized by the ER-resident transmembrane protein LNPK[17,18], which is present at a subset of these junctions[17]. When we expressed LNPK–GFP in cells, it localized preferentially to ER junctions (Fig. 3a). To test whether LNPK-positive ER junctions are preferentially engaged in secretome mRNA translation, we expressed the ER translation reporter *cytERM-SunTag* and immunostained for endogenous LNPK (Extended Data Fig. 2f). Notably, translating *cytERM-SunTag*–MS2 puncta, defined by the presence of both mRNA and SunTag-bound nascent peptide, strongly colocalized with LNPK signal (Fig. 3b, SunTag(+)). By contrast, non-translating cytERM-SunTag–MS2 puncta exhibited significantly less colocalization with LNPK (Fig. 3b, SunTag(−)). Quantitative analysis showed that 80% of translating secretome mRNAs (SunTag(+)) were located within 300 nm of LNPK signal, compared with only 50% of non-translating mRNAs (SunTag(−)) within the same cell (Fig. 3c). These results demonstrate that translating secretome mRNAs are preferentially enriched at LNPK-containing ER junctions.

## LNPK loss reduces secretome translation

To test whether LNPK at ER junctions regulates secretome mRNA translation, we knocked down LNPK (LNPK-KD) in cells expressing *SiT-EGFP* and MS2 and performed single-particle tracking. *SiT-EGFP*–MS2 in LNPK-KD cells showed increased motility (Fig. 3d). Using the MSD cut-off of less than 0.055 $\mu m^2$ to identify translating secretome mRNAs, the translating fraction decreased from 86% in controls to 39% in LNPK-KD cells (Fig. 3e). Translation monitored via the SunTag signal from *cytERM-SunTag* mRNA similarly decreased in LNPK-KD

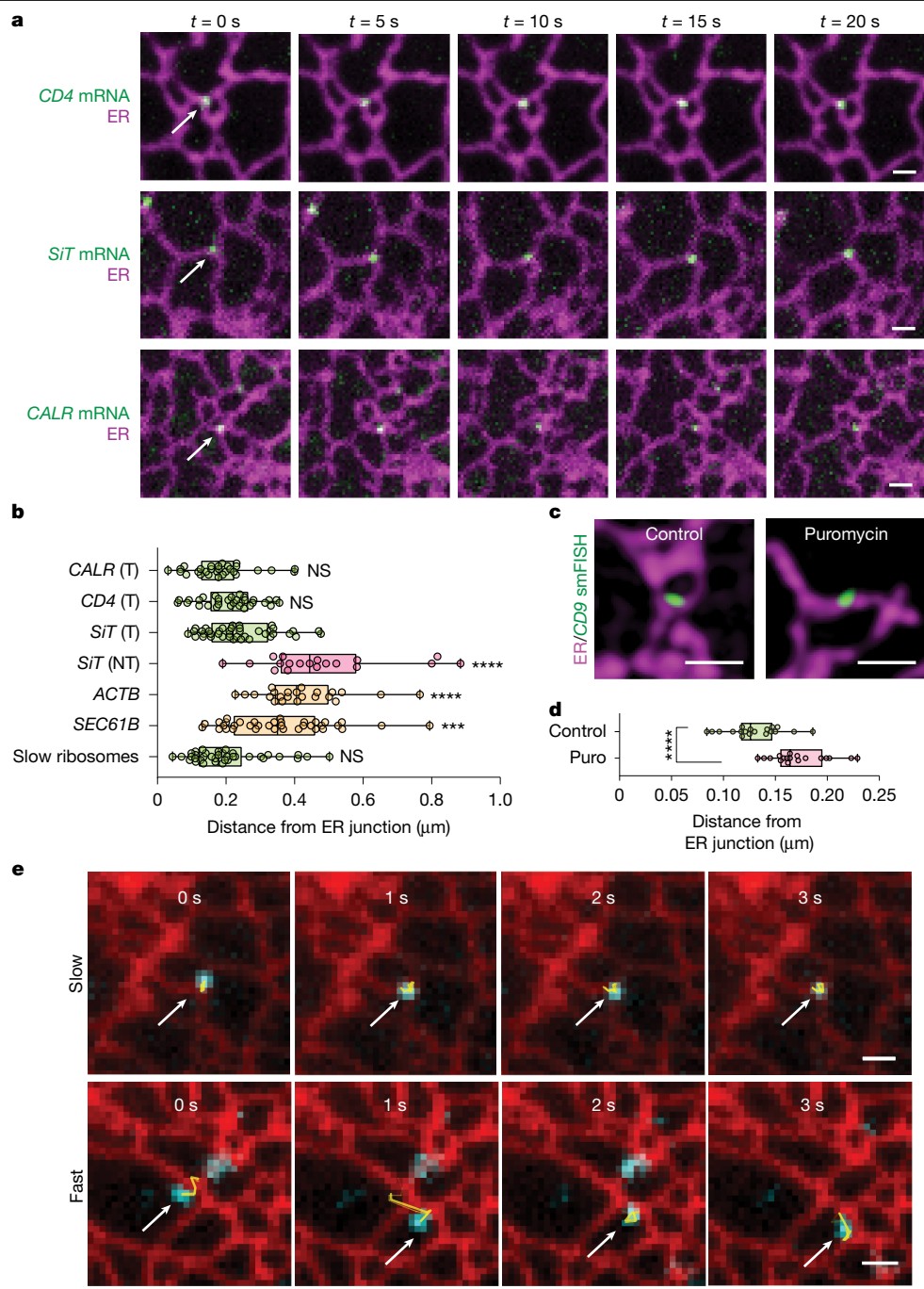

**Fig. 2 | ER junctions are hotspots for secretome mRNA translation.**
**a**, Time-lapse images of translating *CD4-EGFP*, *SiT-EGFP* and *CALR-mEmerald* mRNAs on ER (Sec61β, magenta) in U-2 OS cells. Scale bars, 1 μm. **b**, Box plots of mean distance to nearest ER junction per cell for translating (T) fractions of *CALR* (n = 29 cells), *CD4* (n = 32) and *SiT* (n = 40) mRNA, non-translating (NT) *SiT* mRNA (n = 19), *Halo–ACTB* (n = 23), *Halo–SEC61B* (n = 36) and slow ribosomes (effective $MSD_{\tau=1s} < 0.04\ \mu m^2$, n = 37). Comparisons versus translating *SiT-EGFP*. Dunnett's test, ****$P < 0.0001$, ***$P < 0.001$. **c**, Representative 3D structured illumination microscopy images of HCR-smFISH labelling of endogenous *CD9* mRNA with the ER marker (mEmerald–Sec61β, magenta) in control and puromycin-treated U-2 OS cells. Scale bars, 1 μm. **d**, Quantification of distance of endogenous *CD9* smFISH puncta to the nearest ER junction in control (n = 19 cells) and puromycin-treated (n = 18) cells. Each dot represents the mean per cell. Two-tailed unpaired *t*-test, ****$P < 0.0001$. **e**, Time-lapse images of a ribosome (cyan) with effective $MSD_{\tau=1s} < 0.04\ \mu m^2$ (slow, top) or $MSD_{\tau=1s} > 0.04\ \mu m^2$ (fast, bottom). Scale bars, 1 μm.

cells (Extended Data Figs. 1e and 3g,h). By contrast, knockdown of the ER-shaping protein CLIMP63[5] had no effect, with more than 80% of *SiT-EGFP* mRNAs still undergoing translation (Fig. 3d,e and Extended Data Fig. 1e).

We next examined steady-state protein levels in LNPK-knockout (LNPK-KO) cells. Whereas total protein levels were unchanged, membrane protein levels were strongly reduced (Fig. 3f). Incorporation of the amino acid analogue homopropargylglycine (HPG) confirmed that global synthesis was unaffected, but membrane-associated HPG-labelled proteins were significantly decreased after extraction of cytosolic components (Extended Data Fig. 3a,b). Mass spectrometry of membrane fractions combined with mRNA sequencing further showed reduced protein abundance per mRNA for many membrane-associated genes in LNPK-KO cells (Extended Data Fig. 3c–f). These results

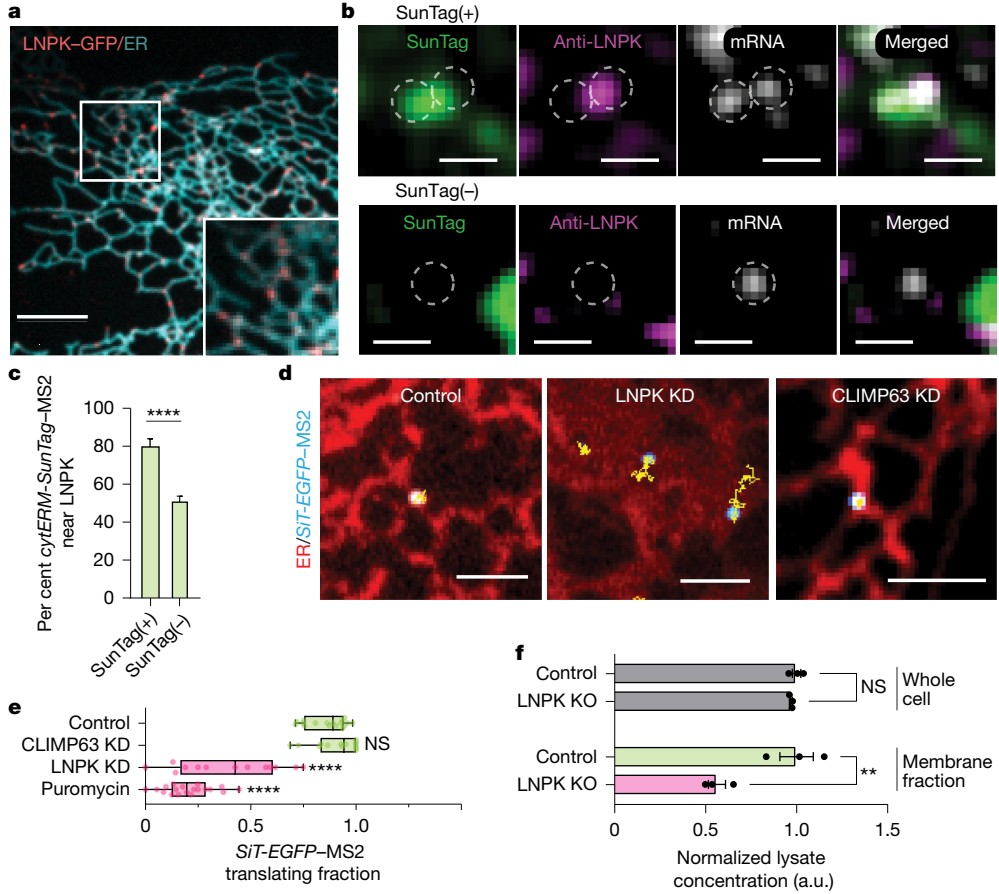

**Fig. 3 | LNPK marks ER junctions that enhance secretome mRNA translation.**
**a**, Representative image of LNPK–GFP and ER (Halo–Sec61β, cyan). Inset, enlarged view. Scale bar, 5 μm. **b**, Top, translating SunTag(+) *cytERM-SunTag*–MS2 (grey) with overlapping SunTag (green) and LNPK (magenta). Bottom, non-translating SunTag(−) *cytERM–SunTag*–MS2, showing no colocalization with LNPK. Scale bars, 1 μm. **c**, Percentage of *cytERM–SunTag*–MS2 puncta that are SunTag(+) or SunTag(−) within 300 nm of LNPK (SunTag(+)) or outside this range (SunTag(−)). $n = 14$ cells. Paired two-tailed *t*-test, ****$P < 0.0001$. **d**, Representative images of *SiT-EGFP*–MS2 and particle trajectories (yellow) on ER (red) in control, LNPK-KD or CLIMP63-KD cells. Scale bars, 1 μm. **e**, Translating fractions of *SiT-EGFP* mRNA in control ($n = 17$ cells), CLIMP63-KD ($n = 14$), LNPK-KD ($n = 13$) and puromycin-treated ($n = 17$) cells. Statistical comparisons were performed against control using Dunnett's test, ****$P < 0.0001$. **f**, Protein yield from whole-cell extracts (top) and membrane fractions (bottom) of $10^6$ control and LNPK-KO U-2 OS cells. Each extract was normalized to its control. a.u., arbitrary units. Dunnett's test, **$P < 0.005$.

demonstrate that LNPK is essential for maintaining membrane and secretory protein synthesis.

## LNPK-rich junctions recruit lysosomes

Previous studies have suggested that lysosomes can associate with axonal sites of translation[19] and with ER junctions[20,21], potentially via LNPK. To directly test for such associations, we performed proximity ligation assays (PLAs) using anti-LNPK in combination with markers of early endosomes (EEA1), lysosomes (LAMP1) or mitochondria (TOMM20) (Fig. 4a). PLA signal was significantly greater with LAMP1 compared to EEA1 or TOMM20, indicating that LNPK-enriched ER regions preferentially associate with lysosomes (Fig. 4b,c). As a specificity control, PLA was performed using anti-LAMP1 and anti-REEP5, an ER protein of similar abundance to LNPK[22] but with no known lysosomal interactions (Fig. 4c). In this case, only background levels of PLA signal were observed, supporting the conclusion that lysosome association is specific to LNPK-positive ER domains.

To assess whether LNPK contributes functionally to ER–lysosome tethering, we used an optogenetic recruitment assay with light-inducible iLID–sspB dimerization (Fig. 4d). Cells co-expressing LAMP1–iLID and ER-targeted sspB–Sec61β exhibited rapid lysosome recruitment to the ER following blue-light stimulation ($t_{1/2}$ (the time required for the recruitment signal to reach half of its saturation value) = 7.9 s),

culminating in the ER wrapping tightly around lysosomes (Fig. 4e,f). By contrast, LNPK-KO cells displayed a markedly slower recruitment ($t_{1/2} = 26$ s) under identical conditions (Fig. 4e,f). Together, these findings indicate that LNPK promotes the efficient formation and stabilization of ER–lysosome contacts.

## Lysosomes are at secretome translation sites

Given that LNPK marks sites of secretome mRNA translation and is positioned near lysosomes, we sought to determine whether lysosomes can associate with active translation sites. Using *cytERM-SunTag*–MS2 reporters alongside fluorescently labelled LAMP1, we observed that translating secretome mRNA puncta often reside adjacent to LAMP1-positive lysosomes (Fig. 4g). Remarkably, the intensity of the SunTag signal was increased when *cytERM-SunTag*–MS2 puncta were in close proximity to lysosomes (Fig. 4h, top), suggesting that secretome mRNAs near lysosomes engage a greater number of ribosomes (Fig. 4h, bottom). Supporting this, *SiT-EGFP*–MS2 exhibiting confined motion, indicative of active translation, was found significantly closer to lysosomes than their highly mobile, non-translating counterparts (Extended Data Fig. 3g). Collectively, these data reveal that enhanced secretome mRNA translation is spatially coordinated at ER subdomains adjacent to lysosomes.

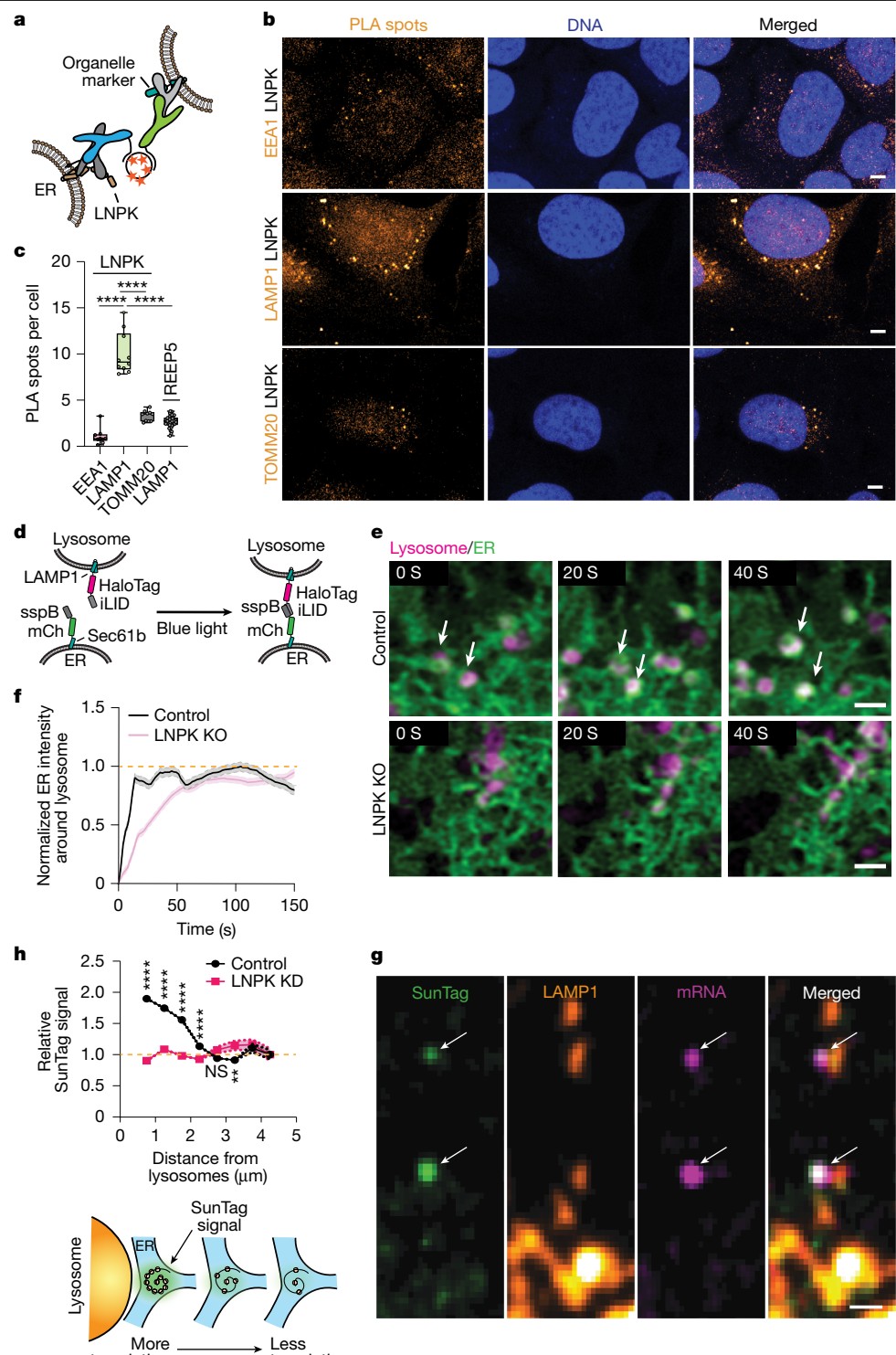

**Fig. 4 | LNPK-enriched ER junctions recruit lysosomes and boost translation.** **a**, Schematic of PLA. Anti-LNPK antibody (via oligo-labelled anti-rabbit, blue) and organelle marker antibodies (via oligo-labelled anti-mouse, green) generated proximity signals amplified by rolling circle amplification and visualized with a fluorescent probe (orange). **b**, Representative images from PLA experiment, showing DNA, PLA puncta (orange) and merged views of LNPK antibody with EEA1 (early endosome), LAMP1 (lysosome) or TOMM20 (mitochondria). Scale bars, 5 μm. **c**, Quantification of PLA spots per cell for LNPK with EEA1, LAMP1 and TOMM20, and for LAMP1 with REEP5 (control). Data from 10 fields of view, 3 replicates. Statistical comparisons versus LNPK–LAMP1. Dunnett's test, ****$P < 0.0001$. **d**, Schematic of the optogenetic ER–lysosome recruitment tool comprising iLID–Halo–LAMP1 (lysosome) and sspB–mCherry–Sec61β (ER). mCh, mCherry. **e**, Representative images of iLID–Halo–LAMP1 (magenta) and sspB–mCherry–Sec61β (green) during 488 nm activation in control and LNPK-KO cells. Scale bars, 1 μm. **f**, Quantification of ER signal within lysosomal mask over time following activation. Control cells: $t_{1/2} = 7.9$ s; LNPK-KO cells: $t_{1/2} = 26$ s. Shaded areas represent 95% confidence intervals. Orange dashed line indicates a normalized intensity of 1. **g**, Representative images of SunTag, LAMP1, *cytERM-SunTag*–MS2 (mRNA) and merged views. Arrows mark translating mRNAs. Scale bar, 1 μm. **h**, Top, relative SunTag intensity versus distance to lysosomes (binned at 500 nm) for *cytERM–SunTag*–MS2 in control and LNPK-KD cells. Normalized to intensity at 4.25 μm. Shaded areas represent 95% confidence intervals. Orange dashed line indicates a relative SunTag intensity of 1. Multiple unpaired *t*-tests were done, ****$P < 0.0001$, **$P < 0.01$. Bottom, schematic showing how lysosome proximity increases SunTag intensity, reflecting higher ribosome occupancy.

Building on this, we investigated the role of LNPK in facilitating this spatially restricted translation boost. Notably, in LNPK-knockdown (LNPK-KD) cells, the increased SunTag signal normally observed near lysosomes was abolished (Fig. 4h). This contrasts with control cells, where lysosome-adjacent translation sites demonstrated higher ribosome engagement. These findings indicate that LNPK is essential for establishing the specialized ER–lysosome interface that promotes elevated secretome mRNA translation.

## Amino acid starvation links translation to lysosomes

Lysosomes are acidic compartments that degrade engulfed proteins into amino acids[23], which are then released into the cytoplasm to support cellular functions. We hypothesized that lysosomes might enhance secretome mRNA translation by supplying a local source of amino acids for protein synthesis. Consistent with this idea, disrupting lysosomal protein degradation–either by raising lysosomal pH with chloroquine or by inhibiting lysosomal proteases–significantly diminished *SiT-EGFP* mRNA translation (Fig. 5a).

Lysosomal degradation is especially critical during amino acid starvation, when cells rely on lysosome-derived amino acids to sustain protein synthesis[24,25]. To test whether secretome mRNA translation becomes more dependent on lysosome proximity under these conditions, we used the *cytERM-SunTag*–MS2 system to examine the spatial relationship between translating mRNAs and lysosomes upon amino acid deprivation. Notably, the association between secretome mRNA translation and lysosome proximity was markedly enhanced during amino acid starvation, with SunTag signal intensities from *cytERM-SunTag*–MS2 puncta near lysosomes significantly higher compared to control conditions (Fig. 5b). This occurred despite an overall decline in *SiT-EGFP* mRNA translation under amino acid starvation (Fig. 5a, Control versus −AA). Together, these findings reveal that when extracellular amino acids are scarce, the efficiency of secretome mRNA translation becomes increasingly reliant on close proximity to lysosomes, highlighting a specialized spatial coordination between nutrient availability and protein synthesis.

## Lysosomes enhance translation initiation

To uncover the regulatory mechanisms driving enhanced secretome mRNA translation near lysosomes, we focused on translation initiation, the rate-limiting step in mammalian protein synthesis[26]. We engineered the 5′ untranslated region (UTR) of the *cytERM-SunTag*–MS2 reporter to contain the cricket paralysis virus internal ribosome entry site (CrPV IRES), which bypasses nearly all canonical translation initiation controls[27,28]. Expression of this CrPV IRES-containing reporter abolished the lysosome-proximal boost in SunTag signal observed with the unmodified *cytERM-SunTag*–MS2 (Fig. 5b), indicating that the lysosome-associated translation enhancement depends on regulated translation initiation.

Next, we compared translation levels of both reporters in wild-type and LNPK-KD cells (Fig. 5c). Translation of the unmodified *cytERM-SunTag*–MS2 reporter was significantly reduced upon LNPK depletion, whereas the CrPV IRES-containing reporter showed no such impairment. This demonstrates that disruption of LNPK impairs secretome mRNA translation by interfering with translation initiation, and that bypassing initiation control can rescue this defect. Together, these results establish that proximity to lysosomes enhances secretome mRNA translation through an LNPK-dependent pathway that acts at the level of translation initiation.

## ISR-like signalling in LNPK-KD cells

Translation initiation can be modulated by the ISR pathway[26], which inhibits protein synthesis via eIF2α phosphorylation. To determine whether ISR signalling contributes to the reduced secretome mRNA translation observed in LNPK-KD cells, we tested the effect of ISRIB, an inhibitor that reverses the translational block imposed by eIF2α phosphorylation[26]. Remarkably, ISRIB treatment restored translation of *cytERM-SunTag* mRNA in LNPK KD cells to control levels (Fig. 5c). This recovery mirrored the effects of ISRIB in cells exposed to thapsigargin, which induces global translation repression through ISR activation, or amino acid starvation, which activates both ISR and inhibits mTOR signalling; in both scenarios, ISRIB reversed the translation inhibition of *SiT-EGFP* mRNA (Fig. 5d). These findings indicate that ISR activation lies upstream of the translation defects caused by LNPK depletion.

By contrast, inhibition of mTOR signalling using rapamycin or torin-1, which does not trigger ISR, had no effect on translation of *SiT-EGFP* (Fig. 5d). This highlights that, unlike ISR, mTOR signalling does not significantly regulate secretome mRNA translation, consistent with previous work showing that mTOR primarily controls translation of mRNAs with 5′ terminal oligopyrimidine (TOP) motifs[29,30], which are largely absent in membrane and secretory mRNAs.

To further probe the connection to ISR, we assessed eIF2α phosphorylation status in LNPK-KO cells. We found an increased ratio of phosphorylated eIF2α to total eIF2α in LNPK-KO cells relative to controls, driven by sustained phosphorylated eIF2α levels alongside decreased total eIF2α (Fig. 5e–g and Extended Data Fig. 4a–d). This supports the notion that eIF2α-dependent translation initiation is impaired in LNPK-deficient cells, linking secretome mRNA translation defects to ISR signalling.

Of note, this LNPK-dependent suppression of eIF2α-mediated translation appears to be distinct from classical ISR activation. Unlike canonical ISR, LNPK depletion selectively diminished synthesis of membrane and secretory proteins without affecting cytosolic proteins (Fig. 3f) and did not induce a robust increase in ATF4, a hallmark ISR effector[31] (Extended Data Fig. 4c,d). Furthermore, LNPK loss did not trigger the unfolded protein response, as indicated by lack of XBP-1 splicing (Extended Data Fig. 4e). Together, these findings suggest that a non-canonical ISR-like pathway modulates secretome mRNA translation downstream of LNPK.

## ISRIB restores translation after LNPK KD

To directly monitor nascent peptide synthesis of secretome mRNAs in live cells, we performed fluorescence recovery after photobleaching (FRAP) experiments targeting SunTag puncta. In control cells, photobleaching of a single SunTag punctum resulted in full fluorescence recovery to pre-bleach levels within 300 s. This recovery reflected new peptide synthesis rather than antibody exchange, as cycloheximide-treated cells showed minimal fluorescence recovery (Extended Data Fig. 4f). By contrast, LNPK-KD cells exhibited markedly impaired recovery kinetics, with fluorescence failing to return to pre-bleach levels within the same timeframe, indicating that LNPK depletion hinders nascent peptide synthesis from secretome mRNAs.

Notably, treatment of LNPK-KD cells with ISRIB restored the SunTag recovery kinetics to levels resembling control cells (Fig. 5h,i), whereas ISRIB had no effect on recovery kinetics in control cells (Fig. 5i and Extended Data Fig. 4f). These findings demonstrate that ISRIB specifically rescues the translation defects induced by LNPK loss. Given that ISRIB reverses eIF2α-dependent translational repression, the results imply that LNPK regulates nascent peptide synthesis of secretome mRNAs through an eIF2-dependent mechanism.

## Discussion

The spatial organization of mRNA translation within cells has been a longstanding question in cell biology, particularly with respect to

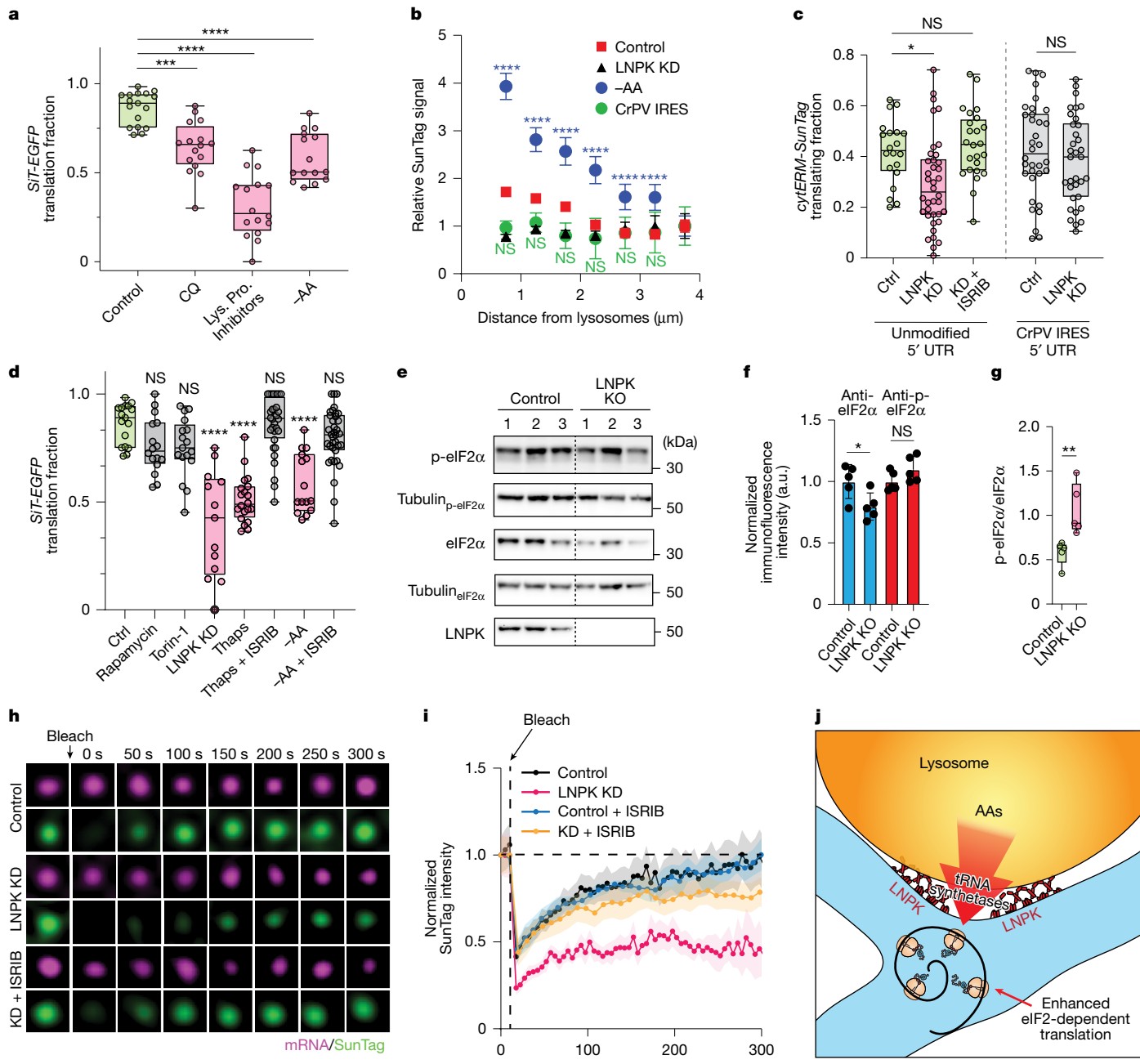

**Fig. 5 | Lysosome activity and LNPK regulate secretome translation via eIF2.**
**a**, Translating fraction of *SiT-EGFP* mRNA under control conditions, with
chloroquine (CQ; 10 μM, 4 h; *n* = 16 cells), lysosomal protease inhibitor (Lys. Pro.
Inhibitors; 1 h; *n* = 16) or amino acid starvation (−AA; 16 h; *n* = 15). Comparisons
versus control; Dunnett's test, ***P = 0.0003, ****P < 0.0001. **b**, Relative SunTag
intensity versus distance from lysosome (500 nm bins) for *cytERM−SunTag−*
MS2 under control conditions, with LNPK knockdown, amino acid depletion
or CrPV IRES expression. Shaded areas represent 95% confidence intervals.
Comparisons versus LNPK KD; multiple unpaired *t*-tests, ****P < 0.0001. **c**, Left,
translating fractions of *cytERM−SunTag* in control (Ctrl; *n* = 20 cells), LNPK-KD
cells (*n* = 38) and LNPK-KD cells with 200 nM ISRIB (*n* = 24). Comparisons versus
control; Dunnett's test, *P = 0.0081 (LNPK KD), P = 0.7951 (LNPK KD plus ISRIB).
Right, translation of *cytERM−SunTag* with CrPV IRES 5′ UTR in control (*n* = 34)
and LNPK-KD (*n* = 34) cells. Unpaired two-tailed *t*-test, P = 0.3777. **d**, Translating
fraction of *SiT-EGFP* mRNA under control conditions, with rapamycin (100 nM,
1 h, *n* = 16), torin-1 (1 μM, 1 h, *n* = 17), LNPK KD (*n* = 13), 1 μM thapsigargin (Thaps; 1 h,
*n* = 20), 1 μM thapsigargin plus 200 nM ISRIB (1 h, *n* = 32), amino acid starvation

(16 h, *n* = 15), or amino acid starvation plus 200 nM ISRIB (1 h, *n* = 36).
Comparisons versus control. Dunnett's test, ****P < 0.0001. **e**, Western blots
of total eIF2α, phosphorylated eIF2α (p-eIF2α), LNPK and tubulin (as loading
control) from control and LNPK-KO cells (3 replicates). **f**, Quantification of total
eIF2α (blue) and p-eIF2α immunofluorescence in control and LNPK-KO cells.
Unpaired two-tailed *t*-test versus control: eIF2α, *P = 0.0346; p-eIF2α, P = 0.14.
**g**, Ratio of p-eIF2α/eIF2α from immunofluorescence (5 replicates). Unpaired
two-tailed *t*-test, **P = 0.0095. **h**, Time-lapse images of FRAP recovery of SunTag
(green) at *cytERM−SunTag* MS2 (magenta) in control cells, LNPK-KD cells and
LNPK-KD cells treated with ISRIB. Arrows indicate photobleaching sites. **i**, FRAP
recovery curves for control (*n* = 31), LNPK-KD (*n* = 31), control plus ISRIB (*n* = 10)
and LNPK-KD plus ISRIB (*n* = 32) conditions. The vertical dashed line indicates
time of photobleaching and the horizontal dashed line indicates a normalized
intensity of 1. Shaded areas represent 95% confidence intervals. **j**, Model showing
how LNPK and lysosomes form ER junctional hubs to enhance secretome
translation via eIF2-dependent initiation. AAs, amino acids.

secretome mRNAs encoding secretory and membrane proteins. Our study provides evidence that ER junctions enriched in LNPK and associated with lysosomes represent critical organizational hubs for this process, thereby offering a new perspective on how the location and regulation of translation are coordinated.

A key finding is that secretome mRNAs display dynamic localization behaviour depending on their translational state. Prior to engaging the translocation machinery, they move freely along the ER network, but upon initiation of co-translational translocation, they become stabilized at ER junctions. These sites, characterized by their reduced curvature and relatively larger surface area compared with surrounding tubules[32], could provide a favourable architecture for polysome assembly. This observation challenges the traditional view that ER sheets, particularly those located around the nucleus, are the exclusive sites of membrane protein translation. Instead, peripheral ER junctions emerge as important, spatially distributed hotspots for secretome synthesis, suggesting a more decentralized and flexible organization of the translational landscape than previously appreciated.

Our data further identify LNPK as an essential determinant of this process. LNPK-stabilized junctions support secretome mRNA translation primarily through regulation of initiation. Loss of LNPK reduces translation, an effect that correlates with elevated phosphorylation of eIF2α and can be reversed by ISRIB treatment. Notably, LNPK depletion selectively impairs secretome mRNA translation without inducing other canonical markers of the integrated stress response, pointing to a LNPK-specific mechanism that is related to eIF2-dependent translation regulation but distinct from classical ISR signalling. The resistance of CrPV IRES-driven reporters to LNPK loss further underscores the central role of eIF2-dependent initiation in LNPK-mediated translation regulation.

Perhaps most intriguing is the role of lysosomes in supporting translation at ER junctions. We demonstrate that lysosome proximity correlates with increased ribosome density on secretome mRNAs and that disruption of lysosomal function selectively impairs this translation. The data support a model in which lysosomes contribute localized amino acid pools and may supply tRNA-charging activities that directly fuel co-translational translocation (Fig. 5j). This coupling becomes particularly relevant under nutrient stress, when cells must preserve the synthesis of secretory and membrane proteins that are critical for survival and signalling. The capacity of lysosome-associated junctions to sustain translation during amino acid starvation highlights inter-organelle communication as a determinant of translational resilience.

The broader biological relevance of this mechanism likely extends across multiple cell types. In neurons, where LNPK dysfunction causes developmental defects, the transport of lysosome-associated RNA granules into axons and dendrites[33,34] raises the possibility that LNPK-enriched ER junctions may couple with the lysosome-associated RNA granules to coordinate local secretory protein synthesis and spatial translation control. Such coupling between ER architecture and RNA transport assemblies offers a potential mechanistic link between membrane organization, spatial regulation of translation and neuronal function. Similarly, LNPK upregulation during B cell differentiation into antibody-secreting plasma cells[35] suggests a conserved role for this protein in supporting heightened secretory demand. The persistence of some secretory protein synthesis in LNPK-KO cells indicates parallel or compensatory pathways, raising important questions about the interplay between LNPK-dependent and LNPK-independent regulatory mechanisms.

In conclusion, these findings position LNPK and lysosome-associated ER junctions as central nodes in the spatial regulation of secretome mRNA translation. By integrating translation initiation control with nutrient sensing and local amino acid availability, this system establishes a flexible and dynamic framework for tuning protein synthesis to both cellular demands and environmental conditions. Future studies will be critical for delineating how general this mechanism is across diverse cell types and physiological contexts, and for exploring how disruptions in this system contribute to disease states, particularly in the nervous and immune systems.

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

# Methods

## Cell culture and electroporation

U-2 OS (ATCC) cells were cultured in phenol-red free DMEM supplemented with 10% fetal bovine serum (Corning), 2 mM L-glutamine (Corning), and 100 IU penicillin and 100 µg ml$^{-1}$ streptomycin (Cell-Gro) at 37 °C and 5% $CO_2$. For the tet-inducible system, FBS that was tested free of tetracycline (Cytvia) was used. Cells were grown on either diluted fibronectin (1:100, Millipore) or Matrigel (1:100, Corning)-coated coverslip chambers or sterilized cover glasses (no. 1.5 Round (EMS)). Lonza Electroporation kit with the U-2 OS specific set-up suggested by the company was used for transient transfection, which was performed 16–20 h prior to imaging. In addition, HeLa (ATCC), COS-7 (ATCC), HEK293T (ATCC) and HT1080 (ATCC) cells were cultured and electroporated according to the manufacturer's instructions. Cells were routinely tested for mycoplasma contamination.

## Plasmids and cloning

A list of plasmids used in this study is provided in Supplementary Table 1.

## Generation of stable cell line and doxycycline induction

To generate lentivirus particles, HEK293T cells were transfected with MCP-GFP, MCP-HaloTag or scFv-sfGFP plasmids, along with viral packaging plasmids. The supernatant was collected 48–72 h after transfection, filtered through a 0.22-µm syringe filter, and concentrated using Lenti-X Concentrator (Takara). U-2 OS cells were exposed to lentiviral particles, and MCP-HaloTag and scFv-sfGFP double positive U-2 OS cells were generated via sequential viral transduction and sorted via fluorescence-activated cell sorting.

To generate cells stably expressing the protein of interest with only MS2 binding sites mRNAs, lentiviral transduction was used on pre-sorted MCP-GFP or MCP-HaloTag positive U-2 OS cells. Cells stably expressing a tet-inducible cytERM–SunTag were generated using the PiggyBac system. MCP-HaloTag/scFv-sfGFP double positive U-2 OS cells were then electroporated with cytERM-SunTag plasmid with hyperactive piggyBac transposase (hyPBase). After 4 days post-electroporation, the cells were selected under 10 µg ml$^{-1}$ Blasticidin S for two weeks. The stable cells expressing MCP-HaloTag/scFv-sfGFP/cytERM–SunTag were induced with 100 ng ml$^{-1}$ doxycycline 3–5 h prior to imaging.

## Labelling JF dyes with HaloTag ligand and SNAPTag ligand

For cells expressing HaloTag, they were incubated in complete media with 100 nM JF646 HaloTag ligand (JF646-HTL) at 37 °C for 30 min and then washed twice. The rinsed cells were then equilibrated in the media at 37 °C for 30 min prior to imaging. Similar methods were used for PA-JF646-HTL, PA-JF549-HTL, JF549-HTL, JF646-HTL and JF635-HTL.

Cells expressing SNAPTag were incubated in complete media with 250 nM JF549 with chloropyrimidine SNAPTag ligand (JF549-cpSNAP) for 1 h at 37 °C and then washed twice. The rinsed cells were then washed twice again after 30 min to remove residual dyes, and then equilibrated in fresh media at 37 °C for 30 min prior to imaging.

## Reagents for condition screening

Rapamycin (Sigma, 2.5 mg ml$^{-1}$ in DMSO), torin-1 (Cell Signaling, 1 mM in DMSO), cycloheximide (Sigma, 100 mg ml$^{-1}$ in DMSO), chloroquine (Sigma, 10 mM in $H_2O$), thapsigargin (Sigma, 1 mM in DMSO), protease cocktail inhibitor (ThermoFisher) and puromycin (ThermoFisher, 10 mg ml$^{-1}$ in $H_2O$). Amino acid-free medium is composed of amino acid-free DMEM (USBiological), 10% dialysed FBS (ThermoFisher), 4.5 g l$^{-1}$ glucose (Sigma), 110 mg l$^{-1}$ sodium pyruvate (Sigma), 3.7 g l$^{-1}$ sodium bicarbonate (Sigma), 2 mM L-glutamine (Corning), 100 IU penicillin and 100 µg ml$^{-1}$ streptomycin (CellGro).

## Single-molecule tracking with HILO microscopy

Single molecule tracking of single ribosomes was conducted using a customized Nikon TiE inverted microscope with a split port for simultaneous imaging onto two Andor iXON DU-897 EM-CCD cameras, equipped with a 100× Apo TIRF 1.49 oil-immersion objective (Nikon). U-2 OS cells were co-transfected with L10A-HaloTag plasmid and either mEmerald Sec61b or SNAPf-Sec61b plasmids and imaged after 16 h. These cells were labelled with either PA-JF549-HTL or PA-JF646-HTL/JF549-cpSNAP[36]. A brief pulse of 405 nm laser was used to excite and activate the PA-JF dyes. The incident angle for excitation lasers was manipulated to achieve highly inclined and laminated optical sheet (HILO) illumination. The emission resulting from simultaneous excitation of two fluorophores was split using either T647LPXR (Chroma) or T565LPXR (Chroma) and collected in two Andor iXon EM-CCDs with emission filters: ET525/50 m (Chroma) for GFP, ET605/70 m (Chroma) for PA-JF549 or JF549, and ET700/75 m (Chroma) for PA-JF646.

## Single-particle tracking with spinning-disk confocal microscopy

Single molecule tracking for single mRNAs was conducted using a customized Nikon TiE inverted microscope, outfitted with a Yokogawa spinning-disk scan head (CSU-X1, Yokogawa) and Andor iXON EM-CCD cameras. Fluorescence was collected using a 100× Apo TIRF 1.49 oil-immersion objective (Nikon). For live-cell imaging, cells were imaged in complete medium and incubated with a stage heater (Tokai Hit) at 37 °C and 5% $CO_2$. Two-colour imaging was performed similarly to single-molecule tracking, as described above. Three-colour imaging was performed using trigger mode. The emission from 561 and 640 nm excitation was collected sequentially into the same camera using a multi-bandpass emission filter (Chroma, ZET405/488/561/647 m).

## Immunofluorescence and PLA

For immunofluorescence labelling, U-2 OS cells stably expressing MCP–HaloTag, scFv–sfGFP or cytERM–SunTag were plated on Matrigel-coated 12 mm coverslips (EMS) in sterile 24-well tissue-culture graded plates (Corning). The cells were induced with 100 ng ml$^{-1}$ doxycycline (Sigma) and labelled with JF dyes as described above. The cells were then fixed with 4% (w/v) paraformaldehyde (PFA) + 0.2% glutaraldehyde for 15 min at room temperature. Cells were permeabilized with 0.1% Triton X-100 for 10 min, then treated with 1% BSA (Sigma) in phosphate-buffered saline (PBS) plus 0.05% Tween-20 (PBS-T) for 1 h. The primary antibody (Rabbit Anti-LNPK antibody, Sigma) was added overnight at 4 °C and washed with 1% BSA + PBS-T. The coverslip was incubated with secondary antibodies conjugated with Alexa Fluor 594 (ThermoFisher) for 1 h at room temperature and rinsed with PBS. The coverslip was mounted on a slide and imaged using a Zeiss 980 AiryScan equipped with a 63× objective.

PLA was performed using the Duolink In situ Orange Starter kit Mouse/Rabbit (Sigma). U-2 OS cells were plated on Matrigel-coated 12 mm coverslips (EMS) in sterile 24-well tissue-culture graded plates (Corning). The manufacturer's protocol was followed, using the following antibodies: rabbit LNPK antibody (Sigma, HPA014205-25), mouse EEA1 antibody (BD Biosciences, 610456), mouse LAMP1 antibody (Abcam, ab25630), mouse TOMM20 antibody (66777-1-Ig) and rabbit REEP5 antibody (ProteinTech, 14643-1-AP).

## Immunoblotting

For immunoblotting, U-2 OS cells were collected by scraping the plate in lysis buffer (25 mM Tris-HCl pH 7.6, 150 mM NaCl, 1% NP-40, 1% sodium deoxycholate, 0.1% SDS) supplemented with EDTA-free protease inhibitor cocktail (Roche) and PhoSTOP (Roche), followed by sonication. The lysate was quantified with a BCA kit (ThermoFisher), following the manufacturer's protocol and diluted using 4× NuPage LDS Sample buffer (ThermoFisher) supplemented with 0.1 M DTT

(Sigma Aldrich). The samples were heated in a 95 °C heat block for 10 min. Equal amounts of lysate (~5 μg) were run on 4–12% NuPage Bis-Tris Mini protein gels (ThermoFisher). After electrophoresis, the proteins were transferred onto 0.2 μm Nitrocellulose membranes using the Trans-Blot Turbo system. The membranes were blocked using Every-Blot blocking buffer (Bio-Rad). Primary antibodies were diluted 1:1,000 in the same blocking buffer and incubated overnight at 4 °C with gentle agitation. After washing 3 times with TBS-T, secondary antibody staining was performed for 1 h at room temperature. The membranes were washed with TBS-T and developed with SuperSignal West Pico PLUS Chemiluminescent Substrate (ThermoFisher). The membranes were imaged on ChemiDoc (Bio-Rad) with optimized exposure for signal detection. Primary antibodies used here are: rabbit ATF-4 antibody (Cell Signaling, 11815S), rabbit LNPK antibody (Sigma, HPA014205-25), mouse Tubulin antibody (Millipore, 05-829), rabbit eIF2S1 (phosphor S51) antibody (Abcam, ab32157), and rabbit eIF2S1 antibody (Atlas Antibodies, HPA064885). Secondary antibodies used here are: goat anti-mouse Ig H&L-HRP (Abcam, ab205719) and goat anti-rabbit Ig H&L-HRP (Abcam, ab205718).

## HCR-smFISH

For HCR-smFISH, we designed and purchased oligonucleotides that hybridize to endogenous human *CD9* mRNA from Molecular Instruments. U-2 OS cells transiently transfected with mEmerald–Sec61β were plated on Matrigel-coated 12 mm coverslips in sterile 24-well tissue-culture graded plates. Cells were fixed with 4% (w/v) PFA + 0.2% glutaraldehyde for 15 min at room temperature. Hybridization was performed based on the manufacturer's protocol for mammalian cells on a coverslip for single-molecule detection. The coverslips were mounted on a glass slide and imaged immediately.

## Single-particle tracking analysis

L10A-HaloTag and mRNA tracking was performed using TrackMate software (FIJI). The resulting trajectories were imported into Matlab using a custom Matlab code to store $x$, $y$ and $t$. Displacement of each trajectory was calculated, and each trajectory was categorized based on the mask to classify cytosolic trajectories and remove nuclear localizations using custom Matlab code.

The MSD of the population was calculated by averaging squared displacement (SD) at a given lag time. The apparent diffusion coefficient ($D_{app}$) was calculated based on a linear fit of MSD versus lag time using the built-in linear fit module in Matlab, described by equation (1).

$$MSD(\tau) = 2nD_{app}\tau + 4\sigma^2 \qquad (1)$$

where $D_{app}$ is apparent diffusion coefficient, $\tau$ is lag time, $n$ is number of dimensions (2) and $\sigma$ is localization error. The first four points were used to fit the curve to estimate the apparent diffusion coefficient as many of these trajectories showed a confined diffusion.

Mean square displacement from each trajectory ($MSD_\tau$) was calculated by averaging the SD within the same trajectories. For determining the translationally active population, we analysed trajectories that are longer than 9 steps at 1 Hz and averaged 9 individual SDs to generate $MSD_\tau$, as described in equation (2).

$$MSD_\tau = \frac{\sum_{i=1}^{n}((x_{i+1}-x_i)^2 + (y_{i+1}-y_i)^2)}{n} \qquad (2)$$

## Monte Carlo simulation of Brownian motion

Monte Carlo simulation of random walk was performed using a custom-built Matlab code. For each time step, individual particles choose displacement in each of the three Cartesian directions, and its distribution is defined by the free, 3D diffusion coefficient $D$. The squared displacement of the corresponding Gaussian propagator is sqrt($2Dt$).

## Motion-based analysis of translating mRNA

Displacements of single trajectories of mRNAs in the cytosol are calculated based on their time progression. Each trajectory longer than 10 steps was used to calculate the mean square displacement in 1 s. Each trajectory was then categorized based on the MSD cut-off of 0.055 μm$^2$ based on equation (2). For determining the translation fraction, we calculated by dividing the number of trajectories that are classified as translation by the number of total classified trajectories.

To generate pseudocoloured single-molecule movies, the positions and time-mark from categorized trajectory was mapped onto two blank matrices ($x,y,t$) where one represents translationally active and the other represents translationally silent mRNAs. The resulted single pixel movies were saved as tagged image file (tif). The gaussian blurring of 50 nm was applied to mimic the localization accuracy of each molecule. The resulted movie was then overlaid on the images of corresponding ER (Sec61β).

## SunTag intensity analysis and FRAP

Foci of scFv$_{GCN4}$-sfGFP signal were localized and quantified using Track-Mate. The background signal was subtracted and each ($x,y,t$) coordinate was correlated to the trajectories of corresponding mRNAs so that the maximum distance away is less than or equal to two pixels. Then, only mRNAs that are in the cytoplasm were selected for analysis. We calculated the translating fraction of mRNA by dividing SunTag(+) mRNAs over all tracked mRNAs in the cytoplasm.

FRAP experiments were performed on a customized Nikon TiE inverted scope equipped with a Yokogawa spinning-disk scan head (CSU-X1, Yokogawa) and Andor iXON EM-CCD cameras. The excitation and emission was performed on 100× Apo TIRF 1.49NA oil-immersion objective (Nikon). Photobleaching was performed using a Bruker Mini-scanner that scans the region of interest. Sequential excitation of 488 and 640 nm laser was performed to collect scFv–sfGFP and MCP–HaloTag signal, respectively through a multi-bandpass emission filter (Chroma, ZET405/488/561/647 nm). The cells were imaged prior to bleaching for 10 consecutive frames and imaged post-bleaching at 5 or 10 s per frame for >5 min. Particles were tracked using Imaris spot tracker (Oxford Instrument) and corresponding bleached SunTag signal was analysed over time.

## Lysosome–SunTag correlation

Lysosomes were tracked using TrackMate with the proper threshold to detect the centre of the lysosome. The distance from the centre of lysosome to each mRNA positions were calculated. The minimum distance from lysosome to each mRNA was then used for analysis. SunTag signals that were quantified during the SunTag intensity analysis was referenced to the minimum lysosome distance. The histogram of average SunTag intensity versus average minimum lysosome distance was quantified by binning the lysosome distance at 500 nm interval, starting from 750 nm to 4.25 μm. The SunTag intensity of the corresponding average lysosome distance binned at a 500-nm interval starting from 750 nm to 4.25 μm was used to generate the histogram.

## Lysosome–ER recruitment and analysis

To optogenetically recruit lysosomes and the ER, we constructed LAMP1–iLID and Sec61β–SspB expression plasmids. Specifically, LAMP1 was fused at its C-terminus to the photosensitive improved light-inducible dimer (iLID) domain, whereas Sec61β (a subunit of the ER Sec61 translocon complex) was fused at its N-terminus to the iLID-binding partner SspB. Distinct fluorescent tags (HaloTag for LAMP1–iLID and mCherry for Sec61β–SspB) were inserted between each protein domain. Optogenetic activation was initiated by exposing U-2 OS cells—either wild-type or lacking endogenous LNPK (LNPK KO)—to 488 nm laser in the region of interest. Time-lapse images were captured every 1–2 s throughout the recruitment assay.

All image processing and analyses were performed using ImageJ/Fiji. Lysosomes were segmented and tracked using the TrackMate plugin with a threshold-based method. Custom MATLAB scripts were then used to measure ER fluorescence intensity within the tracked lysosomal regions over time, and the resulting intensities were normalized for comparative analysis.

### siRNA knockdown and knockdown efficiency

Twenty-five picomoles of small interfering RNAs (siRNAs) targeting *LNPK* or *CLIMP63* (Dharmacon) were transfected into U-2 OS cells using Lipofectamine RNAiMAX (ThermoFisher) in Opti-MEM (ThermoFisher) in a 6-well chamber (Corning). The transfected cells were incubated for 48 to 72 h before imaging. Prior to imaging, the total RNA of the cells from same-day transfection was collected using RNA extraction kits (NEB). The concentration of total RNA was measured using UV spectrometer (ThermoFisher) and performed One-Step quantitative PCR with reverse transcription (RT–qPCR) (NEB) on Roche LightCycler. Knockdown efficiency was calculated based on calculating $\Delta\Delta C_q$ using *ACTB* mRNA as a standard. The list of primers used for this is listed in Supplementary Table 2.

For quantification of XBP-1 splicing in control, LNPK-KD and LNPK-KO cells under different treatments, we extracted total RNA from the cells using Monarch Total RNA extraction kit (NEB). The concentration of total RNA was measured using Nanodrop (ThermoFisher) and performed One-Step RT–qPCR (NEB) on Roche LightCycler using the provided protocol. The extent of unspliced and spliced transcripts of XBP-1 was quantified based on calculating $\Delta C_q$ using total XBP-1 transcript as a standard. The ratio between the estimated fraction of spliced/unspliced XBP-1 was as $2^{-\Delta Cq,\text{spliced}}/2^{-\Delta Cq,\text{unspliced}}$.

### CRISPR knockout of *LNPK*

To generate cells with targeted knockout of human *LNPK*, single guide RNAs (sgRNAs) were designed against *LNPK* (Synthego). The sgRNA sequences (AGCAAAAAAUGGGAGUGUCA, UGUAAACAGAUAGAGAACUG, UCAAGCAUUGGAAGAAUUUA) were then assembled with SpCas9 2NLS Nuclease (Synthego) as per the manufacturer's instructions. The resulting sgRNA–Cas9 complex was then electroporated into U-2 OS cells using a Lonza 4D nucleofector programmed for U-2 OS cells. The cells were expanded and cultured for single colonies by diluting them in 96-well plates (Corning). The genome of each expanded colony was then collected using the Monarch Genomic DNA purification kit, and PCR was performed on the purified DNA using primers provided by Synthego (listed in the Supplementary Table 3). The resulting Sanger sequencing data was analysed using Synthego software to identify positive colonies. These colonies were then checked for LNPK expression by collecting the lysate and performing immunofluorescence or immunoblotting as described above.

### Membrane protein extraction and quantification

For whole-cell extract, U-2 OS cells were collected via scraping the plate in lysis buffer (25 mM Tris-HCl pH 7.6, 150 mM NaCl, 1% NP-40, 1% sodium deoxycholate, 0.1% SDS) supplemented with EDTA-free protease inhibitor cocktail (Roche) and PhoSTOP (Roche), followed by sonication. Membrane proteins were extracted from one million U-2 OS control and LNPK-KO cells using the Mem-PER PLUS Membrane Protein Extraction Kit (ThermoFisher, 89842) following manufacturer's protocol. Protein quantification was performed using the Pierce BCA Protein Assay Kit (ThermoFisher, 23225), with absorbance measured at 562 nm using a Tecan 96-well plate reader.

### HPG incorporation assay

The Click-It HPG Alexa Fluor 488 protein synthesis kit (ThermoFisher) was used to perform the HPG incorporation assay. Prior to the addition of HPG (50 µM), U-2 OS cells were incubated with methionine-free DMEM for 1 h. HPG was incorporated into the cells over a period of 1.5 h. As HPG can also be incorporated into mitochondria, a cycloheximide incubation control was incorporated for all tested conditions. To monitor HPG incorporation into the membrane fraction only, HPG-incorporated cells were incubated with a solution containing 0.025% digitonin, 115 mM potassium acetate, 25 mM HEPES pH 7.4, 2.5 mM MgCl$_2$, 2 mM EGTA, and 150 mM sucrose for 3 min at 37 °C and proceeded with fixation. For the whole-cell fraction, the cells were fixed using 4% PFA and 0.2% glutaraldehyde for 15 min, and the washing and labelling protocol followed the manufacturer's instructions. The labelled cells were mounted using VECTASHIELD antifade with DAPI (Vectorlabs), imaged using Zeiss 980 AiryScan, and quantified using CellProfiler. To remove the effect from mitochondrial translation, the quantified signal of each condition was subtracted with the corresponding cycloheximide treatment.

### CDF curve fit

Cumulative distribution function (CDF) curves were fit with a three-component fit (equation (3)):

$$\text{CDF}(r) = 1 - \left( A_1 \times e^{-\left(\frac{r^2}{4D_1 t}\right)} + A_2 \times e^{-\left(\frac{r^2}{4D_2 t}\right)} + A_3 \times e^{-\left(\frac{r^2}{4D_3 t}\right)} \right) \quad (3)$$

Where $A_x$ is the fraction of molecules with the diffusion coefficient $D_x$, $D_x$ is the diffusion coefficient of the $x$th diffusive species in µm$^2$ s$^{-1}$, $t$ is the single lag time in seconds, and $r$ is the single displacement in micron.

### RNA extraction from cultured cells

U-2 OS control and LNPK-KO cells were plated in a 6-well dish, briefly washed with PBS, and collected in 1 ml Trizol (Invitrogen, 15596026). Total RNA was extracted following the manufacturer's protocol (Invitrogen, 15596026), and both the quality and quantity of the RNA were subsequently assessed using an RNA Qubit (Invitrogen) and an Agilent Bioanalyzer.

### RNA sequencing

cDNA was prepared from 1 ng of total RNA as previously described[37]. Reverse transcription was performed using a barcoded 3' polyT primer containing a unique molecular identifier (UMI), a 5' template-switch oligo (Integrated DNA Technologies), and Maxima H Minus Reverse Transcriptase (ThermoFisher, EP0752), followed by PCR amplification and cDNA purification.

Tagmentation and library preparation were then conducted with 600 pg of cDNA per sample using a modified Nextera XT (Illumina) protocol with dual-indexed P5NEXTPT5 and i7 primers (IDT). The libraries were purified according to the Nextera XT protocol, quantified by qPCR using the Kapa Library Quantification Kit (Kapa Biosystems), and sequenced on a NextSeq 2000. Read 1 (26 bp) captured the sample barcode and UMI, while read 2 (50 bp) sequenced the 3' cDNA fragment. A PhiX control library (Illumina) was spiked in at a final concentration of 15% to enhance colour balance in Read 1.

Sequence alignment was performed as described previously[38]. Sequencing adapters were trimmed from the reads using Cutadapt v2.10 prior to alignment with STAR v2.7.5c against the *Homo sapiens* GRCh38 genome assembly from Ensembl. Gene counts were generated using the STARsolo algorithm and subsequently analysed using a version of DESeq2 on MatLab.

### Solvents and chemicals

We used water (Optima, W6–4), acetonitrile (Optima, A9554), methanol (Optima, A454SK-4), formic acid (Pierce, PI28905), acetone (A949-1, Fisher) and RapiGest SF Surfactant (Waters,186001861). Dithiothreitol (DTT, R0861, Thermo scientific), trifluoroacetic acid (TFA, 302031–100 ML), iodoacetamide (IAA, 407710) and ammonium bicarbonate (40867) were purchased from Sigma Aldrich. Trypsin (V5113) was purchased from Promega.

## Membrane protein digestion and desalting

Membrane proteins were precipitated with cold acetone at 1:4 (v:v) ratio, −20 °C overnight. Protein pellets were centrifuged and washed with cold acetone. Residue acetone was then air dried, and the proteins were re-dissolved using 1% RapiGest according to the manufacturer's protocol. Afterwards, proteins were reduced with 5 mM DTT (65 °C, 30 min) and alkylated using 11 mM IAA at ambient temperature in the dark for 30 min. Trypsin was added at a protein-enzyme ratio of 50:1 for overnight digestion at 37 °C. The digestion was quenched by adding 10% TFA to pH of ~1. The digest was further incubated at 37 °C for 45 min to degrade the RapiGest. The solutions were then centrifuged at 14,000 rpm for 10 min and the supernatant peptides were collected. Desalting of the peptides was performed using C18 ZipTip (Millipore, ZTC18M096) and the eluents were dried using SpeedVac (Thermo Scientific). The samples were stored at −80 °C before being re-suspended in 0.1% formic acid for LC–MS/MS analysis.

## LC–MS/MS analysis

Liquid chromatography separation was performed on a Vanquish Neo System (Thermo Scientific) with an IonOptik Aurora Ultimate C18 column (25 cm length × 75 µm inner diameter × 1.7 µm particle size) at 50 °C and 300 nl min$^{-1}$ flow rate. Mobile phase A consisted of 0.1% formic acid in water. Mobile phase B consisted of 0.1% formic acid in 80% acetonitrile. Eluting peptides were ionized by electrospray ionization and then analysed by an Orbitrap Ascend Tribrid mass spectrometer (tune v.4.1.4244, Thermo Scientific). Ion transfer tube temperature was set to 275 °C. Source positive ion voltage was set to 1,850 V. For single-shot proteomics with data independent acquisition (DIA), the MS1 scan resolution was set to 120,000 (at $m/z$ 200). MS1 scan range was 400–900 $m/z$, AGC target was 225%, maximum injection time mode was set to auto. Precursors were isolated with an isolation width of 12 $m/z$ covering 145–1,450 $m/z$. Precursors were fragmented by HCD at an NCE of 26%. MS2 scans were acquired by Orbitrap with a resolution of 15,000. Maximum injection time was 59 ms for the Ascend.

## Mass spectrometry data processing

For the acquired DIA data, raw files were processed directly using Proteome Discoverer (v.3.1.0.638) with the *H. sapiens* proteome FASTA file (UP000005640). The analysis applied CHIMERYS intelligent search algorithm, allowed for a maximum of one missed cleavage, with cysteine carbamidomethylation (+57.0215 Da) set as a fixed modification, and oxidation (M), phosphorylation (S,T) were selected as variable modifications. The target false discovery rate (FDR) was maintained at 0.01. Fragment mass tolerance was set as 20 ppm.

## Reporting summary

Further information on research design is available in the Nature Portfolio Reporting Summary linked to this article.

## Data availability

Source data underlying all graphs are provided with the paper in Fig-Share (https://doi.org/10.25378/janelia.30182434 (ref. 39)). Datasets of single-molecule imaging datasets (HILO and spinning-disk confocal movies of MS2- and SunTag-labelled secretome mRNAs, ribosome tracking and lysosome recruitment assays), processed particle tracking files, and quantified FRAP measurements have been deposited in Fig-Share (https://doi.org/10.25378/janelia.30182434 (ref. 39)). Uncropped gel and blot images corresponding to Fig. 5e and Extended Data Fig. 4 are provided in Supplementary Information. All listed plasmids are deposited to Addgene. Raw and processed RNA-seq data have been deposited in the Gene Expression Omnibus (GEO) under accession GSE309254. The mass spectrometry proteomics data have been deposited to the ProteomeXchange Consortium via the PRIDE partner repository with the dataset identifier PXD068879. Source data are provided with this paper.

## Code availability

Custom analysis code used in this study is available in FigShare (https://doi.org/10.25378/janelia.30182434 (ref. 39)).

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

**Acknowledgements** We thank K. Schaefer and D. Walpita for their assistance with fluorescence-activated cell sorting and cell culture; H. Yi and team for lentivirus preparation and purification; P. Li and C. Blackstone for providing the LNPK–GFP plasmid; A. Lemire, S. Shrestha and L. Wei for RNA-sequence analysis; and B. English, C. Gladkova, P. Sengupta, V. Wang, D. Hwang and all members of the J.L.-S. laboratory for insightful discussions. This work was funded by the following: Howard Hughes Medical Institute (to H.C., Y.-C.L., J.G., L.D.L and J.L.-S.), National Institute of Health grant R21-MH120496 (to Y.J.Y.), R01-NS083085 (to R.H.S.).

**Author contributions** H.C., Y.-C.L., Y.J.Y., R.H.S. and J.L.-S. conceptualized the project. J.G. and L.D.L. synthesized JF dyes, H.C. performed experiments, Y.J.Y. validated L10A–HaloTag function, H.C. and Y.-C.L. designed constructs and experiments for lysosome proximity measurements, N.W. performed and analysed mass spectrometry data, H.C. performed computational image analysis and simulations, J.L.-S. supervised the project. H.C. and J.L.-S. wrote the manuscript with input from all authors who approved the final version.

**Competing interests** The authors declare no competing interests.

**Additional information**
**Correspondence and requests for materials** should be addressed to Jennifer Lippincott-Schwartz.

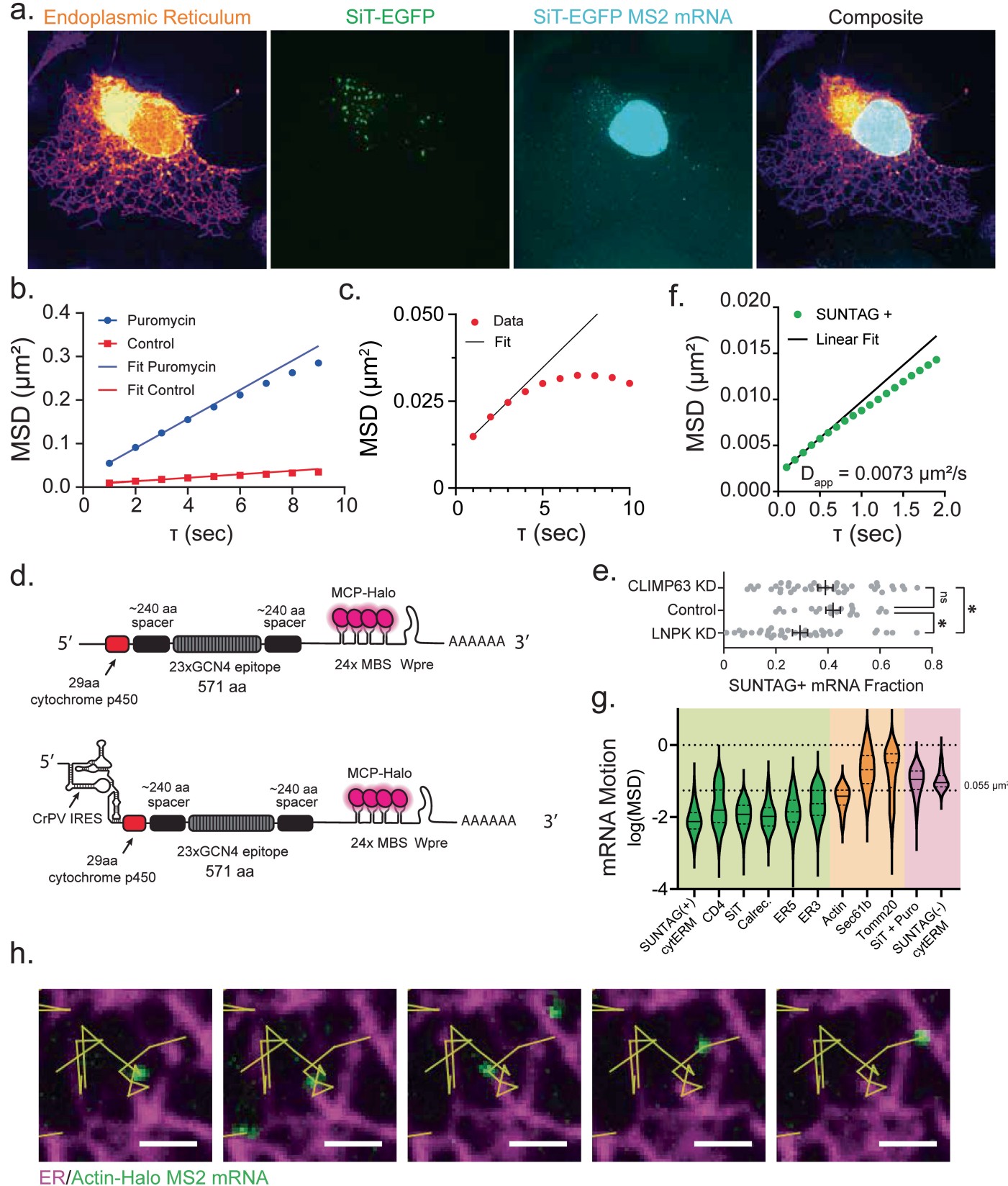

**Extended Data Fig. 1** | See next page for caption.

**Extended Data Fig. 1 | Motion distinguishes translating from nontranslating secretome mRNAs. a**, Golgi-like distribution of SiT-EGFP MS2 mRNA in U-2 OS cells. SiT-EGFP (green), MS2 (cyan), ER (orange). **b**, Mean squared displacement (MSD, $\mu m^2$) vs lag time ($\tau$, sec) for SiT-EGFP MS2 mRNAs under control (red) or puromycin (blue). Fitted curves: $D_{control} = 0.004\ \mu m^2/s$, $D_{puro} = 0.033\ \mu m^2/s$. **c**, MSD vs lag time of SiT-EGFP MS2 trajectories classified as translating ($MSD_{\tau=1sec} < 0.055\ \mu m^2$). **d**, Schematic of cytERM-SUNTAG MS2 construct (upper) and CrPV IRES cytERM-SUNTAG MS2 (lower). **e**, Boxplot of translating fractions of unmodified cytERM-SUNTAG MS2 in control (N = 20 cells), LNPK KD (N = 38), and CLIMP63 KD (N = 39). **f**, MSD vs lag time of SUNTAG(+) cytERM-SUNTAG MS2 mRNAs, fitted Dapp=0.007 $\mu m^2/s$. **g**, Violin plots of log($MSD_{\tau=1sec}$) for SUNTAG(+) cytERM-SUNTAG MS2 (n = 72), CD4-EGFP MS2 (n = 251), SiT-EGFP MS2 (n = 1616), Calreticulin-mEmerald MS2 (n = 250), ER5-mEmerald MS2 (n = 1155), ER3-mEmerald MS2 (n = 2475), β-actin-Halo MS2 (n = 100), SUNTAG-Sec61β MS2 (n = 314), TOMM20-Halo MS2 (n = 233), SiT+puromycin (n = 1608), and SUNTAG(−) cytERM-SUNTAG MS2 (n = 680). Red dotted line, $MSD = 0.055\ \mu m^2$. **h**, Time-lapse images of Actin-Halo MS2 mRNA (green) and ER (magenta). Scale bars, 1 μm.

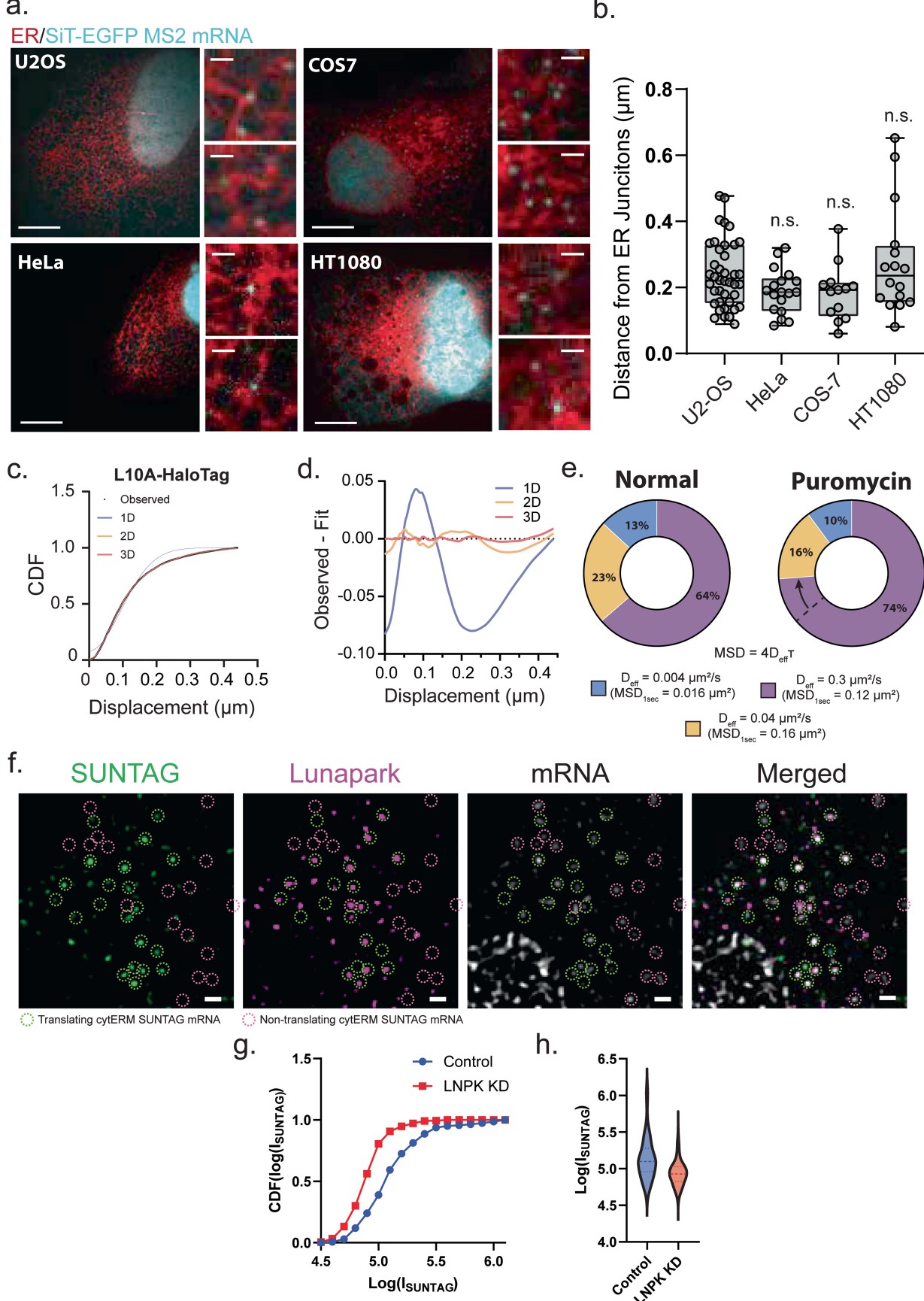

**Extended Data Fig. 2** | See next page for caption.

**Extended Data Fig. 2 | Translating mRNAs and ribosomes localize at ER junctions with LNPK. a**, Wide-field images of SiT-EGFP MS2 mRNAs (cyan) and ER (red) in U-2 OS, COS7, HeLa, and HT1080 cells. Large image scale bars, 10 µm; inset, 1 µm. **b**, Box plots of average distance of translating SiT-EGFP MS2 mRNAs (MSDτ = 1 sec<0.055 µm$^2$) to nearest ER junctions in U-2 OS, HeLa, COS7, and HT1080. Each dot, single cell mean. **c**, Cumulative distribution function (CDF) fitting of L10a-HaloTag ribosome displacements in control cells. Fits: one coefficient (1D, purple), two (2D, yellow), three (3D, red). **d**, Residual plots of observed vs fitted data (from c). **e**, Circular graph of ribosome mobility states in control vs puromycin-treated cells. Fractions: slow D = 0.004 µm$^2$/s (blue), medium D = 0.04 µm$^2$/s (yellow), fast D = 0.3 µm$^2$/s (purple). **f**, Wide-field image of cytERM-SUNTAG MS2 mRNAs (gray), SUNTAG signal (green), and anti-LNPK staining (magenta). Translating puncta circled (green dotted); non-translating puncta circled (magenta). Scale bars, 1 µm. **g**, Cumulative distribution function of log$_{10}$ SUNTAG intensity of cytERM-SUNTAG MS2 mRNAs in control vs LNPK KD cells. **h**, Distribution of log$_{10}$ SUNTAG intensities from cytERM-SUNTAG MS2 puncta in control vs LNPK KD cells.

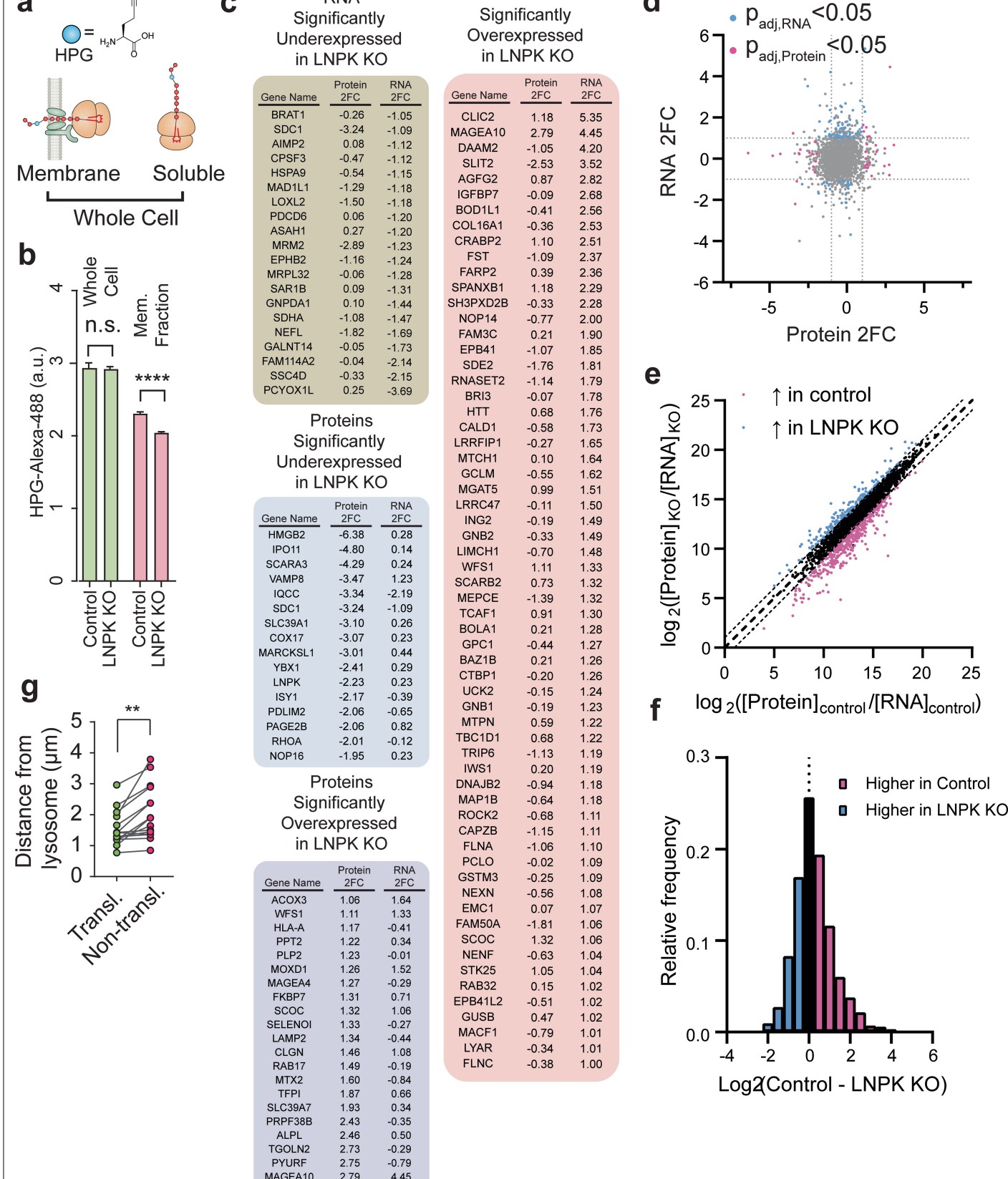

**Extended Data Fig. 3** | See next page for caption.

**Extended Data Fig. 3 | LNPK loss reduces membrane protein synthesis and translation near lysosomes. a**, Schematic of L-homopropargylglycine (HPG) incorporation assay to label newly synthesized proteins in whole-cell and membrane fractions. **b**, Click-labeled Alexa-488-HPG signal in whole-cell and digitonin-extracted membrane fractions of control (green) and LNPK KO U-2 OS cells (magenta). Whole-cell: no significant difference (unpaired two-tailed t-test, P = 0.8492). Membrane fraction: significant reduction in LNPK KO (****P < 0.0001). **c**, Tables of proteins and mRNAs differentially regulated in LNPK KO cells ( ≥ 2-fold change in protein or RNA, adjusted P < 0.05). **d**, Scatter plot of protein fold-change (Protein 2FC, LNPK KO/WT) vs RNA fold-change (RNA 2FC, LNPK KO/WT). Pink, proteins with adjusted P < 0.05; blue, mRNAs with adjusted P < 0.05. Dotted line, 2-fold threshold. **e**, $Log_2$ protein/mRNA ratios in KO vs WT. Central dotted line, 1:1 ratio; flanking dotted lines, 2-fold boundaries. Pink dots, higher in control; blue, higher in LNPK KO. **f**, Histogram of $log_2$ difference in protein/mRNA ratio between KO and control. Pink bars, genes with higher ratios in control; blue bars, higher ratios in LNPK KO. **g**, The average distance between SiT-EGFP mRNA and the nearest lysosome (µm) of translating (T, green) and non-translating (NT, red) mRNA. Each line represents the same cell. N = 15 cells.

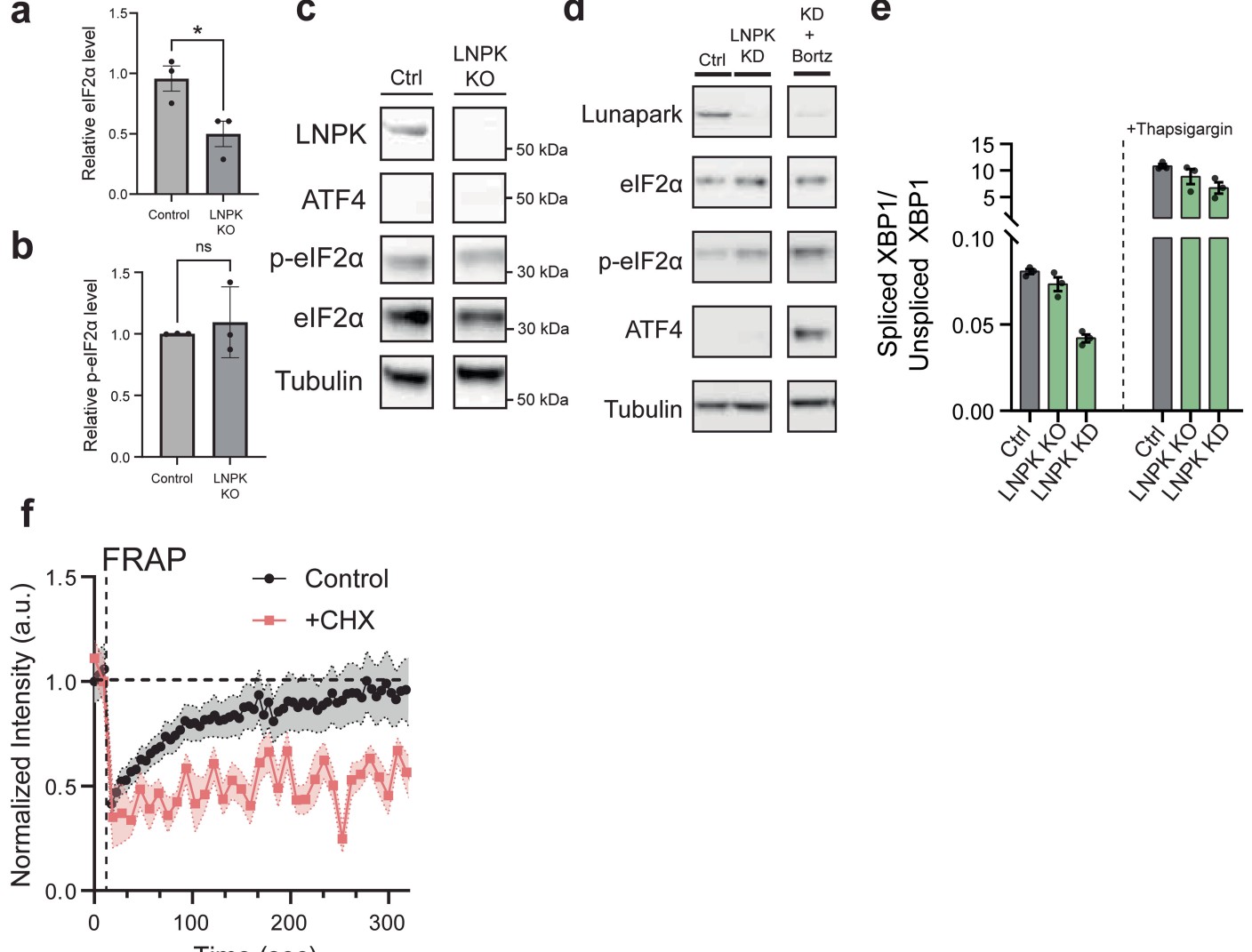

**Extended Data Fig. 4 | LNPK depletion alters eIF2 signaling and translation recovery. a**, Relative eIF2α levels (normalized to tubulin) in control and LNPK KO U-2 OS cells. Unpaired two-tailed t-test, P = 0.037. **b**, Relative phosphorylated eIF2α (p-eIF2α) levels in control and LNPK KO cells. Unpaired two-tailed t-test, P = 0.14. **c**, Immunoblots of ATF4, Tubulin, eIF2α, and p-eIF2α from control (WT), LNPK KO, and U-2 OS cells treated with bortezomib (100 nM, 4 h). All blots imaged under identical conditions. **d**, Western blots of LNPK, eIF2α, p-eIF2α, ATF4, and Tubulin from control siRNA, LNPK siRNA (LNPK KD), and LNPK siRNA+bortezomib (100 nM, 4 h). **e**, qPCR quantification of spliced/unspliced XBP-1 ratio in control, LNPK KO, and LNPK KD cells, with or without thapsigargin (1 μM, 1 h). Data show no induction of unfolded protein response by LNPK loss. **f**, FRAP recovery of cytERM-SUNTAG MS2 puncta in control cells (black, n = 32) vs cycloheximide-treated cells (red, n = 10). Recovery is blocked by cycloheximide, confirming fluorescence recovery reflects new peptide synthesis rather than antibody exchange.

# Reporting Summary

## Statistics

For all statistical analyses, confirm that the following items are present in the figure legend, table legend, main text, or Methods section.

| n/a | Confirmed | |
|---|---|---|
| ☐ | ☒ | The exact sample size (*n*) for each experimental group/condition, given as a discrete number and unit of measurement |
| ☐ | ☒ | A statement on whether measurements were taken from distinct samples or whether the same sample was measured repeatedly |
| ☐ | ☒ | The statistical test(s) used AND whether they are one- or two-sided<br>*Only common tests should be described solely by name; describe more complex techniques in the Methods section.* |
| ☒ | ☐ | A description of all covariates tested |
| ☐ | ☒ | A description of any assumptions or corrections, such as tests of normality and adjustment for multiple comparisons |
| ☐ | ☒ | A full description of the statistical parameters including central tendency (e.g. means) or other basic estimates (e.g. regression coefficient) AND variation (e.g. standard deviation) or associated estimates of uncertainty (e.g. confidence intervals) |
| ☐ | ☒ | For null hypothesis testing, the test statistic (e.g. *F*, *t*, *r*) with confidence intervals, effect sizes, degrees of freedom and *P* value noted<br>*Give P values as exact values whenever suitable.* |
| ☒ | ☐ | For Bayesian analysis, information on the choice of priors and Markov chain Monte Carlo settings |
| ☒ | ☐ | For hierarchical and complex designs, identification of the appropriate level for tests and full reporting of outcomes |
| ☒ | ☐ | Estimates of effect sizes (e.g. Cohen's *d*, Pearson's *r*), indicating how they were calculated |

*Our web collection on statistics for biologists contains articles on many of the points above.*

## Software and code

Policy information about availability of computer code

| Data collection | Imaging data were acquired on customized Nikon TiE microscopes configured for spinning-disk and HILO illumination with Andor EM-CCD cameras, using Nikon Elements AR 6.0 for spinning-disk imaging and ZEN Black 2.3/Zen Blue for confocal imaging. Western blot signals were captured on a Bio-Rad ChemiDoc system. RNA sequencing was performed on an Illumina NextSeq 2000, and mass spectrometry was carried out using a Thermo Orbitrap Ascend Tribrid mass spectrometer coupled to a Vanquish Neo LC system. |
|---|---|
| Data analysis | MATLAB (custom scripts for MSD analysis, and Monte Carlo simulations), Imaris, Illastik, Fiji, TrackMate (Fiji), Imaris, CellProfiler, STAR (v2.7.5c) for RNA-seq alignment, DESeq2 in MATLAB for transcriptome analysis, and DIA-NN (v1.9.1) for mass spectrometry |

For manuscripts utilizing custom algorithms or software that are central to the research but not yet described in published literature, software must be made available to editors and reviewers. We strongly encourage code deposition in a community repository (e.g. GitHub). See the Nature Portfolio guidelines for submitting code & software for further information.

## Data

Policy information about availability of data

All manuscripts must include a data availability statement. This statement should provide the following information, where applicable:

- Accession codes, unique identifiers, or web links for publicly available datasets
- A description of any restrictions on data availability
- For clinical datasets or third party data, please ensure that the statement adheres to our policy

Source data underlying all graphs are provided with the paper in FigShare (DOI:10.25378/janelia.30153850). Datasets of single-molecule imaging datasets (HILO and spinning disk confocal movies of MS2- and SUNTAG-labeled secretome mRNAs, ribosome tracking, and lysosome recruitment assays), processed particle-tracking files, and quantified fluorescence recovery after photobleaching (FRAP) measurements have been deposited in FigShare (DOI:10.25378/janelia.30153850). Uncropped gel and blot images corresponding to Figs. 5e and Extended Data Fig. 4 are provided in Supplementary Information. All listed plasmids are deposited to Addgene.

## Research involving human participants, their data, or biological material

Policy information about studies with human participants or human data. See also policy information about sex, gender (identity/presentation), and sexual orientation and race, ethnicity and racism.

| | |
|---|---|
| Reporting on sex and gender | N/A |
| Reporting on race, ethnicity, or other socially relevant groupings | N/A |
| Population characteristics | N/A |
| Recruitment | N/A |
| Ethics oversight | N/A |

Note that full information on the approval of the study protocol must also be provided in the manuscript.

# Field-specific reporting

Please select the one below that is the best fit for your research. If you are not sure, read the appropriate sections before making your selection.

☒ Life sciences ☐ Behavioural & social sciences ☐ Ecological, evolutionary & environmental sciences

For a reference copy of the document with all sections, see nature.com/documents/nr-reporting-summary-flat.pdf

# Life sciences study design

All studies must disclose on these points even when the disclosure is negative.

| | |
|---|---|
| Sample size | Sample sizes were chosen based on established practice for single-molecule imaging and translation studies, and were sufficient to detect significant differences. Exact n values (cells, trajectories, replicates) are reported in each figure legend. For example, thousands of individual mRNA trajectories across 13–40 cells were analyzed per condition, and western blots were repeated with three independent lysates. |
| Data exclusions | Trajectories shorter than 10 steps were excluded from MSD-based classification, as these could not be reliably analyzed. Nuclear signals were excluded due to MCP accumulation, and poor-quality images were omitted prior to analysis. |
| Replication | All key findings were reproduced in at least two independent biological replicates. RNA-seq and proteomics experiments included biological replicates. |
| Randomization | We did not randomize any aspects of this study. |
| Blinding | Data analysis was performed using automated pipelines (MATLAB, TrackMate, Imaris) without prior knowledge of sample identity. Manual validation of ER junction assignments and PLA puncta was performed without knowledge of condition where feasible. |

# Reporting for specific materials, systems and methods

We require information from authors about some types of materials, experimental systems and methods used in many studies. Here, indicate whether each material, system or method listed is relevant to your study. If you are not sure if a list item applies to your research, read the appropriate section before selecting a response.

## Materials & experimental systems

| n/a | Involved in the study |
|-----|----------------------|
| ☐ | ☒ Antibodies |
| ☐ | ☒ Eukaryotic cell lines |
| ☒ | ☐ Palaeontology and archaeology |
| ☒ | ☐ Animals and other organisms |
| ☒ | ☐ Clinical data |
| ☒ | ☐ Dual use research of concern |
| ☒ | ☐ Plants |

## Methods

| n/a | Involved in the study |
|-----|----------------------|
| ☒ | ☐ ChIP-seq |
| ☒ | ☐ Flow cytometry |
| ☒ | ☐ MRI-based neuroimaging |

## Antibodies

| Antibodies used | Rabbit LNPK antibody (Sigma, HPA014205-25), Mouse EEA1 antibody (BD Biosciences, 610456), Mouse LAMP1 antibody (Abcam, ab25630), Mouse TOMM20 antibody (ProteinTech, 66777-1-Ig).Rabbit anti-ATF-4 antibody (Cell Signaling, 11815S), Rabbit anti-LNPK antibody (Sigma, HPA014205-25), Mouse anti-Tubulin antibody (Millipore, 05-829), Rabbit anti-eIF2S1 (phosphor S51) antibody (Abcam, ab32157), and Rabbit anti-eIF2S1 antibody (Atlas Antibodies, HPA064885). Secondary antibodies used here are: Goat anti-mouse Ig H&L-HRP (Abcam, ab205719), Goat anti-rabbit Ig H&L-HRP (Abcam, ab205718), Rabbit REEP5 antibody (ProteinTech, 14643-1-AP). |
|-----------------|-------------|
| Validation | Refer to manufacturer's website |

## Eukaryotic cell lines

Policy information about cell lines and Sex and Gender in Research

| Cell line source(s) | All experiments in the paper are performed using U-2 OS, COS7, HEK293T, HeLa, and HT1080 cells from ATCC, experiments are performed within 40 passages of the initial shock provided. |
|---------------------|-------------|
| Authentication | We have not performed any authentication of the lines. There were no obvious differences in any property examined. |
| Mycoplasma contamination | Cells were all free of mycoplasma at the time of experimentation, and are tested routinely during passaging. |
| Commonly misidentified lines (See ICLAC register) | No commonly misidentified cell lines were used in this study. |

## Plants

| Seed stocks | N/A |
|-------------|-----|
| Novel plant genotypes | N/A |
| Authentication | N/A |

