## [Peer Review File · Nature]

Secretome translation shaped by Lysosomes and Lunapark-marked ER Junctions

Corresponding Author: Dr Jennifer Lippincott-Schwartz

Version 0:

Reviewer comments:

Referee #1

(Remarks to the Author)

In this paper Choi et al using advanced single molecule imaging approaches to study mRNA translation on the ER. First, they show that secretome mRNAs are localized on the ER when translated, but not when untranslated, confirming previous studies. They then go on to show that when translated, secretome mRNAs localize specifically to 3-way junctions of the ER and that they frequently localize in the vicinity of the lunapark protein and lysosomes. Finally, they also reveal a functional link between the translation of secretome mRNAs and lunapark and lysosomes, as secretome mRNA translation is reduced upon either lunapark knockdown (KD) or inhibition of normal lysosome function. The authors offer some mechanistic insight into this function link by showing that the negative effect of lunapark knockdown on secretome translation can be rescued by ISRIB, suggesting that lunapark KD activates the integrated stress response, which in turn may inhibit secretome mRNA translation. Overall, this is a well written, interesting and technically mostly very solid paper, that identifies a new way by which secretome mRNAs are spatially regulated, and identifies an interesting role for lysosomes in regulating their translation.

In addition to a number of technical concerns (outlined below), I have one major concern that affects the overall model proposed in the paper. The authors suggest that lunapark controls secretome mRNA translation through control of translation initiation (through the ISR and eIF2alpha phosphorylation), but when assessing the translation defect in secretome mRNAs upon lunapark knockdown, I am not convinced that the phenotype they are observing actually reflects a defect in translation initiation. My concerns are focused on Figures 5g-i where the translation phenotype is described:

The authors are assuming that all GFP recovery reflects turnover of ribosomes on the mRNA, but also the SUNTAG antibody can turnover on the nascent polypeptides. The authors need to include a control where the cells are treated with a translation inhibitor (eg emetine) and show the GFP recovery rates. Even if such controls have been performed previously, it is important to do this under the precise experimental conditions used here.

The recovery rate after photobleaching seems very fast. Looking at the graph in 5i, 50% of the recovery has occurred at around 70s (not 100s). Translation elongation rates in cancer cells is typically ~3 amino acids per second, so in 70 sec the ribosomes will have translated just 210 amino acids, which I assume is quite a bit less than half of the mRNA. In fact, in this time more than half of the mRNA should have been translated by new ribosomes (ribosomes loaded onto the mRNA after photobleaching) to recover 50% of the initial fluorescence, since ribosomes on the upstream half of the mRNA produce less fluorescence than those in the downstream half (because they have not yet synthesized all the epitopes), so for recovery of half the total fluorescence, ribosomes will need to translate more than half of the mRNAs. Can this really be done in just 70 sec? Can the authors perform a harringtonine run-off experiment to measure the elongation rate in their system to see if it quantitatively matches with the FRAP values?

I have major concerns about the experiments and interpretation of the lunapark KD and FRAP analysis. If the initiation rates are lower after lunapark knockdown, then the GFP spot intensity should be lower, but the recovery rate after photobleaching (relative to the intensity at the start of the FRAP experiment) should be similar to control cells, because ribosome turnover rate as assessed by FRAP reports on elongation rate, not initiation rate. In addition, the major phenotype of lunapark KD is

incomplete recovery of fluorescence not slower recovery. It is not clear to me why recovery would be incomplete, perhaps it suggests that ribosomes are stalled on the mRNA? Together, the data are not consistent with the authors' model that lunapark controls translation initiation through eIF2alpha. To me this is a major problem for the main conclusions of the paper. If lunapark knockdown is not causing a translation initiation defect, then it is unclear how the lunapark part of the paper is linked to the eIF2alpha and ISR parts of the paper.

Other points:

Fig 2D: the average distance of endogenous CD9 mRNAs from ER 3-way junctions AFTER puromycin addition is just ~0.16 micrometer, which is similar or even slightly closer to 3-way junctions as mRNAs that are translating on the ER described in figure 2b. Can the authors really conclude that endogenous CD9 mRNAs are targeted to 3-way junctions through translation? Why are CD9 mRNAs not at a distance of 0.5 micrometer from 3-way junctions after puromycin treatment as is the case for the actin mRNAs?

In this reviewer's opinion, cutting off the y-axis to make effects look bigger than they are, is not appropriate, especially if there is no break indicated in the axis. This is seen in multiple graphs throughout the paper, including 2d, 3g, 4d, 5b,f.

Figure 3: The authors state: "Together, the enrichment of translating secretome mRNAs at Lunapark containing ER junctions suggested". It was not entirely clear to me whether all 3-way junctions have lunapark staining. If not, do translating mRNAs preferentially localize to lunapark-containing 3-way junctions in comparison to 3-way junctions devoid of lunapark? I could not find this information, but this seems needed to make the claim above.

Figure 4b: The number of PLA background foci likely depends on the overall amount of staining with the different antibodies (eg Lamp1, EEA1). Thus, it would be nice to see a control here, for example by performing PLA between the same organelle markers and another ER protein, not localized at 3-way junctions.

Figure 5a. Here the translating fraction is plotted, but it would be nicer to assess whether the distance effect shown in figure 4d is affected by these treatments. I would recommend plotting SunTag intensity relative to distance from lysosomes from the perturbations shown in fig 5a.

Figure 5: I would also suggest to plot the GFP intensities of translation, not just the fraction translating, which does not tell the whole story if one wants to know whether translation is affected by a specific treatment.

Figure 5d: Is the effect of thapsigargin treatment on translation specific to secretome mRNAs? If not, this would not be a very novel finding, as it is well known that an increase in eIF2alpha phosphorylation reduces overall translation. The authors should see if translation of, for example, the actin mRNA is not reduced by thapsigargin treatment (which I would find surprising if it isn't, but I might be mistaken).

Figure 5e. It seems that the eIF2a protein levels go down, rather than that the phosphorylated form goes up. Have the authors repeated this experiment multiple times and is this effect reproducible? This is a somewhat surprising effect, that is not necessarily in line with the simple interpretation of the authors (that the ratio of phosphorylated vs non-phosphorylated protein changes).

Discussion; "Moreover, we showed that Lunapark, when interacting with lysosomes, provides a platform for more efficient translation of secretome mRNAs." As well as other related statements (eg lines 374-375, 382-383)". I'm not sure if this statement is warranted. Do the authors show that the effects of lunapark KD are specifically caused by loss of an interaction of Lunapark with lysosomes? Are lysosomes no longer positioned near ER junctions in the absence of lunapark? It seems that lunapark KD activates the ISR, which mediates the reduced translation of secretome mRNAs, but it is not clear to me whether this effect is related to lysosomes.

Minor points

Fig S1A: the middle panel doesn't look like magenta. Also, a merge of protein and mRNA staining would be nice to include.

Fig 1E: schematic is a bit hard to see, due to the small size

In the text describing figure one, the authors use the term 'confined motion'. However, they don't show confined motion, they just show slow motion. They can of course assess whether motion is confined from their MSD curves (if the MSD curves plateau), and should do so if they wish to use the term confined motion.

Fig 2B. While the actin mRNA control is nice, it could be informative to plot the average distance from a random point within the analyzed area to an ER 3-way junction. This will show how strongly the secretome mRNAs are enriched at 3-way junctions and whether actin mRNAs are also enriched near 3-way junctions, or are truly not enriched, as claimed on line 159

Fig S3 should include quantification, I don't think the reader can assess whether the average distance from mRNAs to 3-way junctions is 0.2 or 0.5 micrometer from these images.

(Remarks on code availability)

Referee #2

(Remarks to the Author)

The paper by Choi et al investigates how the translation of mRNAs encoding membrane and luminal proteins on the endoplasmic reticulum (ER) is coordinated with the positioning of the ER relative to other organelles, specifically lysosomes. Through live cell single molecule imaging, the researchers discover that the translation of these mRNAs primarily occurs at ER junctions marked by the protein Lunapark and in the vicinity of lysosomes. Depletion of Lunapark leads to a decrease in ribosome densities on secretome mRNAs near lysosomes and a selective reduction in the translation of these mRNAs. Secretome translation at ER-lysosome junctions was affected by the ISR and by conditions that affect lysosomal function, implicating a link between local protein synthesis on the ER and local amino acid release by lysosomes. Overall, the findings propose a novel regulatory mechanism where lysosomes, in conjunction with Lunapark, locally pattern and regulate the translation of mRNAs encoding secretory and membrane proteins.

The work is based on elegant live cell single molecule imaging, which provides detailed insights into mRNA translation dynamics in real-time and propose a role for Lunapark-mediated junctions in this process. Overall, the findings point to a deep connection of intracellular protein synthesis and organelle coordination.

However, significant weaknesses are noted, particularly in the proposed mechanisms underlying many of the observed phenomena. As it stands the manuscript is largely descriptive and provides no insights into how proximity to Lunapark leads to translation initiation. The evidence that Lunapark and ISR specifically affect translation of secretome mRNAs is not sufficiently demonstrated to support the key claims of the manuscript. Similarly, the importance of lysosome proximity is barely hinted at by the data provided, with no clear rationale.

Specific points:

- 1- Fig. 3g: this result is difficult to understand. Shouldn't digitonin permeabilization shut down translation altogether? The reported difference in HPG incorporation between control and Lunapark KO seems modest and could have many explanations. Independent validation using a more sophisticated approach such as ribosome footprinting should be carried out.
- 2- The PLA experiments supporting preferential proximity of lysosomes to Lunapark-positive ER junctions should be properly controlled by comparing with other ER proteins not found at junctions, such as calnexin, VAPs, and others.
- 3- Fig. 4d: KD of Lunapark causes translating mRNAs to no longer be closely apposed to lysosomes. How Lunapark loss impacts ER-lysosome contacts in general is not assessed, thus the specificity of this effect is unclear. Appropriate fluorescence and ultrastructural data should be provided.
- 4- If proximity to lysosomes is key to preferential translation of secretory transcripts, then manipulations that force proximity between ER and lysosomes should specifically rescue translation of membrane and secretory proteins when Lunapark is depleted.
- 5- Fig.5c lack of effect of rapamycin and, especially, Torin on secretome mRNA translation is quite unexpected as mTOR inhibition has broad, translation-suppressing effects and mTOR signaling is increasingly connected to secretion. Again, this raises a concern that using SunTAG alone is not sufficient to assess translation, and that a more fine-grained type of analysis, such as ribosome profiling, should be employed.
- 6- Inhibiting the ISR is a rather broad manipulation that could affect all substates, not just secretory ones. Can the authors show that, indeed, ISRIB preferentially enhances secretory transcript translation and that this effect depends on ER proximity to lysosomes?
- 7- Fig. 5h-5i, what is the effect of ISRIB alone (w/o Lunapark KD) on SUNTAG recovery? ISRIB can boost translation under baseline conditions as well, thus whether its action represents a true bypass of Lunapark KD or a boost irrespective of Lunapark status should be assessed.
- 8- The idea that Lunapark loss results in 'noncanonical' (ATF4- and UPR-independent) ISR is potentially interesting but vague. Lacking a more defined molecular mechanism, one is left wondering whether a general disruption of ER morphology and/or decreased association of translation initiation factors could explain the reduced translation at ER-associated ribosomes.
- 9- The interpretation of how lysosomal inhibition impacts secretome mRNA translation is largely speculative. pH dissipation has many pleiotropic effects, as no direct test of amino acid release from lysosome is carried out. There are several lysosomal amino acid transporters that the authors could knock down to determine if any of them is required for this process.

(Remarks on code availability)

Referee #3

(Remarks to the Author)

This is a very interesting manuscript that localizes translation of mRNA coding for secretory proteins to ER junctions enriched for the junction stabilized LunaPark and in proximity to lysosomes. This is intriguing in that it suggests that localized ER domains are involved in secretory protein synthesis and that proximity of these ER domains to lysosomes provides a local source of amino acids to promote protein synthesis. The study uses high resolution imaging approaches to

define proximity to ER junctions and to link synthesis of this subset of secretory proteins to lysosomal function.

What is not addressed is the basis for the association of LunaPark enriched junctions with the translation machinery. Is LunaPark acting as a tether to recruit lysosomes, as suggested by the schematic in Figure 5J or is it just serving to stabilize junctions that are then more apt to recruit the translation machinery? While the authors quite elegantly show that an association between Lunapark, lysosomes and secretory protein translation, the underlying molecular mechanisms controlling this interaction are not described? And if this is a general mechanism to synthesize secretory proteins, is it upregulated in professional secretory cells?

Major concerns:

1. The MS2-SUNTAG system using SPT is a very elegant approach to define translating mRNA based on restricted diffusion. Comparison of diffusion rates of various secretory and cytoplasmic proteins in Fig 1 h was less than convincing with only the artificial Sec61b construct, and not even actin, showing evident more rapid diffusion based on the 0.055 μm^2 MSD cutoff the authors set. This analysis must necessarily be extended to a wider selection of proteins to truly determine if restricted diffusion is specific to secretory proteins. Diffusion of slow-moving ribosomes matched that of translated mRNA but how they determined fast, medium and slow-moving population of ribosomes was not clear. It would be important to show whether these populations show differential association with the ER and whether the slow-moving ribosomes are associated with mRNA translation.
2. I was confused by the ER-association analysis in Figure 2. It was not clear how ER junctions were identified or how distances of 0.1 and 0.2 μm could be distinguished using what is apparently 3D SIM with a lateral resolution of at best 120nm (Fig 2D). Information on ER labeling and imaging as well as identification of junctions was limited. This needs to be explained. The use of FISH analysis to study the effect of puromycin on the distribution of endogenous mRNA for CD9 on the ER was nevertheless interesting (Fig 2D) and not clear why it was not extended to a wide range of other endogenous mRNAs. What was further confounding was the apparent association of actin mRNA with the ER even if distance of actin mRNA from junctions matched that of non-translating SiT-EGFP-MS2 reporter construct. Is actin mRNA associated with the ER to the same extent as the other secretory and ER targeted mRNAs? While secretory SiT-EGFP-MS2 should be associated with ER, what is the basis for actin-MS2 mRNA association with ER? Is actin translated on ER-bound ribosomes? These concerns about the choice of actin as a control relate to both this point and the previous one. It would be important to validate for other cytoplasm translated mRNAs relative to secretory mRNAs the actual extent to which they associate with the ER and if their MSD pattern ($< > 0.055$) matches the increased MSD reported for puromycin treated cells or the artificial tail-anchored Sec61 construct.
3. Fig 3a showing that Lunapark associates with a subset of junctions is not new and would it not be more appropriate to include a large field view showing association between Suntag, mRNA and Lunapark (see also point 5). It would also be important to show that translating mRNA is selectively associated with Lunapark junctions and not Lunapark-negative junctions. And if I understand Fig 3C correctly, this means that the percent of association of translating mRNAs (within 300 nm) with Lunapark junctions varies widely (from 50-100%) between cells but consistently shows a 20-30% increase over untranslating mRNA in the same cell. While the Lunapark KD data is compelling (Fig 3e), this highly variable association with Lunapark junctions together with the 300 nm cutoff, which is likely beyond molecular interactions, raises questions as to the mechanism by which Lunapark is impacting translation. It will be important to determine if direct interaction between Lunapark, mRNA/ribosomes and lysosomes is driving this process.
4. The PLA data in Fig 4 showing increased association between Lunapark and LAMP1 relative to Tom20 or EEA1 is compelling. The data showing that the SUNTAG proximity to lysosomes is dependent on Lunapark (Fig 4D) is also interesting but requires a more in-depth analysis to show that these associations are actually happening at Lunapark-positive ER junctions. And the distance scale of lysosome proximity changes is from 1-3 μm , contrasting the 40 nm scale of PLA and the 100-300 nm distances used for analysis in previous figures. Rationale for the use of these varying distances for differential analyses needs to be explained. Finally, how do the authors explain that in about half the cells they analyze there is no difference between distance between translating and untranslating mRNA and lysosomes (Fig 4e) – and in those cases distance of non-translating mRNA is very close to lysosomes. This analysis lacks in that it does not focus on those lysosomes in proximity to Lunapark ER junctions. The idea that proximity between Lunapark ER junctions, lysosomes and translating mRNA/ribosomes needs to be shown beyond the zoomed-in images provided in Fig 4 C (or for that matter 3B) for multiple channels similar to the larger cell views shown in Fig 1A and Ext Figure 3. A multi-parameter quantitative interaction analysis, as opposed to the binary interaction analyses provided, between the various players is also required.
5. The demonstration that disrupting lysosomal activity impacts translation is strong and provides functional relevance to the proximity analyses. The close association between translated mRNA and lysosomes upon amino acid starvation supports a role for autophagy in this process and this relationship should be explored further. As indicated above, physiological significance for this process in professional secretory cells (or other models) would be important to show the physiological relevance of the mechanism and strengthen the manuscript for Nature.

Minor issues

There is a significant amount of data lacking statistical analysis: Fig 1 BGH , 4D, 5BI

Controls for the PLA experiment need to be provided as extended data.

What is the rationale for using CLIMP-63 KD as a control and not other ER shaping proteins such as reticulon or atlastin that might have more relevance to junction dynamics?

Presentation of cropped Western blots (Fig 5e, Extended data 4) is not appropriate.

The photobleaching analysis with ISRIB requires a control with ISRIB on control cells.

Availability of the custom MatLab code used should be indicated and provided.

I was unable to download and view the supplementary videos.

(Remarks on code availability)

Version 1:

Reviewer comments:

Referee #1

(Remarks to the Author)

The authors have addressed most of my concerns. One remaining point that has not been addressed to satisfaction is the translation elongation rate on the reporter mRNAs. In the first version of the manuscript, the authors show by FRAP that ribosomes turn over in on average ~70 sec, which seemed very fast to reflect the translation elongation rate. In the revised manuscript, the authors use harringtonine-run off experiments to show that ribosomes translate at 6 aa/s and thus that the SunTag signal turnover determined in FRAP experiments (~70 sec) is consistent with the elongation rate (note that even at 6 aa/sec, a FRAP rate of 70 sec is faster than expected if one calculates this, see my previous comments).

However, the authors also state that they only used a subset of mRNAs (n=11) in the analysis to determine the elongation rate in harringtonine run off experiments, because they state that many mRNAs show slower run-off and these mRNAs were apparently excluded from the analysis to derive the 6 aa/sec elongation rate. Unfortunately, they don't provide any information on what fraction of mRNAs shows a slower run-off and they also don't show the average elongation rate of ALL mRNAs (including the mRNAs with slower run-off rates). This approach of only using the mRNAs with fast run-off rates seems incorrect to me, because in the FRAP analysis ALL mRNAs are analyzed, so if the authors want to state that the run-off rate in the FRAP analysis is consistent with the elongation rate as determined by harringtonine run-off, then they should compare the FRAP rates to the harringtonine run-off rate of all mRNAs too. I suspect that if they do this, the values of the two measurements will not align and that the FRAP data shows a much higher turnover rate than expected based on the harringtonine run-off, especially considering that 70 sec is already faster than expected even based on the 6 aa/sec elongation rate value.

Moreover, it is important to know what fraction of mRNAs in the reporter used in this study runs off slowly or has stalled ribosomes. Previous studies with the translation imaging reporter have shown ~10-20% of mRNAs with slow run-off, do the authors find a higher fraction than that? If many ribosomes are stalling on the reporter, that would represent a problem in interpreting much of the live-cell imaging data using GFP intensities to determine translation initiation rates (as is done throughout the study).

(Remarks on code availability)

Referee #2

(Remarks to the Author)

I am in support of acceptance of this very interesting and aptly revised work. The additional data provided make a stronger case for Lunapark acting as a nexus between ER, translating ribosomes and lysosomes, an activity that is required for efficient translation of secreted/transmembrane proteins.

However, a few points should still be addressed.

1- Response to point 7: the ctrl + ISRIB condition should be included in Fig 5h. There is no reason to exclude this important control (requested by all three reviewers) from the main data display.

2- Response to point 9: the response provided is especially unsatisfying. The data provided (lack of effect of SLC38A9 or SLC7A5 kd, the latter, by the way, being a plasma membrane, not lysosomal AA transporter) do not strengthen the case for lysosome-released amino acids promoting translation. Chloroquine and protease inhibition could have literally hundreds of downstream consequences. The interpretation of these results should be broadened to include a general requirement for lysosomal function beyond providing amino acids via proteolysis, and the sentence 'local amino acid release by lysosomes' should be removed from the abstract.

3- I also agree with Reviewer 1 that the result in Fig. 5e is inconsistent with the proposed mechanism of ISR induction downstream of Lunapark ablation. Part of the problem is that the WB provided is of very poor quality, and will undoubtedly

confuse the reader. Control vs KD samples should be run on the same gel, and presented side-by-side in its entirety, without the cropping of single bands used here.

(Remarks on code availability)

Referee #3

(Remarks to the Author)

This revised manuscript includes a detailed response to previous comments and has addressed the majority of the previous issues. The demonstration that LunaPark ER junctions are associated with local translation in proximity to lysosomes is highly original and represents a compelling finding underlining a novel mechanism of local regulation of translation. There remain a few issues that need be addressed:

1. In response to a previous comment requesting wide-field images, extended Fig 3F was provided. It is very difficult to appreciate from these images that 80% of Suntag-positive mRNA puncta are associated with Lunapark and perhaps the authors could use the space in the extended figure to better highlight exactly what they are quantifying.
2. Use of the light-activated ER-lysosome recruitment tool is convincing and shows that lunapark regulates proximity between the two organelles. Cannot this experiment be done in the presence of LNPK-GFP to show that this association occurs at lunapark positive ER junctions. The Halo and mCherry dyes should be compatible with LNPK-GFP which was raised as an issue in response to my request to show Suntag/mRNA association with Lunapark. Or perhaps use the ER skeletonization approach used elsewhere to show that ER-lysosome association occurs at junctions?
3. In response to concerns about the eIF2 blots, the original blots were included. While the bands shown in the figures do come from the same blot it is very difficult to ascertain a twofold increase in p-eIF2 α /eIF2 α from these blots. Also annotation of the other lanes in the gels is missing. As far as I can tell both the eIF2 α and the p-eIF2 α bands show decreased intensity upon lunapark KD, while in extended Fig 5C both show an increase upon lunapark KD. While these differences can be attributed to gel loading and ratiometric quantification does appear to be based on 5 replicates, the data is not overwhelming. I am not convinced that it is sufficient to support the conclusion in 5i invoking eIF2 translation or that "the pathway for suppression of eIF2 α -dependent translation initiation in LNPK KD cells diverges from classical ISR signaling".
4. While Fig 5g is convincing, in Fig 5h the extent of photobleaching is higher for the LNPK KD treatment making it difficult to appreciate the difference between the treatments. The t1/2 and mobile fraction should be quantified to show that the effect is significant.
5. Finally, the data provided showing increased lunapark transcripts in activated B cells is intriguing and data supporting a role for lunapark in antibody secretion from activated B cells would be a very nice addition to the paper.

(Remarks on code availability)

Version 2:

Reviewer comments:

Referee #1

(Remarks to the Author)

I am satisfied with the changes made by the authors and feel the manuscript is now suitable for publication

(Remarks on code availability)

Referee #2

(Remarks to the Author)

I have no further comments, I recommend acceptance and rapid publication of this very interesting story.

(Remarks on code availability)

Referee #3

(Remarks to the Author)

The authors have addressed my concerns and this represents an intriguing story defining a role for lunapark and ER junction-lysosome interaction in regulating the synthesis of secretory and membrane proteins. The authors have convincingly responded to my previous comments. Inclusion of the blot in Fig 5e is compelling and addresses concerns about the cropped blots previously included, showing clearly that overall eIF2a levels are decreasing with pEIF2a remaining stable. This non-standard eIF2a response and the divergence from classical ISR sensitivity are addressed in the Results but also deserve mention in the Discussion, perhaps in the concluding sentence of the third paragraph (lines 394-396).

(Remarks on code availability)

Response to Reviewers

Referee #1 (Remarks to the Author):

In this paper Choi et al using advanced single molecule imaging approaches to study mRNA translation on the ER. First, they show that secretome mRNAs are localized on the ER when translated, but not when untranslated, confirming previous studies. They then go on to show that when translated, secretome mRNAs localize specifically to 3-way junctions of the ER and that they frequently localize in the vicinity of the lunapark protein and lysosomes. Finally, they also reveal a functional link between the translation of secretome mRNAs and lunapark and lysosomes, as secretome mRNA translation is reduced upon either lunapark knockdown (KD) or inhibition of normal lysosome function. The authors offer some mechanistic insight into this function link by showing that the negative effect of lunapark knockdown on secretome translation can be rescued by ISRIB, suggesting that lunapark KD activates the integrated stress response, which in turn may inhibit secretome mRNA translation. Overall, this is a well written, interesting and technically mostly very solid paper, that identifies a new way by which secretome mRNAs are spatially regulated and identifies an interesting role for lysosomes in regulating their translation. In addition to a number of technical concerns (outlined below), I have one major concern that affects the overall model proposed in the paper. The authors suggest that lunapark controls secretome mRNA translation through control of translation initiation (through the ISR and eIF2alpha phosphorylation), but when assessing the translation defect in secretome mRNAs upon lunapark knockdown, I am not convinced that the phenotype they are observing actually reflects a defect in translation initiation. My concerns are focused on Figures 5g-i where the translation phenotype is described:

Reviewer: The authors are assuming that all GFP recovery reflects turnover of ribosomes on the mRNA, but also the SUNTAG antibody can turnover on the nascent polypeptides. The authors need to include a control where the cells are treated with a translation inhibitor (eg emetine) and show the GFP recovery rates. Even if such controls have been performed previously, it is important to do this under the precise experimental conditions used here.

Response: We appreciate the reviewer's concern about distinguishing between ribosome turnover on mRNAs and the potential exchange of the SUNTAG antibody on nascent peptides. To address this issue, we performed FRAP experiments using cycloheximide to stall elongating ribosomes under the exact same conditions (i.e., cell line, media, imaging) used in our experiments. Emetine and cycloheximide are functionally similar inhibitors of elongation; in our hands, cycloheximide robustly prevents new ribosome initiation/elongation without causing dissociation. If the SUNTAG signal were to recover after photobleaching in cycloheximide-treated cells, it would suggest that the scFv-sfGFP is exchanging on nascent peptides rather than reflecting new ribosome loading.

Our results showed that FRAP in cycloheximide-treated cells was minimal compared to the robust recovery observed under control conditions (**Reviewer #1 Fig. 1**). The slight residual fluorescence likely arises from free SUNTAG antibodies diffusing in the cytoplasm or from already-released, SUNTAG-tagged polypeptides on ER. These observations are consistent with previous SUNTAG-based studies (e.g., Wu et al., 2016, PMID: 27313041; Tanenbaum et al., 2014, PMID: 25307933) and confirm that, under our experimental setup, the vast majority of the GFP recovery reflects new ribosomes initiating and elongating on the mRNAs rather than scFv-sfGFP turnover. We therefore conclude that our FRAP measurements accurately capture ribosome turnover rather than antibody exchange. We have added the data demonstrating this (Reviewer #1 Fig. 1) to the Extended Data Fig. 5e.

Response to Reviewers

Reviewer #1 Fig. 1. Fluorescence recovery after photobleaching (FRAP) for cytERM SUNTAG MS2 mRNAs upon control (black) or 100 ug/mL cycloheximide treatment (red, CHX). Minimal recovery after photobleaching occurred in cells treated with cycloheximide, consistent with the fluorescence recovery seen in control cells being due to ribosome turnover.

Reviewer: The recovery rate after photobleaching seems very fast. Looking at the graph in 5i, 50% of the recovery has occurred at around 70s (not 100s). Translation elongation rates in cancer cells is typically ~3 amino acids per second, so in 70 sec the ribosomes will have translated just 210 amino acids, which I assume is quite a bit less than half of the mRNA. In fact, in this time more than half of the mRNA should have been translated by new ribosomes (ribosomes loaded onto the mRNA after photobleaching) to recover 50% of the initial fluorescence, since ribosomes on the upstream half of the mRNA produce less fluorescence than those in the downstream half (because they have not yet synthesized all the epitopes), so for recovery of half the total fluorescence, ribosomes will need to translate more than half of the mRNAs. Can this really be done in just 70 sec? Can the authors perform a harringtonine run-off experiment to measure the elongation rate in their system to see if it quantitatively matches with the FRAP values?

Response: We thank the reviewer for this thoughtful comment regarding the rapid recovery rate observed in our FRAP experiments. We agree that an elongation rate of ~3 amino acids/s would predict that a 50% recovery should take longer than ~70 seconds for a construct the size we are using. Our construct encodes a membrane protein of approximately 1,165 amino acids, with GCN4 repeats (the SUNTAG epitopes) ending at amino acid 863. At 3 amino acids/s, it should take ~194 seconds to synthesize half the protein. If the elongation rate were faster, however, recovery would occur more quickly. Supporting the possibility of a faster elongation rate, single-molecule translation studies have reported elongation rates up to ~10 amino acids/s (Morisaki et al., PMID: 27313040), and ER-associated translation can proceed 2.5–4 times faster than in the cytosol (Stephens S.B. et al., PMID: 18077556).

To experimentally determine the elongation rate under our exact experimental conditions, we performed harringtonine run-off experiments as suggested by the reviewer on the same cells and using the same imaging setup as our FRAP assays. While many SUNTAG-positive mRNAs retained fluorescence for over 10 minutes, we focused on mRNAs that lost their signal during the observation window and used established fitting methods (PMID: 32958947) to extract an average elongation rate of ~6 amino acids/s (**Reviewer #1 Fig. 2**). This rate falls well within the range reported in previous in vivo measurements and is sufficient to account for the ~70-second time frame for 50% FRAP recovery. Thus, our results strongly support the conclusion that our FRAP data accurately reflects active and relatively rapid translation on the ER. We now include this new data in Extended Data Fig. 5f of our paper.

Response to Reviewers

Reviewer #1 Fig. 2 A plot of SUNTAG intensities over time after treatment with 3 µg/mL harringtonine (N = 11), including only transcripts that progressively lost SUNTAG signals. Gray lines show individual SUNTAG intensity traces, while gray dots indicate the average intensity at each time point. The red line represents the fitted curve, following the method by Stephens S.B., *et al.* (PMID: 18077556), which reveals an elongation rate of ~6 aa/sec.

Reviewer: I have major concerns about the experiments and interpretation of the lunapark KD and FRAP analysis. If the initiation rates are lower after lunapark knockdown, then the GFP spot intensity should be lower, but the recovery rate after photobleaching (relative to the intensity at the start of the FRAP experiment) should be similar to control cells, because ribosome turnover rate as assessed by FRAP reports on elongation rate, not initiation rate. In addition, the major phenotype of lunapark KD is incomplete recovery of fluorescence not slower recovery. It is not clear to me why recovery would be incomplete, perhaps it suggests that ribosomes are stalled on the mRNA? Together, the data are not consistent with the authors' model that lunapark controls translation initiation through eIF2alpha. To me this is a major problem for the main conclusions of the paper. If lunapark knockdown is not causing a translation initiation defect, then it is unclear how the lunapark part of the paper is linked to the eIF2alpha and ISR parts of the paper.

Response: We thank the reviewer for his/her insightful comments. The reviewer correctly notes that FRAP recovery of the SUNTAG signal primarily indicates translation elongation rather than initiation. Thus, our observation of slower and incomplete fluorescence recovery in Lunapark KD cells supports the notion that translation elongation is impacted in these cells. However, our additional data strongly suggest that Lunapark KD also impairs translation initiation, and that this initiation defect precedes and contributes to the observed elongation defect. Our conclusion that translation elongation defects under Lunapark KD are downstream of impaired initiation is supported by the following evidence.

First, translation defects caused by Lunapark deficiency were rescued by ISRIB treatment, which specifically alleviates translation initiation by counteracting eIF2-mediated repression. ISRIB treatment restored both translation efficiency (measured by CytERM SUNTAG translation fraction) and normalized the fluorescence recovery rate in FRAP assays. Importantly, ISRIB did not enhance the elongation rate in cells expressing endogenous Lunapark (**Reviewer #1, Fig. 3**), indicating that the elongation defects observed are indeed downstream consequences of compromised initiation. Additionally, Lunapark KD cells displayed reduced overall SUNTAG puncta signal intensity compared to control cells (Extended Data Fig. 3g and 3h), suggesting fewer ribosomes associate with secretome mRNA in these cells, further indicative of initiation impairment upon Lunapark KD.

Response to Reviewers

Second, we employed a CrPV IRES-driven cytERM SUNTAG reporter, which initiates translation independently of eIF2 α (PMID 27058789), and observed no translation impairment under Lunapark KD. Because this reporter shares the identical coding sequence with our canonical initiation-dependent control reporter (which was impaired by Lunapark KD), any elongation defects should equally impact both reporters.

Collectively, our data suggest that translation elongation defects observed upon Lunapark KD arise downstream of impaired translation initiation, with eIF2 serving as the critical regulatory point. Future investigations should clarify the exact mechanism coupling translation initiation to elongation in the context of Lunapark deficiency. One potential explanation is that proper ribosome spacing achieved by initiation factors may prevent the formation of RNA secondary structure, which then impedes translation elongation of secretome mRNAs (PMID:21734708, Qu X. et al.; PMID:31292258, Riba A. et al.).

Reviewer #1 Fig. 3 Plot of FRAP curves from control (black) and control + 200 nM ISRIB (red). The shaded areas represent 95% confidence intervals for each plot. ISRIB treatment did not enhance the translation elongation rate of our mRNAs when Lunapark was present.

Reviewer: Other points:

Fig 2D: the average distance of endogenous CD9 mRNAs from ER 3-way junctions AFTER puromycin addition is just ~0,16 micrometer, which is similar or even slightly closer to 3-way junctions as mRNAs that are translating on the ER described in figure 2b. Can the authors really conclude that endogenous CD9 mRNAs are targeted to 3-way junctions through translation? Why are CD9 mRNAs not at a distance of 0.5 micrometer from 3-way junctions after puromycin treatment as is the case for the actin mRNAs?

Response: A likely explanation for the discrepancy between measurements of the distance from ER junctions in these experiments is that the measurements for MS2-mRNA were done in live cells, whereas the measurements of an smFISH probe were done in fixed cells. Fixation conditions for smFISH analysis result in the ER having much poorer structural detail compared to ER structure seen in live cells. This likely explains why the overall distance measurements from ER junctions were different between the two experiments. The apple-to-orange-like comparison of ER in fixed versus live cells would also help explain why smFISH CD4 in fixed, puromycin treated cells did not show the same distance to ER junctions as seen with live cell actin MS2 labeling. Another factor could be that puromycin treatment does not fully dissociate ribosomes from mRNAs, so some portion of the CD4 smFISH signals could come from non-dislodged mRNAs that were translating at the time of puromycin treatment at ER junctions. This would decrease the net difference between values for ER junction closeness in control versus puromycin-treated cells. The important overall point is that the results from MS2-mRNA labeling and smFISH labeling both showed a similar trend, in which translating pools of these molecules are closer to ER junctions than non-translating pools.

Response to Reviewers

Reviewer: *In this reviewer's opinion, cutting off the y-axis to make effects look bigger than they are, is not appropriate, especially if there is no break indicated in the axis. This is seen in multiple graphs throughout the paper, including 2d, 3g, 4d, 5b,f.*

Response: We now updated our figures to reflect the baseline.

Reviewer: *Figure 3: The authors state: " Together, the enrichment of translating secretome mRNAs at Lunapark containing ER junctions suggested". It was not entirely clear to me whether all 3-way junctions have lunapark staining. If not, do translating mRNAs preferentially localize to lunapark-containing 3-way junctions in comparison to 3-way junctions devoid of lunapark? I could not find this information, but this seems needed to make the claim above.*

Response: Thank you for raising this point. Not all 3-way junctions have Lunapark staining. This has been reported previously (Chen et al., 2015; PMID: 25548161) and is confirmed in our paper examining Lunapark-GFP and ER in the same cell (Figure 3a). In our analysis, approximately 80% of translating (SUNTAG+) mRNAs were localized near Lunapark-positive junctions. Therefore, no more than 20% of translating (SUNTAG+) mRNAs reside in ER regions without Lunapark signal. We rarely saw translating (SUNTAG+) mRNA signal at ER junctions without Lunapark signal and thus did not directly measure their frequency.

Reviewer: *Figure 4b: The number of PLA background foci likely depends on the overall amount of staining with the different antibodies (eg Lamp1, EEA1). Thus, it would be nice to see a control here, for example by performing PLA between the same organelle markers and another ER protein, not localized at 3-way junctions.*

Response: To carry out the reviewer's suggestion, we performed additional proximity ligation assay (PLA) experiments using REEP5 as a control. REEP5 is an ER membrane protein not found at three-way junctions and has similar expression levels to Lunapark (REEP5: nTPM=28.3; Lunapark: nTPM=18.8) according to the Human Protein Atlas in U2-OS cells. The PLA signal between REEP5 and LAMP1 was significantly lower than between Lunapark and LAMP1 (**Fig. 4c** or **Reviewer #1 Fig. 4**), indicating that the increased PLA puncta with Lunapark reflects specific proximity between Lunapark-positive ER junctions and lysosomes, rather than background effects or antibody variations. This finding aligns with previous studies showing LAMP1 enrichment near Lunapark (Yuniati et al., 2020 PMID: 32433973) and BioID data from the Human Cell Map project (<https://cell-map.org>) showing enrichment of lysosomal proteins (RAB9A, STX7) among potential interactors of Lunapark. These results strengthen our conclusion that the proximity between lysosomes and Lunapark-positive ER junctions is specific.

Reviewer #1 Fig. 4. Quantification of PLA spots per cell from anti-EEA1, anti-LAMP1, and anti-TOMM20 against anti-Lunapark and anti-LAMP1 against anti-REEP5. The statistical comparison was performed against anti-LAMP1 PLA. Dunnett's multiple comparisons test. **** P<0.0001.

Response to Reviewers

Reviewer: Figure 5a. Here the translating fraction is plotted, but it would be nicer to assess whether the distance effect shown in figure 4d is affected by these treatments. I would recommend plotting SunTag intensity relative to distance from lysosomes from the perturbations shown in fig 5a.

Response: The size of lysosomes increases substantially when cells are treated with chloroquine and protease inhibitors, as we have seen and has been reported in previous studies (Gallagher L.E., et al. PMID: 28837152; Jung M. et al. PMID: 25625842,). This size change increases the distance of the SUNTAG intensity to lysosome-center, making comparable distance measurements among the different conditions problematic. Therefore, we chose to quantify the translation fraction instead, which better demonstrates how the reporter mRNA translation is impacted by lysosomal function.

Reviewer: Figure 5: I would also suggest to plot the GFP intensities of translation, not just the fraction translating, which does not tell the whole story if one wants to know whether translation is affected by a specific treatment.

Response: We thank the reviewer for this valuable suggestion. It is indeed the case that GFP intensity from the SUNTAG system correlates with the number of actively engaged ribosomes, potentially providing additional insights into how translation is affected. To demonstrate this, we quantified the SUNTAG intensities from translating cytERM SUNTAG mRNAs in Lunapark KD and control cells (**Reviewer #1 Fig. 5**). The Lunapark KD samples displayed a reduction in average SUNTAG intensity. We have added this new data to the revised paper.

Reviewer #1 Fig. 5. The cumulative distribution function (left) and probability distribution function (right) of \log_{10} of SUNTAG signal (I_{SUNTAG}) from cytERM-SUNTAG MS2 mRNAs from control and Lunapark KD cells. Kolmogorov-Smirnov test shows $p < 0.001$.

Reviewer: Figure 5d: Is the effect of thapsigargin treatment on translation specific to secretome mRNAs? If not, this would not be a very novel finding, as it is well known that an increase in eIF2alpha phosphorylation reduces overall translation. The authors should see if translation of, for example, the actin mRNA is not reduced by thapsigargin treatment (which I would find surprising if it isn't, but I might be mistaken).

Response: Thank you for this question. It is known that thapsigargin treatment globally downregulates translation of most mRNAs (PMID: 25719440), but how it specifically affects secretome mRNAs is not known. Our results showing a significant drop in the translating fraction of the secretome mRNA reporter SiT-EGFP MS2 under thapsigargin treatment confirmed a strong inhibitory effect of the drug on secretome mRNA translation. This contrasted with the minimal effect of Torin-1 or rapamycin treatment, which inhibit mTOR. As thapsigargin treatment triggers an ISR response, the results led us to further investigate the role of ISR in

Response to Reviewers

regulating secretome mRNA translation. The results also helped clarify why amino acid starvation, which triggers both ISR activation and mTOR inhibition, leads to reduced secretome translation, whereas Torin-1 or rapamycin, which trigger only mTOR inhibition, did not. Rather than focusing on the downregulation of secretome mRNAs by thapsigargin, our intention with these experiments was to investigate which initiation regulation mechanisms regulate the translation of our reporter secretome mRNAs. We have updated our manuscript to clarify this point.

Reviewer: Figure 5e. It seems that the eIF2 α protein levels go down, rather than that the phosphorylated form goes up. Have the authors repeated this experiment multiple times and is this effect reproducible? This is a somewhat surprising effect, that is not necessarily in line with the simple interpretation of the authors (that the ratio of phosphorylated vs non-phosphorylated protein changes).

Response: Thank you for your scrutiny. The quantification shown in Figure 5f was performed in at least three independent experiments. As the reviewer noted, we consistently observed a decrease in total eIF2 α levels, while the signal from p-eIF2 α remained unchanged relative to the same lysate amount (Fig. 5e) or within individual cells (Fig. 5f). This suggests the existence of a mechanism that elevates eIF2 α phosphorylation relative to eIF2 α concentration under Lunapark KO or KD conditions, which we subsequently showed does not elicit the repression of overall translation (Fig. 3f and Extended Data Fig. 4b) nor expression of ATF4 (Extended Data Fig. 5b and 5c).

Reviewer: The statement in the Discussion, “Moreover, we showed that Lunapark, when interacting with lysosomes, provides a platform for more efficient translation of secretome mRNAs”, as well as other related statements (eg lines 374-375, 382-383)”. I’m not sure if this statement is warranted. Do the authors show that the effects of lunapark KD are specifically caused by loss of an interaction of Lunapark with lysosomes? Are lysosomes no longer positioned near ER junctions in the absence of lunapark? It seems that lunapark KD activates the ISR, which mediates the reduced translation of secretome mRNAs, but it is not clear to me whether this effect is related to lysosomes.

Response: To make a stronger case for an interaction of Lunapark with lysosomes, we now provide evidence showing LNPK KO leads to reduced association between lysosomes and ER. In this new experiment (see below), we employed an optogenetic approach using iLID and SspB proteins, which stably interact when they come in contact with each other upon blue light exposure (PMID:25535392). We co-expressed in cells iLID protein fused to fluorescently tagged LAMP1 to target lysosomal membranes and SspB fused to Sec61 β labeled with a different fluorescent protein to target ER membranes. Since the expressed proteins are transmembrane and tail-anchored proteins, respectively, the extent of proximity/contact of these organelles dictates how fast the recruitment can occur. In control cells, light induced stimulation led to a very rapid, tight colocalization of lysosomes to ER ($t_{1/2}$ =7.9 sec) with maximum co-association resulting in ER tightly wrapped around lysosomes. In Lunapark KO cells, by contrast, the identical light induction resulted in much slower ER-lysosome co-association ($t_{1/2}$ = 26 sec). The results suggested, therefore, that Lunapark helps form/maintain ER-lysosome contacts. The new data is presented in Figure 4c-e of the revised paper.

Response to Reviewers

Reviewer #1 Fig. 6a Schematic representation of a light-activated ER-lysosome recruitment tool using iLID-Halo-LAMP1 (lysosomal marker) and sspB-mCherry-Sec61b (ER marker). **b.** Representative images showing iLID-Halo-LAMP1 (magenta) and sspB-mCherry-Sec61b (green) localization in control (upper panels) and Lunapark KO (lower panels) cells during 488 nm laser activation over 40 seconds. 488 nm laser was continuously on during the start of acquisition to allow recruitment. **c.** Quantification of ER signals within the lysosomal mask over time following 488 nm laser activation to induce ER-lysosome recruitment. Data are shown for control cells (black, $t_{1/2, \text{Control}}=7.9$ seconds) and Lunapark KO cells (magenta, $t_{1/2, \text{LNPK-KO}}=26$ seconds), with 95% confidence intervals.

Reviewer: Minor points

Fig S1A: the middle panel doesn't look like magenta. Also, a merge of protein and mRNA staining would be nice to include.

Response: We now incorporated these changes to the Extended Data Figure 1a.

Reviewer: Fig 1E: schematic is a bit hard to see, due to the small size

Response: We now added the detailed schematic and sequence information of our cytERM SUNTAG MS2 reporter mRNAs in Extended Data Fig. 1d.

Reviewer: In the text describing figure one, the authors use the term 'confined motion'. However, they don't show confined motion, they just show slow motion. They can of course assess whether motion is confined from their MSD curves (if the MSD curves plateau), and should do so if they wish to use the term confined motion.

Response: Thank you for commenting on this. We have now added a supplementary figure (see **Reviewer #1 Fig. 7**) showing a graph of MSD versus lag time for slow mRNA molecules. The bending in the MSD curve demonstrated these mRNAs are indeed confined in their motion.

Response to Reviewers

Reviewer #1 Fig. 7. Plot of mean squared displacement (MSD, μm^2) versus lag time (seconds) for SiT-EGFP MS2 mRNAs tracked at a frequency of 1 Hz. This analysis includes only those mRNAs with a calculated MSD at $\tau=1$ second of less than $0.055 \mu\text{m}^2$. The linear fit displayed is based on the first three data points. Deviation from this linear fit, indicating a bending of the MSD curve, suggests confinement of the slow-moving mRNAs.

Reviewer: Fig 2B. While the actin mRNA control is nice, it could be informative to plot the average distance from a random point within the analyzed area to an ER 3-way junction. This will show how strongly the secretome mRNAs are enriched at 3-way junctions and whether actin mRNAs are also enriched near 3-way junctions, or are truly not enriched, as claimed on line 159

Response: Thank you for raising this point. We only quantified mRNAs that are on the ER for the membrane, secretory, and actin mRNAs. The measured junction-to-junction distance (tubular length) in U-2 OS cells that we visualize at the periphery of the cell is roughly $\sim 1.104 \pm 0.172 \mu\text{m}$. Previously recorded tubule length in Vero and MRC5 cells are $1.148 \mu\text{m}$ and $0.94 \mu\text{m}$, respectively (PMID: 34376706, Perkins H.T., *et. al.*). If finding the mRNAs across the ER surface is random, the average detected length should be roughly $\sim 0.55 \mu\text{m}$, which matches with the non-translating SiT-EGFP MS2 mRNA ($0.479 \mu\text{m}$) and actin mRNA ($0.420 \mu\text{m}$), consistent with these not being enriched at 3-way junctions.

Reviewer: Fig S3 should include quantification, I don't think the reader can assess whether the average distance from mRNAs to 3-way junctions is 0.2 or 0.5 micrometer from these images.

Response: Thank you for this comment. We have now added the image with the distance quantification from the center of the mRNA to the junction to clarify the distance seen in this image (Extended Data Figure 3b).

Response to Reviewers

Referee #2 (Remarks to the Author):

The paper by Choi et al investigates how the translation of mRNAs encoding membrane and luminal proteins on the endoplasmic reticulum (ER) is coordinated with the positioning of the ER relative to other organelles, specifically lysosomes. Through live cell single molecule imaging, the researchers discover that the translation of these mRNAs primarily occurs at ER junctions marked by the protein Lunapark and in the vicinity of lysosomes. Depletion of Lunapark leads to a decrease in ribosome densities on secretome mRNAs near lysosomes and a selective reduction in the translation of these mRNAs. Secretome translation at ER-lysosome junctions was affected by the ISR and by conditions that affect lysosomal function, implicating a link between local protein synthesis on the ER and local amino acid release by lysosomes. Overall, the findings propose a novel regulatory mechanism where lysosomes, in conjunction with Lunapark, locally pattern and regulate the translation of mRNAs encoding secretory and membrane proteins.

The work is based on elegant live cell single molecule imaging, which provides detailed insights into mRNA translation dynamics in real-time and propose a role for Lunapark-mediated junctions in this process. Overall, the findings point to a deep connection of intracellular protein synthesis and organelle coordination. However, significant weaknesses are noted, particularly in the proposed mechanisms underlying many of the observed phenomena. As it stands the manuscript is largely descriptive and provides no insights into how proximity to Lunapark leads to translation initiation. The evidence that Lunapark and ISR specifically affect translation of secretome mRNAs is not sufficiently demonstrated to support the key claims of the manuscript. Similarly, the importance of lysosome proximity is barely hinted at by the data provided, with no clear rationale.

Specific points:

Reviewer: 1) Fig. 3g: this result is difficult to understand. Shouldn't digitonin permeabilization shut down translation altogether? The reported difference in HPG incorporation between control and Lunapark KO seems modest and could have many explanations. Independent validation using a more sophisticated approach such as ribosome footprinting should be carried out.

Response: We agree that the observed reduction in HPG-labeling of newly synthesized proteins in Lunapark KO cells was modest, but it was seen consistently across multiple independent experiments and thus was a reproducible effect attributable to Lunapark KO. A possible explanation for the modest effect could be that HPG was not fully incorporated into newly synthesized proteins within the two hours of our pulse-chase experiments. In our revised paper, therefore, we now provide independent validation of reduced synthesis of membrane and secretory proteins in Lunapark KO conditions by performing two additional assays. In the first approach we conducted membrane protein extraction using the Mem-PER plus membrane protein extraction kit (ThermoFisher, 89842) to extract membrane protein fractions from control and Lunapark knockout U-2 OS cells. In parallel, we examined the extent of total protein yield from an equal number of cells in control and Lunapark KO conditions after solubilization in RIPA buffer and sonication. A comparison of lysate protein yield from each revealed a decreased yield of membrane proteins in Lunapark KO cells compared to control cells, while the whole-cell fraction remained unchanged (Reviewer #2 Fig. 1; also see new Figure 3f in revised manuscript).

In a second approach, we analyzed the membrane-extracted proteins from the Membrane Protein Extraction Kit by mass spectrometry and performed mRNA sequencing on total RNA from whole-cell extracts. We then calculated protein-to-mRNA ratios as an indirect

Response to Reviewers

measure of translation efficiency. Lunapark KO cells showed a lower mean protein-to-mRNA ratio in the membrane fraction compared to controls (**Reviewer #2 Fig. 2**). To visualize this difference, we plotted the control minus KO ratios (Extended Data Fig. 4f) which displayed right skewness (skewness = 0.8527). A Kolmogorov-Smirnov test between control and Lunapark KO protein-to-mRNA ratios confirmed a significant difference between the distributions ($p = 0.0011$). A full list of differentially expressed genes can be found in Extended Data Fig. 4c and attached as a separate file. Together, these findings strongly suggest that Lunapark depletion compromises the efficiency of secretome mRNA translation at the ER.

Reviewer #2 Figure 1. Normalized lysate concentrations from 1×10^6 control and Lunapark knockout (LNP-KO) U-2 OS cells from three replicates. Whole-cell lysates were collected post-sonication in a lysate buffer and analyzed using a BCA assay. Membrane fractions (Mem) were isolated using the Mem-PER Plus Membrane Protein Extraction Kit and quantified via BCA analysis. All lysate concentrations were normalized to the means of the corresponding control. This result corroborates HPG incorporation data, suggesting that the depletion of Lunapark selectively reduces the membrane fraction of proteins.

Reviewer #2 Figure 2. Log₂-scale scatter plot comparing protein-to-mRNA ratios in control (x-axis) versus Lunapark knockout (KO; y-axis) cells. Protein abundance was quantified by mass spectrometry and normalized to corresponding mRNA levels measured by RNA-seq. Each dot represents an individual protein-mRNA pair. The thicker solid diagonal line indicates a 1:1 ratio, while the thinner dashed lines indicate ± 1 log₂ unit differences (representing two-fold changes). Magenta dots indicate genes exhibiting more than a two-fold reduction in protein-per-mRNA abundance in Lunapark KO cells, while blue dots indicate genes with more than a two-fold increase. The enrichment of magenta dots suggests that Lunapark knockout generally decreases membrane protein synthesis efficiency per mRNA.

Reviewer: 2) The PLA experiments supporting preferential proximity of lysosomes to Lunapark-positive ER junctions should be properly controlled by comparing with other ER proteins not found at junctions, such as calnexin, VAPs, and others.

Response: We thank the reviewer for this valuable suggestion regarding our proximity ligation assay (PLA) experiments. To address this concern, we performed additional proximity ligation assay (PLA) experiments using REEP5 as a control. REEP5 is an ER membrane protein not found at three-way junctions and has similar expression levels to Lunapark (REEP5: nTPM=28.3; Lunapark: nTPM=18.8) according to the Human Protein Atlas in U2-OS cells. The

Response to Reviewers

PLA signal between REEP5 and LAMP1 was significantly lower than between Lunapark and LAMP1 (**Fig. 4c** or **Reviewer #1 Fig. 3**), indicating that the increased PLA puncta with Lunapark reflects specific proximity between Lunapark-positive ER junctions and lysosomes, rather than background effects or antibody variations. This finding aligns with previous studies showing LAMP1 enrichment near Lunapark (Yuniati et al., 2020 PMID: 32433973) and BioID data from the Human Cell Map project (<https://cell-map.org>) showing enrichment of lysosomal proteins (RAB9A, STX7) among potential interactors of Lunapark. These results strengthen our conclusion that the proximity between lysosomes and Lunapark-positive ER junctions is specific.

Reviewer #2 Fig. 3. Quantification of PLA spots per cell from anti-EEA1, anti-LAMP1, and anti-TOMM20 against anti-Lunapark and anti-LAMP1 against anti-REEP5. The statistical comparison was performed against anti-LAMP1 PLA. Dunnett's multiple comparisons test. **** P<0.0001.

Reviewer: 3) Fig. 4d: KD of Lunapark causes translating mRNAs to no longer be closely apposed to lysosomes. How Lunapark loss impacts ER-lysosome contacts in general is not assessed, thus the specificity of this effect is unclear. Appropriate fluorescence and ultrastructural data should be provided.

Response: To assess how Lunapark loss impacts ER-lysosome contacts, we employed an optogenetic approach using iLID and SspB proteins, which stably interact when they contact each other upon blue light exposure (PMID:25535392). We co-expressed in cells iLID protein fused to fluorescently tagged LAMP1 to target lysosomal membranes and SspB fused to Sec61 β labeled with a different fluorescent protein to target ER membranes. Since the expressed proteins are transmembrane and tail-anchored proteins, respectively, the extent of proximity/contact of these organelles dictates how fast the recruitment can occur. In control cells, light induced stimulation led to a very rapid, tight colocalization of lysosomes to ER ($t_{1/2}$ =7.9 sec) with maximum co-association resulting in ER tightly wrapped around lysosomes (**Fig. 4e and f** or **Reviewer #2 Fig. 4**, and Supplementary Video 8). In Lunapark KO cells, by contrast, the identical light induction resulted in much slower ER-lysosome co-association ($t_{1/2}$ =26 sec) (**Fig. 4e and f** or **Reviewer #2 Fig. 4**, and Supplementary Video 9). The results suggested, therefore, that Lunapark helps form/maintain ER-lysosome contacts. These new data are presented in Figure 5e of the revised paper.

Response to Reviewers

Reviewer #2 Fig. 4 Schematic representation of a light-activated ER-lysosome recruitment tool using iLID-Halo-LAMP1 (lysosomal marker) and sspB-mCherry-Sec61b (ER marker). **b.** Representative images showing iLID-Halo-LAMP1 (magenta) and sspB-mCherry-Sec61b (green) localization in control (upper panels) and Lunapark KO (lower panels) cells during 488 nm laser activation over 40 seconds. 488 nm laser was continuously on during the start of acquisition to allow recruitment. **c.** Quantification of ER signals within the lysosomal mask over time following 488 nm laser activation to induce ER-lysosome recruitment. Data are shown for control cells (black, $t_{1/2, \text{Control}}=7.9$ seconds) and Lunapark KO cells (magenta, $t_{1/2, \text{LNP-KO}}=26$ seconds), with 95% confidence intervals.

Reviewer: 4) If proximity to lysosomes is key to preferential translation of secretory transcripts, then manipulations that force proximity between ER and lysosomes should specifically rescue translation of membrane and secretory proteins when Lunapark is depleted.

Response: We observed that even when secretome mRNAs are located near lysosomes in Lunapark-depleted cells, there was no translation enhancement of these mRNAs relative to secretome mRNAs far away from lysosomes (Fig. 4h, magenta). This finding suggests that mere proximity to lysosomes is insufficient to restore translation efficiency in the absence of Lunapark. We further found that Lunapark knockdown leads to impaired FRAP of SUNTAG signal from translating secretome mRNA, which could be reversed by bypassing ISR signaling through ISRIB treatment. These observations suggest that Lunapark's role in secretome mRNA translation extends beyond serving as a structural bridge facilitating ER-lysosome proximity, with Lunapark likely involved in translation initiation through regulation of the activity of eukaryotic initiation factor eIF2. We have clarified this thinking in the revised discussion.

Reviewer: 5) Fig.5c lack of effect of rapamycin and, especially, Torin on secretome mRNA translation is quite unexpected as mTOR inhibition has broad, translation-suppressing effects and mTOR signaling is increasingly connected to secretion. Again, this raises a concern that using SunTAG alone is not sufficient to assess translation, and that a more fine-grained type of analysis, such as ribosome profiling, should be employed.

Response: We appreciate the reviewer's insights regarding Figure 5c and the relationship between mTOR inhibition and protein synthesis. While mTOR inhibition broadly suppresses protein synthesis and affects secretory processes (PMID: 33398329, 22552098, 21512002, 35948564), its translational effects are primarily selective for mRNAs containing 5' terminal oligopyrimidine (TOP) motifs, which are largely absent in membrane and secretory protein transcripts (Thoreen et al., PMID: 22552098; Hong et al., PMID: 28650797). Consistent with

Response to Reviewers

this, our reporter assay showed that mTOR inhibitors (rapamycin and Torin) did not substantially affect secretome mRNA translation (Fig. 5d), suggesting any mTOR regulation of secretory protein synthesis would occur primarily through trafficking rather than translation. In contrast, amino acid starvation, which inhibits mTOR activity and triggers ISR, significantly decreased the translation fraction of secretome mRNAs. This translational suppression was reversed by the ISR inhibitor ISRIB (Fig. 5d), indicating that the ISR pathway, rather than mTOR inhibition, primarily mediates the translational repression during amino acid starvation in U-2 OS cells. The ISRIB-mediated restoration of translation suggests that eIF2 α phosphorylation drives this translational repression through mTOR-independent mechanisms. We acknowledge the limitations of relying solely on the SUNTAG system for translation assessment. While this system provides valuable single-molecule resolution of translation dynamics, complementary approaches like spatially-resolved ribosome profiling would offer comprehensive transcriptome-wide insights. Such analysis would require sophisticated methods to label lysosomes-associated ribosomes using APEX or light-activated BioID, however, which extend beyond the scope of this revision time frame.

Reviewer: 6) Inhibiting the ISR is a rather broad manipulation that could affect all substates, not just secretory ones. Can the authors show that, indeed, ISRIB preferentially enhances secretory transcript translation and that this effect depends on ER proximity to lysosomes?

Response: Our data do not suggest that ISRIB treatment preferentially enhances the secretome mRNA translation under normal conditions. Rather, our data suggest that eIF2 activity serves as the rate-limiting factor for efficient translation of membrane and secretory proteins when Lunapark-mediated ER-lysosome contacts are diminished following Lunapark KD and this effect can be overridden by ISRIB. We elaborate below on our rationale for eIF2 being the critical regulatory step at Lunapark-mediated ER-lysosome junctions. Proximity ligation assays (Fig. 4c) and light-activated recruitment experiments (Fig. 4e, f, see **Response #3 to Reviewer #2**) demonstrated Lunapark's close association with lysosomes and its possible role in stabilizing ER-lysosome contacts. These ER-lysosome contacts facilitate secretome mRNA translation, based on our findings showing increased SUNTAG signals near lysosomes, which occurs through a mechanism dependent on (1) Lunapark presence (Fig. 4h(i)), (2) amino acid availability (Fig. 5b, AA-), and (3) translation initiation regulation (Fig. 5b, CrPV-IRES). Furthermore, we observed increased phosphorylation of eIF2 α upon Lunapark depletion, correlating with a reduced translation fraction (Fig. 5c) and impaired translation dynamics (Fig. 5h). This impairment is rescued by ISRIB treatment. Given that ISRIB does not generally enhance translation under normal conditions (PMID:25719440) and does not affect translation dynamics of secretome mRNAs in control cells (see **Response #7 to Reviewer #2**), its rescue effect in our study appears specific to alleviating the translational inhibition triggered by Lunapark depletion. This suggests that the activity of eIF2 is dysregulated under Lunapark depletion.

Reviewer: 7) Fig. 5h-5i, what is the effect of ISRIB alone (w/o Lunapark KD) on SUNTAG recovery? ISRIB can boost translation under baseline conditions as well, thus whether its action represents a true bypass of Lunapark KD or a boost irrespective of Lunapark status should be assessed.

Response: We appreciate the reviewer's question about ISRIB's effects on SUNTAG recovery in cells with normal Lunapark levels. To distinguish between a specific bypass of Lunapark depletion and a general enhancement of translation, we examined ISRIB's effects under baseline conditions. In control cells (without Lunapark knockdown), ISRIB treatment did not significantly alter the recovery kinetics after photobleaching of our cytERM SUNTAG MS2

Response to Reviewers

mRNAs (**Reviewer #2 Fig. 5**). This maintenance of baseline translation levels aligns with previous findings that ISRIB does not enhance translation under unstressed conditions (PMID:25719440). Therefore, ISRIB's rescue effect is specific to conditions of Lunapark depletion and thus represents a true bypass of Lunapark depletion.

Reviewer #2. Fig. 5. Fluorescence recovery after photobleaching (FRAP) curves for cytERM SUNTAG MS2 mRNAs under control/Wildtype conditions (black) and following 1 hr treatment with 200 nM ISRIB (red). The results demonstrate that ISRIB treatment does not significantly alter FRAP recovery dynamics compared to the control condition.

Reviewer: 8) The idea that Lunapark loss results in 'noncanonical' (ATF4- and UPR-independent) ISR is potentially interesting but vague. Lacking a more defined molecular mechanism, one is left wondering whether a general disruption of ER morphology and/or decreased association of translation initiation factors could explain the reduced translation at ER-associated ribosomes.

Response: We appreciate the reviewer's insights regarding potential mechanisms linking Lunapark loss to reduced translation at ER-associated ribosomes. To test whether general ER morphology disruption could explain our observations, we investigated CLIMP63, a protein essential for ER sheet thickness. While CLIMP63 knockdown significantly alters ER morphology (Shibata et al., 2010; PMID: 20637421), it did not significantly reduce the translating mRNA fraction in our SUNTAG reporter system (Extended Fig. 1e). This suggests that the translation defect observed with Lunapark loss is not simply due to altered ER structure. In the revised manuscript's discussion, we now provide a fuller description of the possible roles of Lunapark in membrane and secretory mRNA translation. Specifically, we mention that lysosomes house translation factors that regulate both tRNA charging and eIF2B activity so their proximity to Lunapark-containing ER junctions could ensure a steady supply not only of amino acids but translation factors for efficient membrane and secretory synthesis.

Reviewer: 9) The interpretation of how lysosomal inhibition impacts secretome mRNA translation is largely speculative. pH dissipation has many pleiotropic effects, as no direct test of amino acid release from lysosome is carried out. There are several lysosomal amino acid transporters that the authors could knock down to determine if any of them is required for this process.

Response: We appreciate the reviewer's concern about how lysosomal inhibition impacts secretome mRNA translation. To investigate whether any lysosomal amino acid transporters are required for this process, we performed individual knockdowns of two key lysosomal amino acid transporters, SLC7A5 and SLC38A9. Using our cytERM SUNTAG mRNA reporter system, we found that depleting either transporter alone did not significantly affect the translating fraction of secretory mRNAs (see **Reviewer #2 Fig. 6**). This result may reflect functional redundancy among lysosomal amino acid transporters. Further experiments are thus required to identify the

Response to Reviewers

role of amino transport from the lysosomes. What is nonetheless clear from our results is that the protein degradation function within lysosomes is important to support the translation of transmembrane protein synthesis. While we currently lack tools to directly measure amino acid release at ER-lysosome contact sites, we are working to develop sensitive assays for monitoring local amino acid concentrations in real-time at these interfaces, but this future work requires further technology development.

Reviewer #2. Fig. 6 Boxplots representing the translating fractions of unmodified 5' UTR cytERM SUNTAG MS2 mRNA across different conditions: control (N = 20 cells), SLC7A5 knockdown (KD, N = 35 cells), and SLC38A9 KD (N = 33 cells). Statistical comparisons were performed between each group. The data suggest that the knockdown of a single amino acid transporter does not significantly impair the translation of secretome mRNAs.

Response to Reviewers

Referee #3 (Remarks to the Author):

This is a very interesting manuscript that localizes translation of mRNA coding for secretory proteins to ER junctions enriched for the junction stabilized LunaPark and in proximity to lysosomes. This is intriguing in that it suggests that localized ER domains are involved in secretory protein synthesis and that proximity of these ER domains to lysosomes provides a local source of amino acids to promote protein synthesis. The study uses high resolution imaging approaches to define proximity to ER junctions and to link synthesis of this subset of secretory proteins to lysosomal function.

What is not addressed is the basis for the association of LunaPark enriched junctions with the translation machinery. Is LunaPark acting as a tether to recruit lysosomes, as suggested by the schematic in Figure 5J or is it just serving to stabilize junctions that are then more apt to recruit the translation machinery? While the authors quite elegantly show that an association between Lunapark, lysosomes and secretory protein translation, the underlying molecular mechanisms controlling this interaction are not described? And if this is a general mechanism to synthesize secretory proteins, is it upregulated in professional secretory cells?

Major concerns:

Reviewer: 1) The MS2-SUNTAG system using SPT is a very elegant approach to define translating mRNA based on restricted diffusion. Comparison of diffusion rates of various secretory and cytoplasmic proteins in Fig 1 h was less than convincing with only the artificial Sec61b construct, and not even actin, showing evident more rapid diffusion based on the 0.055 μm^2 MSD cutoff the authors set. This analysis must necessarily be extended to a wider selection of proteins to truly determine if restricted diffusion is specific to secretory proteins.

Response:

In response to the reviewer's suggestion, we expanded our analysis to include a broader range of proteins, incorporating TOMM20 as an additional cytosolic protein and ER5 and ER3 as ER luminal proteins (Extended Data Fig. 1g and Reviewer #3 Fig.1). Although the motion of some fractions of mRNAs for cytosolic proteins were indistinguishable from non-translating secretome mRNAs, those mRNAs encoding membrane and luminal proteins predominantly displayed slow, confined motion. Therefore, while our MSD cut-off of $0.055\mu\text{m}^2$ may not reliably describe the translation state of cytosolic protein mRNAs, it was an effective way for assessing the translation state of the secretome mRNAs encoding membrane and secretory proteins. The slow motion observed for actin is likely due to its binding to stress fibers, which has been previously demonstrated (PMID:26760529, Singer).

Response to Reviewers

Reviewer #3 Fig. 1. Violin plots of $\log(\text{MSD}_{t=1\text{sec}})$ of SUNTAG(+) cytERM SUNTAG MS2 (SUNTAG(+) cytERM, $n=72$), CD4-EGFP MS2 (CD4, $n=251$), SiT-EGFP MS2 (SiT, $n=1616$), Calreticulin-mEmerald MS2 (Calrec., $n=250$), ER5-mEmerald MS2 (ER5, $n=1155$), ER3-mEmerald MS2 (ER3, $n=2475$), beta-actin HaloTag MS2 (Actin $n=100$), or SUNTAG Sec61 β MS2 (Sec61b, $n=314$), TOMM20-Halo MS2 (Tomm20, $n=233$), SiT-EGFP MS2 + Puromycin (SiT+Puro, $n=1608$), and SUNTAG(-) cytERM SUNTAG MS2 (SUNTAG(-) cytERM, $n=680$) where mean squared displacement (MSD) has a unit of μm^2 . The black dotted line indicates MSD at $0.055 \mu\text{m}^2$. Green indicates secretome mRNAs, orange indicates cytosolic protein mRNAs, and purple indicates non-translating secretome mRNAs

Reviewer: Diffusion of slow-moving ribosomes matched that of translated mRNA but how they determined fast, medium and slow-moving population of ribosomes was not clear. It would be important to show whether these populations show differential association with the ER and whether the slow-moving ribosomes are associated with mRNA translation.

Response: To identify distinct ribosome populations with varying diffusion behaviors, we analyzed the cumulative distribution functions (CDFs) of ribosome diffusion coefficients, as described in the Methods section (**Reviewer #3 Fig. 2**). A single diffusion coefficient could not adequately capture the observed data. Instead, models featuring multiple components provided a significantly better fit, indicating the presence of multiple ribosome mobility populations on ER membranes. Given that the measured diffusion coefficients for SiT-EGFP-MS2 ($0.002 \mu\text{m}^2/\text{s}$) and Sec61b mRNA ($0.01 \mu\text{m}^2/\text{s}$) were determined at the same 14 Hz sampling rate, we inferred that more than two ribosome mobility populations coexist. Accordingly, we identified three distinct mobility populations with average diffusion coefficients of $0.3 \mu\text{m}^2/\text{s}$ (fast), $0.04 \mu\text{m}^2/\text{s}$ (medium), and $0.004 \mu\text{m}^2/\text{s}$ (slow) (Extended Data Fig. 3e). Notably, the slow mobility population aligns closely with the diffusion coefficient of translating SiT-EGFP-MS2, whereas the medium population corresponds well to Sec61 β mRNAs. Furthermore, puromycin treatment reduced the medium- and slow-moving populations while increasing the fast-moving fraction (**Reviewer #3 Fig. 3**), suggesting that the medium- and slow-moving subpopulations represent actively translating ribosomes. In the revised paper, we provide additional clarification of the rationale behind this approach and its interpretations.

Response to Reviewers

Reviewer #3 Fig. 2. Cumulative Distribution Function (CDF) of displacements (in μm) for all single-molecule trajectories of L10A-HaloTag labeled with PA-JF dyes, tracked at a frequency of 17 Hz. Black dots (line) depict the displacement data for L10A-HaloTag. The purple, yellow, and red lines represent the fit using a single-population model, two-population, and three-population, respectively. The residual graph of the corresponding fit is shown on the right.

Reviewer #3 Fig. 3. Circular graphs displaying the population distribution of L10A-HaloTag tracked at 17 Hz, based on CDF fitting in control and puromycin-treated cells. Color coding represents different diffusion coefficients: blue for $D=0.004 \mu\text{m}^2/\text{s}$, yellow for $D=0.04 \mu\text{m}^2/\text{s}$, and purple for $D=0.3 \mu\text{m}^2/\text{s}$. These results indicate that the slow (blue) and medium (yellow) moving populations are sensitive to puromycin treatment, suggesting that these populations represent translating ribosomes.

Reviewer: 2) I was confused by the ER-association analysis in Figure 2. It was not clear how ER junctions were identified or how distances of 0.1 and 0.2 μm could be distinguished using what is apparently 3D SIM with a lateral resolution of at best 120nm (Fig 2D). Information on ER labeling and imaging as well as identification of junctions was limited. This needs to be explained.

Response: We thank the reviewer for raising these points. To label the ER membrane, we used Sec61 β tagged with GFP, a widely used ER reporter that localizes to ER membranes but does not interact with the translocon because of its fluorescent tag. We employed a segmentation algorithm followed by skeletonization and manual curation of ER junctions along with ER-bound mRNAs by carefully examining each image at each frame of the movie (**Reviewer #3 Fig. 4**). We have now added this graphical representation to Extended Data Figure 6 to aid readers. With respect to the measured distances of 0.1 and 0.2 μm , we agree that 3D Structured Illumination Microscopy (SIM) has a lateral resolution of approximately 120 nm. However, single-molecule FISH (smFISH) affords a point-detection resolution of <50 nm, enabling us to discriminate these distances. We measured each distance from the center of a single-molecule spot to the nearest manually curated ER junction.

Response to Reviewers

Reviewer #3 Fig. 4 Schematic representation of the image analysis used to measure distances from ER junctions. First, the ER is segmented using WEKA, followed by skeletonization and the identification of branching points. Next, single-molecule tracking of MS2-labeled mRNAs is performed, excluding nuclear signals and categorizing the resulting tracks based on their diffusion coefficients. Finally, the ER and mRNA data are overlaid, and the distance between each mRNA (when bound to the ER) and its nearest ER junction is measured.

Reviewer: The use of FISH analysis to study the effect of puromycin on the distribution of endogenous mRNA for CD9 on the ER was nevertheless interesting (Fig 2D) and not clear why it was not extended to a wide range of other endogenous mRNAs.

Response: We restricted our FISH analysis to CD9 mRNA, encoding an ubiquitously expressed secretory protein, because of the limitations of the FISH technique. FISH analysis cannot reveal the translation status of individual mRNAs, so it required us to employ puromycin to inhibit translation. Although puromycin inhibits translation by causing premature chain termination, some ribosomes (and consequently mRNAs) may remain bound, resulting in a mixture of translating and non-translating mRNAs. Another concern is that the fixation conditions for FISH analysis result in ER structure having much poorer structural detail compared to ER structure in live cells. This might explain why the overall distance measurements from ER junctions were different for MS2 mRNA (measured in live cells) compared to FISH (measured in fixed cells). Given these limitations with FISH, we prioritized our experiments to using MS2 secretome mRNA probes in live cells.

Reviewer: What was further confounding was the apparent association of actin mRNA with the ER even if distance of actin mRNA from junctions matched that of non-translating SiT-EGFP-MS2 reporter construct. Is actin mRNA associated with the ER to the same extent as the other secretory and ER targeted mRNAs? While secretory SiT-EGFP-MS2 should be associated with ER, what is the basis for actin-MS2 mRNA association with ER? Is actin translated on ER-bound ribosomes? These concerns about the choice of actin as a control relate to both this point and the previous one. It would be important to validate for other cytoplasm translated mRNAs relative to secretory mRNAs the actual extent to which they associate with the ER and if their MSD pattern ($< > 0.055$) matches the increased MSD reported for puromycin treated cells or the artificial tail-anchored Sec61 construct.

Response: Previous studies have shown that many cytoplasmic mRNAs, including actin mRNA, can transiently interact with the ER (PMID: 31230715, 25142066). These interactions likely arise from the close physical proximity of the ER to dynamic cytoskeletal structures and the presence of positively charged receptors (e.g., RRBP1) on the ER, which may nonspecifically attract negatively charged ribosomes and RNAs (PMID: 29281824).

Response to Reviewers

Our observations support this view: although we sometimes see stable associations of actin mRNA with the ER, the majority of these interactions are transient (Extended Data Figure 1h). In contrast, secretory and ER-targeted mRNAs consistently show stable ER association, consistent with their co-translational insertion on ER-bound ribosomes. Thus, while actin mRNA can make transient contact with the ER, it is not associated with the ER to the same extent as secretory or ER-targeted mRNAs, and its transient interactions likely reflect nonspecific binding rather than productive ER-bound translation. In addition to actin mRNA, we now added the quantification of sec61b mRNA as another example of a protein that translates in the cytosol.

Reviewer: 3) Fig 3a showing that Lunapark associates with a subset of junctions is not new and would it not be more appropriate to include a large field view showing association between Suntag, mRNA and Lunapark (see also point 5).

Response: We thank the reviewer for their thoughtful feedback. Previous work in COS-7 cells has demonstrated Lunapark's association with a subset of ER junctions (Chen et al., 2015; PMID: 25548161). In Figure 3a, our aim was to confirm and extend these findings to U-2 OS cells, thereby confirming Lunapark localization. To enhance clarity, we have provided wide-field images in Extended Figure 3f, depicting co-localization of the SUNTAG signal, MS2-labeled mRNAs, and Lunapark. These broader views will help readers better appreciate the overall spatial relationships within the cell.

Reviewer: It would also be important to show that translating mRNA is selectively associated with Lunapark junctions and not Lunapark-negative junctions. And if I understand Fig 3C correctly, this means that the percent of association of translating mRNAs (within 300 nm) with Lunapark junctions varies widely (from 50-100%) between cells but consistently shows a 20-30% increase over untranslating mRNA in the same cell.

Response: Our data indicate that approximately 80% of translating (SUNTAG+) mRNAs are located near Lunapark-positive junctions, meaning that fewer than 20% are found on other ER domains such as Lunapark-negative junctions, tubules, or sheets. We now changed our Fig. 3c to show this change in percentage. Although these values were variable between cells (ranging from 50–100% of translating mRNAs associated with Lunapark junctions), we believe this is because the extent that mRNA can stretch out over the range of 300 nm, as discussed below. We have clarified this point in the revised version of Figure 3c in the main text.

Reviewer: While the Lunapark KD data is compelling (Fig 3e), this highly variable association with Lunapark junctions together with the 300 nm cutoff, which is likely beyond molecular interactions, raises questions as to the mechanism by which Lunapark is impacting translation.

Response: Although 300 nm exceeds the typical scale of direct molecular interactions, translating mRNAs can extend over significant distances (up to ~1,700 nm if it were linear for our 3–5 kb reporters). Thus, a 300 nm threshold accommodates the possible spatial extent of these large ribonucleoprotein complexes. Nevertheless, both translating and non-translating mRNAs may occur within or beyond this range, contributing to the variability observed in Figure 3C.

Reviewer: It will be important to determine if direct interaction between Lunapark, mRNA/ribosomes and lysosomes is driving this process.

Response:

Response to Reviewers

We appreciate the suggestion and would like to clarify our findings. First, our data indicate that Lunapark is proximal to translating mRNAs (Fig. 3b and c) and lysosomes (Fig. 4c) independently. Moreover, loss of Lunapark reduces the lysosome proximity-dependent enhancement of translation for our mRNA reporter (Fig. 4h(i)). In addition, we now show that Lunapark knockout reduces ER-lysosome contact formation (**Reviewer #3 Fig. 5**). To demonstrate this, we employed an optogenetic approach using iLID (PMID: 25535392). In this system, a photosensitive domain fused to a lysosomal membrane protein is recruited to an ER-resident protein upon illumination. Because the expressed proteins are transmembrane and tail-anchored, the speed of recruitment reflects the extent of organelle proximity (i.e., contact sites). In control cells, light-induced ER-lysosome contacts result in rapid encapsulation of lysosomes by the ER (Fig. 4e; also see Reviewer #3 Fig. 5 and Supplementary Video 8). In contrast, Lunapark knockout cells exhibited significantly slower ER association with lysosomes even under light-induced conditions, underscoring Lunapark's essential role in maintaining these contacts. We have now incorporated these results into the manuscript (Fig. 4e).

Reviewer #3 Fig. 5 Schematic representation of a light-activated ER-lysosome recruitment tool using iLID-Halo-LAMP1 (lysosomal marker) and sspB-mCherry-Sec61b (ER marker). **b.** Representative images showing iLID-Halo-LAMP1 (magenta) and sspB-mCherry-Sec61b (green) localization in control (upper panels) and Lunapark KO (lower panels) cells during 488 nm laser activation over 60 seconds. 488 nm laser was continuously on during the start of acquisition to allow recruitment. **c.** Quantification of ER signals within the lysosomal mask over time following 488 nm laser activation to induce ER-lysosome recruitment. Data are shown for control cells (black, $t_{1/2, \text{Control}}=7.9$ seconds) and Lunapark KO cells (magenta, $t_{1/2, \text{LNPCK-KO}}=26$ seconds), with 95% confidence intervals.

Reviewer: 4) The PLA data in Fig 4 showing increased association between Lunapark and LAMP1 relative to Tom20 or EEA1 is compelling. The data showing that the SUNTAG proximity to lysosomes is dependent on Lunapark (Fig 4D) is also interesting but requires a more in-depth

Response to Reviewers

analysis to show that these associations are actually occurring at Lunapark-positive ER junctions.

Response: We thank the reviewer for the suggestion to directly visualize the SUNTAG signal on lysosome-proximal mRNAs at Lunapark-marked junctions. While such direct visualization would be ideal, technical constraints have made it unfeasible. Our functional Lunapark construct is GFP-tagged, and efforts to generate red-fluorescent versions (e.g., mApple or mCherry) led to mislocalization. Moreover, the scFv used for SUNTAG detection is also GFP-tagged, preventing concurrent imaging of both targets in the same channel.

Instead, we rely on several complementary observations to draw our conclusion that SUNTAG-labeled mRNAs are enriched near lysosomes at Lunapark-positive ER junctions:

1. Approximately 80% of translating (SUNTAG-positive) mRNAs co-localize with Lunapark-marked ER junctions (Fig. 3b).
2. Lunapark is found near LAMP1-labeled lysosomes (Fig. 4a-c), and lysosome–ER association is reduced upon Lunapark loss (Fig. 4e-f).
3. LAMP1-positive lysosomes are closely associated with SUNTAG-positive mRNAs (Fig. 4h and Extended Data Fig. 4h).
4. The enhancement of lysosome-proximity–dependent translation requires Lunapark (Fig. 4g).

Reviewer: *And the distance scale of lysosome proximity changes is from 1-3 um, contrasting the 40 nm scale of PLA and the 100-300 nm distances used for analysis in previous figures. Rationale for the use of these varying distances for differential analyses needs to be explained.*

Response: Regarding the distance scales used in our analyses, we understand the concern about the varying thresholds. At the molecular level, Lunapark-mediated ER-lysosome interactions presumably occur through protein-protein contacts (<40 nm), while Lunapark-mRNA distance was quantified at a larger scale (<300 nm) due to the size of mRNA molecules (see response to **Reviewer #3 Comment 3**). On the other hand, Lunapark-mediated ER-lysosome contacts appear to establish a ‘translation support zone’ at ER-lysosome contacts, extending ~2 μm from lysosomes. We believe this large range occurs because the release of amino acids from lysosomes (or signaling molecules) likely creates a diffusion gradient extending beyond the immediate vicinity of the lysosome. We believe that amino acids released from lysosomes can influence translation efficiency over this larger spatial scale, resulting in enhanced translation of mRNAs located within this gradient. We updated our Discussion to clarify these points.

Reviewer: *Finally, how do the authors explain that in about half the cells they analyze there is no difference between distance between translating and untranslating mRNA and lysosomes (Fig 4e) – and in those cases distance of non- translating mRNA is very close to lysosomes. This analysis lacks in that it does not focus on those lysosomes in proximity to Lunapark ER junctions.*

In Figure 4e, we employed a different method from Figure 4d by determining translation status based on the diffusion of SiT-EGFP-MS2-labeled mRNAs. This technique offers a binary readout—classifying mRNAs simply as translating or non-translating based on their motion patterns instead of by measuring the number of ribosomes attached captured by the SUNTAG signal. Since our SUNTAG data indicate that translation can occur away from lysosomes, albeit without the enhanced translation seen near lysosomes, the binary diffusion-based approach may be less sensitive to this localized enhancement. This likely explains why, in about half of

Response to Reviewers

the cells analyzed, there was no observed difference in lysosome proximity between translating and non-translating mRNAs.

Reviewer: The idea that proximity between Lunapark ER junctions, lysosomes and translating mRNA/ribosomes needs to be shown beyond the zoomed-in images provided in Fig 4 C (or for that matter 3B) for multiple channels similar to the larger cell views shown in Fig 1A and Ext Figure 3.

Response: We now provided this in the supplementary figure.

Reviewer: A multi-parameter quantitative interaction analysis, as opposed to the binary interaction analyses provided, between the various players is also required.

Response: We appreciate the reviewer's suggestion regarding a multi-parameter quantitative interaction analysis. In our study, the analysis of SUNTAG intensity versus mRNA–lysosome distance effectively serves this purpose, as it simultaneously quantifies ribosome loading (via SUNTAG intensity) and the spatial relationship to lysosomes. Moreover, by integrating conditions such as KD, amino acid starvation, and the cricket paralysis IRES, we are able to dissect the contributions of translation initiation, lysosome-mediated amino acid processing, and Lunapark's role in these interactions.

Reviewer: 5) The demonstration that disrupting lysosomal activity impacts translation is strong and provides functional relevance to the proximity analyses. The close association between translated mRNA and lysosomes upon amino acid starvation supports a role for autophagy in this process and this relationship should be explored further.

Response: We appreciate the reviewer's recognition of our findings regarding the impact of lysosomal activity on translation and its functional relevance. While our data showing a close association between translated mRNA and lysosomes during amino acid starvation hint at a potential role for autophagy, we would like to explore this relationship in greater depth in the future since it represents an interesting direction for future studies. For instance, exploring whether macroautophagy, microautophagy, or endocytic uptake is primarily responsible for supplying amino acids near lysosomes for translation would be an interesting next step.

Reviewer: As indicated above, physiological significance for this process in professional secretory cells (or other models) would be important to show the physiological relevance of the mechanism and strengthen the manuscript for Nature.

Response: We also appreciate the suggestion to investigate the physiological significance of this phenomenon in professional secretory cells. While we currently lack assays to directly measure translation in these cell types, we do have preliminary evidence suggesting a role for Lunapark. Specifically, during the differentiation of naïve B cells into plasmablasts, which produce and secrete substantial amounts of antibodies, Lunapark mRNA levels increase approximately threefold (Human Protein Atlas, Reviewer #3 Fig. 6). This upregulation implies that Lunapark could contribute to enhancing secretory protein translation in professional secretory cells. Although indirect, these observations point to a broader physiological relevance of our findings and underscore the potential importance of investigating how cells regulate protein synthesis in response to secretory demands.

Response to Reviewers

Reviewer #3 Fig. 6. Bar graph illustrating the normalized Transcripts per Million (nTPM) of Lunapark in naïve B cells and plasmablasts, based on data from the Protein Atlas. The graph compares the expression levels of Lunapark in these two cell types, providing insights into its differential expression during B cell differentiation.

Minor issues:

Reviewer: *There is a significant amount of data lacking statistical analysis: Fig 1 BGH , 4D, 5BI*

Response: We have updated our figures with statistical tests.

Reviewer: *Controls for the PLA experiment need to be provided as extended data.*

Response: To address this concern, we performed additional proximity ligation assay (PLA) experiments using REEP5 as a control. REEP5 is an ER membrane protein not found at three-way junctions and has similar expression levels to Lunapark (REEP5: nTPM=28.3; Lunapark: nTPM=18.8) according to the Human Protein Atlas in U2-OS cells. The PLA signal between REEP5 and LAMP1 was significantly lower than between Lunapark and LAMP1 (Fig. 4c or **Reviewer #3 Fig. 7**), indicating that the increased PLA puncta with Lunapark reflects specific proximity between Lunapark-positive ER junctions and lysosomes, rather than background effects or antibody variations. This finding aligns with previous studies showing LAMP1 enrichment near Lunapark (Yuniati et al., 2020 PMID: 32433973) and BioID data from the Human Cell Map project (<https://cell-map.org>) showing enrichment of lysosomal proteins (RAB9A, STX7) among potential interactors of Lunapark. These results strengthen our conclusion that the proximity between lysosomes and Lunapark-positive ER junctions is specific.

Reviewer #3 Fig. 7. Quantification of PLA spots per cell from anti-EEA1, anti-LAMP1, and anti-TOMM20 against anti-Lunapark and anti-LAMP1 against anti-REEP5. The statistical comparison was performed against anti-LAMP1 PLA. Dunnett's multiple comparisons test. **** P<0.0001.

Response to Reviewers

Reviewer: What is the rationale for using CLIMP-63 KD as a control and not other ER shaping proteins such as reticulon or atlastin that might have more relevance to junction dynamics?

Response: A previous study indicates that ER junctions consist of a tubular network with tubules interconnected by small, equilateral, triangular sheets (PMID:25404289). Given that CLIMP-63 has been implicated in generating ER sheets and possibly rough ER formation (PMID:7673362, 21111237) and has also been observed at ER junctions (PMID:35471903), we investigated whether CLIMP-63 could influence the translation of secretome mRNAs. In contrast, although reticulons and atlastins are involved in regulating ER junctions, they are not highly enriched at these sites and are not proposed to control the level of rough ER. Therefore, we did not use markers for these proteins in our experiments.

Reviewer: Presentation of cropped Western blots (Fig 5e, Extended data 4) is not appropriate.

Response: We have updated the western blot images to show the full blot instead of cropped western blots in Supporting Document.

Reviewer: The photobleaching analysis with ISRIB requires a control with ISRIB on control cells.

Response: Thank you for this insight comment. We repeated the ISRIB treatment experiment on control cells and observed no difference in recovery. Therefore, ISRIB does not boost translation in control conditions (**Reviewer #3 Fig. 8**).

Reviewer #3 Fig. 8 Plot of FRAP recovery curves from control (black) and control + 200 nM ISRIB (red). The shaded areas represent 95% confidence intervals for each plot.

Reviewer: Availability of the custom MatLab code used should be indicated and provided.

Response: This will be available at our github website.

Reviewer: I was unable to download and view the supplementary videos.

Response to Reviewers

Response: We apologize for these issues. When we tested at our end, we were able to download and view supplementary movies. We will coordinate with the editor to see if movies can be directly updated into a folder where download is possible.

Referees' comments:

Referee #1 (Remarks to the Author):

The authors have addressed most of my concerns. One remaining point that has not been addressed to satisfaction is the translation elongation rate on the reporter mRNAs. In the first version of the manuscript, the authors show by FRAP that ribosomes turn over in on average ~70 sec, which seemed very fast to reflect the translation elongation rate. In the revised manuscript, the authors use harringtonine-run off experiments to show that ribosomes translate at 6 aa/s and thus that the SunTag signal turnover determined in FRAP experiments (~70 sec) is consistent with the elongation rate (note that even at 6 aa/sec, a FRAP rate of 70 sec is faster than expected if one calculates this, see my previous comments).

However, the authors also state that they only used a subset of mRNAs (n=11) in the analysis to determine the elongation rate in harringtonine run off experiments, because they state that many mRNAs show slower run-off and these mRNAs were apparently excluded from the analysis to derive the 6 aa/sec elongation rate. Unfortunately, they don't provide any information on what fraction of mRNAs shows a slower run-off and they also don't show the average elongation rate of ALL mRNAs (including the mRNAs with slower run-off rates). This approach of only using the mRNAs with fast run-off rates seems incorrect to me, because in the FRAP analysis ALL mRNAs are analyzed, so if the authors want to state that the run-off rate in the FRAP analysis is consistent with the elongation rate as determined by harringtonine run-off, then they should compare the FRAP rates to the harringtonine run-off rate of all mRNAs too. I suspect that if they do this, the values of the two measurements will not align and that the FRAP data shows a much higher turnover rate than expected based on the harringtonine run-off, especially considering that 70 sec is already faster than expected even based on the 6 aa/sec elongation rate value.

Moreover, it is important to know what fraction of mRNAs in the reporter used in this study runs off slowly or has stalled ribosomes. Previous studies with the translation imaging reporter have shown ~10-20% of mRNAs with slow run-off, do the authors find a higher fraction than that? If many ribosomes are stalling on the reporter, that would represent a problem in interpreting much of the live-cell imaging data using GFP intensities to determine translation initiation rates (as is done throughout the study).

We thank the reviewer for their thoughtful comments. We agree that a comparison between FRAP recovery times and elongation rates derived from harringtonine run-off experiments from all translating mRNAs would be valuable. However, we found that extended harringtonine treatment induces ER stress responses that undermine the validity of this direct comparison, which has been previously documented in PMID:26051182. Similarly, within 15 minutes of harringtonine treatment, we observed aggregation of SUNTAG puncta near the nuclear periphery, progressive loss of MS2 signal, and substantial remodeling of ER morphology. Although $81.7 \pm 2.2\%$ of SUNTAG-positive mRNAs were cleared within 30 minutes in the harringtonine experiments, these stress-associated changes introduce secondary effects that confound interpretation and prevent a fair comparison with our FRAP measurements. To avoid potential misinterpretation of our results and in consideration of the reviewer's concern, we have removed the harringtonine run-off experiments from the manuscript and revised the relevant section of the Results on the interpretation of the SUNTAG FRAP experiments (Line 348-362).

We now limit this section to the finding that ISRIB-treatment reverses LNPK KD's inhibitory effect on SUNTAG recovery after photobleaching.

Referee #2 (Remarks to the Author):

I am in support of acceptance of this very interesting and aptly revised work. The additional data provided make a stronger case for Lunapark acting as a nexus between ER, translating ribosomes and lysosomes, an activity that is required for efficient translation of secreted/transmembrane proteins.

However, a few points should still be addressed.

1- Response to point 7: the ctrl + ISRIB condition should be included in Fig 5h. There is no reason to exclude this important control (requested by all three reviewers) from the main data display.

We agree that including the control + ISRIB condition in Fig. 5h would enhance clarity and completeness, particularly as this control directly demonstrates the specificity of the ISRIB treatment in Lunapark KD condition. In response, we have updated Fig. 5h to include the fluorescence recovery montage for the control + ISRIB condition, which shows recovery kinetics that mirror those of the untreated control.

2- Response to point 9: the response provided is especially unsatisfying. The data provided (lack of effect of SLC38A9 or SLC7A5 kd, the latter, by the way, being a plasma membrane, not lysosomal AA transporter) do not strengthen the case for lysosome-released amino acids promoting translation. Chloroquine and protease inhibition could have literally hundreds of downstream consequences. The interpretation of these results should be broadened to include a general requirement for lysosomal function beyond providing amino acids via proteolysis, and the sentence 'local amino acid release by lysosomes' should be removed from the abstract.

We thank the reviewer for pointing out the need for more careful interpretation. We agree that the current data do not conclusively demonstrate that lysosome-derived amino acids are the direct source fueling local translation. While our data are consistent with a model in which lysosomal function promotes translation of secretome mRNAs near lysosomes, the precise molecular mechanism (for example, via amino acid release, regulation of translation factors, or local activation of signaling pathway) remains to be elucidated.

In response, we have revised the relevant portions of the text to reflect a broader interpretation.

Specifically:

- The sentence "local amino acid release by lysosomes" has been removed from the Abstract, which now states:
"...implying a linkage between local protein synthesis on ER and local lysosomal activity."

3- I also agree with Reviewer 1 that the result in Fig. 5e is inconsistent with the proposed mechanism of ISR induction downstream of Lunapark ablation. Part of the problem is that the WB provided is of very poor quality, and will undoubtedly confuse the reader. Control vs KD samples should be run on the same gel, and presented side-by-side in its entirety, without the cropping of single bands used here.

We agree with the reviewer that the Western blot in Fig. 5e did not meet the clarity and rigor expected, especially for such a key mechanistic point. We have now repeated the experiment, with three independent lysates from control and Lunapark KO on the same gel and presented the full blot with all relevant bands visible side-by-side, without cropping. The revised blot confirms the elevated ratio of phospho-eIF2 α to total eIF2 α in Lunapark KO cells, and is now shown in Fig. 5e-g and Extended Data Fig. 5a-d.

Our reasoning behind the divergence from classical ISR signaling is that the Lunapark KD effect is highly selectively to membrane and secretory protein mRNAs. Instead of increasing the phosphorylated eIF2 α as occurs under UPR or amino acid starvation, the effect led to decreased overall levels of eIF2 α without affecting the concentration of phosphorylated eIF2 α , which increases the relative abundance of phosphorylated eIF2 α . This occurred without the expression of ATF4, which is a classical marker for ISR. We now revised our document in line 332-345 to clarify these points.

Referee #3 (Remarks to the Author):

This revised manuscript includes a detailed response to previous comments and has addressed the majority of the previous issues. The demonstration that LunaPark ER junctions are associated with local translation in proximity to lysosomes is highly original and represents a compelling finding underlining a novel mechanism of local regulation of translation. There remain a few issues that need be addressed:

1. In response to a previous comment requesting wide-field images, extended Fig 3F was provided. It is very difficult to appreciate from these images that 80% of Suntag-positive mRNA puncta are associated with Lunapark and perhaps the authors could use the space in the extended figure to better highlight exactly what they are quantifying.

We appreciate this feedback and agree that the current presentation in Extended Data Fig. 3F does not clearly convey the colocalization data. We updated the Extended Data Fig. 3F by circling the examples of translating (green) and non-translating cytERM SUNTAG mRNAs (magenta) on each channel images to clarify their colocalization.

2. Use of the light-activated ER-lysosome recruitment tool is convincing and shows that lunapark regulates proximity between the two organelles. Cannot this experiment be done in the presence of LNPK-GFP to show that this association occurs at lunapark positive ER junctions. The Halo and mCherry dyes should be compatible with LNPK-GFP which was raised as an issue in response to my request to show Suntag/mRNA association with Lunapark. Or perhaps

use the ER skeletonization approach used elsewhere to show that ER-lysosome association occurs at junctions?

We appreciate the reviewer's suggestion and fully agree that demonstrating that ER-lysosome contacts occur specifically at Lunapark-positive junctions would strengthen our conclusion.

Unfortunately, the iLID-based optogenetic system we use to induce ER-lysosome proximity is activated by 488 nm light, which is the same wavelength required to excite GFP. Thus, checking the expression of LNPK-GFP in this assay would inadvertently trigger light-induced recruitment prior to imaging and confound the interpretation. We attempted a workaround by imaging at minimal exposure, but even low levels of GFP excitation triggered premature lysosome-ER contact formation, precluding reliable colocalization analysis in this context.

Our previous work using skeletonization-based ER mapping to quantify ER-lysosome contacts (PMID:30454650, Fig. 5) showed that large polygonal ER domains can constrict into smaller corrals that tightly surround the late endosome or lysosome, thereby creating clusters of ER junctions closely associated with lysosomes. However, this skeletonization approach is not applicable in Lunapark knockout cells, as the ER adopts a more sheet-like morphology with few discernible junctions. We thus rely on our current data using PLA and light-inducible recruitment assays to draw the conclusion that Lunapark facilitates ER-lysosome contacts.

3. In response to concerns about the eIF2 blots, the original blots were included. While the bands shown in the figures do come from the same blot it is very difficult to ascertain a twofold increase in p-eIF2alpha/eIF2alpha from these blots. Also annotation of the other lanes in the gels is missing. As far as I can tell both the eIF2alpha and the p-eIF2 alpha bands show decreased intensity upon lunapark KD, while in extended Fig 5C both show an increase upon lunapark KD. While these differences can be attributed to gel loading and ratiometric quantification does appear to be based on 5 replicates, the data is not overwhelming. I am not convinced that it is sufficient to support the conclusion in 5i invoking eIF2 translation or that "the pathway for suppression of eIF2 α -dependent translation initiation in LNPK KD cells diverges from classical ISR signaling".

We appreciate the reviewer's careful scrutiny of our data. We agree that the blot in its prior form was suboptimal and that the data should be presented with greater clarity. To address this, we repeated the western blot analyses and now provide side-by-side comparisons in Fig. 5e-g and Extended Data Fig. 5a-d.

These revised experiments consistently show that depletion of LNPK results in a reduction of total eIF2 α levels without a proportional change in phosphorylated eIF2 α (Fig. 5e-g, Extended Data Fig. 5a-d). This shift increases the relative proportion of p-eIF2 α in the translation initiation pool. Functionally, the translation defect observed in LNPK knockdown or knockout cells can be rescued by ISRIB (Fig. 5c and 5i), which counteracts the inhibitory effect of p-eIF2 α on translation initiation. This rescue supports the interpretation that the translational inhibition involves the eIF2 α dependent translation regulatory pathway, a key mechanism in the integrated stress response (ISR).

However, we believe this pathway diverges from classical ISR signaling for three reasons. First, classical ISR activation generally causes a broad reduction in global translation, whereas LNPk depletion selectively impairs the translation of membrane and secretory protein mRNAs (Fig. 3e, 3f, and Extended Data Fig. 4b). Second, canonical ISR markers such as ATF4 are not upregulated in LNPk-depleted cells (Extended Data Fig. 5c-d). Third, transcriptome data do not reveal the characteristic gene expression signature associated with classical ISR activation (Extended Data Fig. 4c). Thus, although ISRIB can rescue the translational defect in both cases, LNPk depletion appears to act through a distinct mechanism resulting in a selective translational effect rather than a generalized suppression.

4. While Fig 5g is convincing, in Fig 5h the extent of photobleaching is higher for the LNPk KD treatment making it difficult to appreciate the difference between the treatments. The $t_{1/2}$ and mobile fraction should be quantified to show that the effect is significant.

We appreciate the reviewer's suggestion. $t_{1/2}$ is a meaningful parameter in systems exhibiting exponential FRAP recovery like diffusional recovery; however, SUNTAG FRAP recovery typically does not follow exponential kinetics. Instead, it is often analyzed over the linear portion of the recovery curve, as described in prior studies (e.g., Pichon et al., *J Cell Biol*, 2016; Morisaki et al., *Cold Spring Harb Perspect Biol*, 2018). In our system, however, recovery deviates from linearity, likely due to heterogeneity in elongation dynamics among ribosomes within individual polysomes (as mentioned in Morisaki et al., *Cold Spring Harb Perspect Biol*, 2018). This is probably attributed to SRP-dependent pausing events that affect the translation of secretome mRNAs as well as insertion into the membrane, which presumably follow different elongation kinetics, making curve fitting and parameter extraction more complex.

Nonetheless, in response to the reviewer's request, we quantified both the $t_{1/2}$ and mobile fraction across conditions: Control = 55 sec (~100% mobile), LNPk KD = 56 sec (56% mobile), Control + ISRIB = 56 sec (~100% mobile), and LNPk KD + ISRIB = 40 sec (80% mobile). These values support the conclusion that Lunapark depletion impairs translation dynamics, and that ISRIB treatment partially restores LNPk KD dependent translation impairment. We now revised our document in Line 346-362.

5. Finally, the data provided showing increased lunapark transcripts in activated B cells is intriguing and data supporting a role for lunapark in antibody secretion from activated B cells would be a very nice addition to the paper.

We appreciate the reviewer's interest in the B cell data. While we agree that functional studies of Lunapark in antibody-secreting cells would be exciting, such experiments are beyond the scope and focus of the current study, which centers on mechanistic insights into ER-lysosome-translation coupling in U-2 OS cells. We have therefore chosen to include the Lunapark expression in differentiating B cells in the discussion as a potential avenue for future work.